# Platelet transcription factors license the pro-inflammatory cytokine response of human monocytes

Ibrahim Hawwari [1,6]✉, Lukas Rossnagel[1,6], Nathalia Rosero[1], Salie Maasewerd [1], Matilde B Vasconcelos[1], Marius Jentzsch[2], Agnieszka Demczuk [1], Lino L Teichmann[3], Lisa Meffert[3], Damien Bertheloot [1], Lucas S Ribeiro [1], Sebastian Kallabis[1], Felix Meissner[1], Moshe Arditi [4,5], Asli E Atici[4,5], Magali Noval Rivas [4,5] & Bernardo S Franklin [1]✉

## Abstract

In humans, blood Classical CD14[+] monocytes contribute to host defense by secreting large amounts of pro-inflammatory cytokines. Their aberrant activity causes hyper-inflammation and life-threatening cytokine storms, while dysfunctional monocytes are associated with 'immunoparalysis', a state of immune hypo responsiveness and reduced pro-inflammatory gene expression, predisposing individuals to opportunistic infections. Understanding how monocyte functions are regulated is critical to prevent these harmful outcomes. We reveal platelets' vital role in the pro-inflammatory cytokine responses of human monocytes. Naturally low platelet counts in patients with immune thrombocytopenia or removal of platelets from healthy monocytes result in monocyte immunoparalysis, marked by impaired cytokine response to immune challenge and weakened host defense transcriptional programs. Remarkably, supplementing monocytes with fresh platelets reverses these conditions. We discovered that platelets serve as reservoirs of key cytokine transcription regulators, such as NF-κB and MAPK p38, and pinpointed the enrichment of platelet NF-κB2 in human monocytes by proteomics. Platelets proportionally restore impaired cytokine production in human monocytes lacking MAPK p38α, NF-κB p65, and NF-κB2. We uncovered a vesicle-mediated platelet-monocyte-propagation of inflammatory transcription regulators, positioning platelets as central checkpoints in monocyte inflammation.

**Keywords** Hyperinflammation; Immunoparalysis; Immune Thrombocytopenia; Toll-like Receptors; Inflammasomes
**Subject Categories** Haematology; Immunology

## Introduction

Classical CD14[+]CD16[−] monocytes constitute about 85% of the circulating monocyte pool in human blood. Compared to other subsets, classical monocytes are adeptly prepared for host defense programs, excel in innate immune detection of pathogens, and display an increased capability to secrete pro-inflammatory cytokines (Wong et al, 2011). Monocyte-driven innate immune responses can yield different extreme outcomes. While aberrant monocyte activity triggers hyper-inflammation and cytokine storms (Bonnet et al, 2021; Fajgenbaum and June, 2020; Ferreira et al, 2021; Jafarzadeh et al, 2020; Junqueira et al, 2022; Schulte-Schrepping et al, 2020; Vanderbeke et al, 2021), dysfunctional monocytes contribute to a life-threatening state of hypo-responsiveness, known as 'immunoparalysis', which impairs defense against opportunistic infections. Immunoparalysis is often observed after sepsis (Arens et al, 2016; Frazier and Hall, 2008; Roquilly et al, 2020), major visceral surgery (Frazier and Hall, 2008), and has recently been associated with the severity of SARS-CoV-2 infections (Agrati et al, 2020; Arunachalam et al, 2020). Monocytes are also vital cells mediating "trained immunity", a series of long-lasting epigenetic and metabolic adaptations that enhance innate immune responsiveness upon subsequent encounters with pathogen molecules (Bekkering et al, 2014; Netea et al, 2016). Hence, understanding the mechanisms regulating monocyte functions is exceptionally relevant in diverse clinical settings, as inappropriate monocyte activities can have long-lasting immunological consequences (Agrati et al, 2020; Arunachalam et al, 2020; Bonnet et al, 2021; Ferreira et al, 2021; Jafarzadeh et al, 2020; Junqueira et al, 2022; Schulte-Schrepping et al, 2020; Vanderbeke et al, 2021).

Here, we reveal a crucial role for platelets in the effector pro-inflammatory cytokine functions of human and mouse monocytes. In the bloodstream, monocytes continuously interact with platelets, forming monocyte-platelet aggregates (MPAs) under physiological conditions (Rinder et al, 1991). MPAs increase in numerous

[1]Institute of Innate Immunity, Medical Faculty, University of Bonn, Bonn, Germany. [2]Institute of Clinical Chemistry and Clinical Pharmacology, University Hospital Bonn, University of Bonn, Bonn, Germany. [3]Department of Medicine III, University Hospital Bonn, Bonn, Germany. [4]Department of Pediatrics, Division of Pediatric Infectious Diseases, Guerin Children's, Cedars Sinai Medical Center, Los Angeles, CA, USA. [5]Infectious and Immunologic Diseases Research Center (IIDRC), Department of Biomedical Sciences, Cedars-Sinai Medical Center, Los Angeles, CA, USA. [6]These authors contributed equally: Ibrahim Hawwari, Lukas Rossnagel. ✉E-mail: ihawwari@uni-bonn.de; franklin@uni-bonn.de

inflammatory and thrombotic disorders and are usually associated with poor outcomes (Allen et al, 2019; Cervia-Hasler et al, 2024; Liang et al, 2015; Maher et al, 2022; Manne et al, 2020; Stephen et al, 2013). The interaction with platelets enhances various monocyte functions, with implications in innate and acquired immunity (D'Mello et al, 2017; Fu et al, 2021; Han et al, 2020; Rong et al, 2014; Singhal et al, 2017) through partially understood mechanisms (Kral et al, 2016). Nevertheless, the physiological functions of platelets or MPAs to monocyte innate functions remain undefined. Moreover, studies in the human system are lacking and were primarily done in conditions where platelets are added to monocytes or describing pathological conditions where MPAs are enhanced.

We revealed a previously unrecognized critical dependency on platelets for the pro-inflammatory cytokine production of human monocytes. Using different human monocyte isolation kits and "untouched" monocytes from patients with primary immune thrombocytopenia (ITP), an autoimmune disease characterized by low blood platelet counts, and in vivo platelet depletion in mice, we demonstrate that platelet numbers directly impact the cytokine output of CD14$^+$ monocytes towards Toll-like receptor (TLR) and Nod-like receptor (NLR) stimulation. Removal of platelets from healthy monocytes caused monocyte immunoparalysis, characterized by transcriptional silencing of pro-inflammatory genes and impaired capacity to secrete pro-inflammatory cytokines. Notably, monocyte immunoparalysis is dynamic and can be reversed by monocyte replenishment with fresh platelets. Moreover, monocytes from ITP patients were inherently impaired in their capacity to produce cytokines in response to TLR and NLR stimulation. Remarkably, supplementation with fresh platelets reactivated the immune functions of ITP monocytes, reinvigorating their cytokine responses.

Mechanistically, we show that platelets are abundant cellular sources of transcription factors (TFs) known to regulate the production of pro-inflammatory cytokines, including the NF-κB subunits p65 (RelA), p52/p100 (NF-κB2) and p38 MAPK. Furthermore, using Stable Isotope Labeling with Amino acids in Cell culture (SILAC) combined with high-resolution mass spectrometry (MS)-based proteomics, and loss- and gain-of-function experiments, we demonstrated the enrichment of platelet TFs in human monocytes exposed to megakaryocyte and platelet vesicles. We also revealed the platelet's capacity to bypass pharmacological or genetic ablation of p38 MAPK, RelA, and NF-κB2 in human monocytes and restore their impaired cytokine secretion in a platelet:monocyte ratio-dependent manner.

Our findings elucidate the pivotal role of platelet-derived TFs in regulating monocyte pro-inflammatory activity, suggesting the potential therapeutic application of platelet supplementation against monocyte immunoparalysis. Our research challenges the earlier notion of monocytes being autonomous immune cells, unveiling their reliance on platelets for optimal pro-inflammatory cytokine production.

# Results

## Platelets are required for cytokine secretion by human monocytes

Hyper-inflammation, resulting from inflammasome activation in monocytes, leads to excessive production of pro-inflammatory

cytokines (Junqueira et al, 2022; Rodrigues et al, 2021). To thoroughly analyze the inflammasome-mediated cytokine response in human monocytes, we isolated classical monocytes (CD14$^+$CD16$^-$) using immune-magnetic isolation (Fig. EV1 and Methods). Consistent with previous studies (Bhattacharjee et al, 2017; Han et al, 2020; Rolfes et al, 2020), monocytes purified through standard magnetic isolation methods, termed "standard monocytes" (StdMo), comprised a mixture of platelet-free monocytes, monocyte-platelet aggregates (MPAs), and free platelets (Fig. EV1A–D). Employing a kit-provided platelet-depleting antibody to remove platelets from StdMo allowed us to obtain platelet-depleted monocytes (PdMo) while maintaining stable monocyte counts (Fig. EV1A–C). PdMo remained viable and were phenotypically identical to StdMo (Fig. EV1A). However, upon exposure to lipopolysaccharide (LPS) or LPS + nigericin (LPS + Nig), the secretion of IL-1β, TNFα, and IL-6 in PdMo drastically declined compared to StdMo, highlighting an essential role of platelets in monocyte cytokine production (Figs. 1A,B and EV1E). Notably, reintroducing autologous platelets to PdMo (PdMo + Plts) revitalized PdMo's impaired cytokine responses. Importantly, platelets alone (Plts) did not secrete any of the measured cytokines, confirming monocytes as their primary source (Figs. 1A,B and EV1E). As we did not observe phenotypic differences between StdMo and PdMo based on cell surface expression of CD14 and CD16 (Fig. EV1A), we next examined whether the differences in LPS responsiveness of PdMos were influenced by availability of LPS receptors (Tatematsu et al, 2016). We assessed the expression dynamics of the LPS receptors TLR4 and CD14 in PdMo vs StdMo over time by immunostaining and Flow Cytometry (Fig. EV1F–H). Expression of CD14 and TLR4 was comparable in all time points between StdMo and PdMos, indicating that the impaired response of PdMos was not due to differential LPS receptor availability.

Activation of inflammasome sensors, such as NLRP3, nucleates the assembly of the adapter ASC into micron size aggregates (ASC specks) which function as signaling platforms to maximize the activation of the cystine protease Caspase-1, the maturation of IL-1 cytokines, and the triggering of pyroptosis, a highly inflammatory type of cell death (Bertheloot et al, 2021). ASC specks are a well-established readout for inflammasome activation (Fernandes-Alnemri et al, 2007; Franklin et al, 2014). We therefore imaged and comparatively quantified ASC speck formation in StdMo, or platelet-depleted primary human monocytes (PdMo). Confirming an impaired inflammasome activation, PdMos displayed decreased rates of ASC speck formation compared to StdMos (Fig. 1C,D; Appendix Fig. S1). Mirroring the production of IL-1 cytokines, formation of ASC specks was restored in PdMos added with platelets (Fig. 1C,D). Consistent with caspase-1 activation and pyroptosis downstream of inflammasome activation, nigericin-treated StdMo displayed increased caspase-1 activity (Fig. 1E) and reduced cell viability (Fig. EV1I). However, platelet depletion (PdMo) led to decreased caspase-1 activity (Fig. 1E), along with decreased intracellular maturation of IL-1β (Figs. 1B and EV1E) while increasing cell viability in response to nigericin (Fig. EV1I). Re-supplementation of PdMos with platelets (PdMo + Plts) restored caspase-1 activity, IL-1β maturation (Fig. 1B), and release (Figs. 1A and EV1E) and re-sensitized PdMo to nigericin-induced pyroptosis (Fig. EV1I). These findings reveal the dependency on platelets for the monocyte inflammasome response.

To validate our findings using alternative methods, we juxtaposed two popular and broadly used techniques for isolating

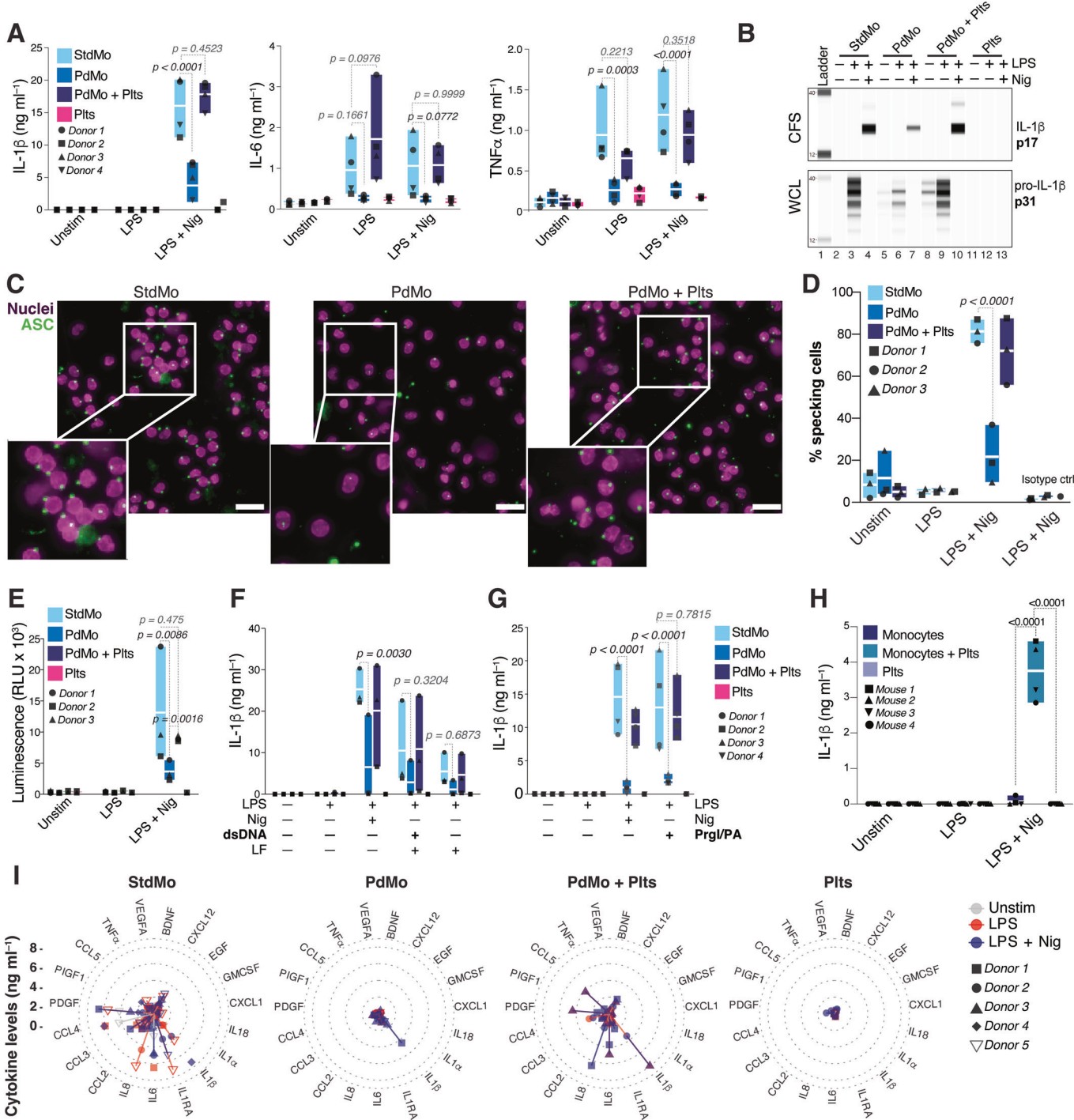

CD14+ monocytes: positive and negative selection. We found different degrees of platelet contamination between positive vs. negative selection of CD14+ monocytes (Fig. EV2). Positively selected StdMo contained fewer platelets (Fig. EV2B,C) but responded poorly to TLR and inflammasome stimulation than negatively isolated StdMo, which had significantly more platelets (Fig. EV2D). The impairment of CD14-positively-selected monocytes was not due to pre-engagement of this receptor by the magnetic beads (Bhattacharjee et al, 2017), as these cells also

displayed reduced cytokine response to Pam3CysK4, a TLR1/2 agonist. Notably, supplementing positively selected monocytes with platelets enhanced their cytokine responses (Fig. EV2D). We next recapitulated these findings in mouse monocytes isolated from mouse PBMCs. We employed FICOLL-isolation of PBMCs to minimize neutrophil contamination followed by magnetic positive selection to isolate CD11b+ Ly6C+ monocytes (Fig. EV2E). Equivalently to positively isolated human monocytes (Fig. EV2B) positive isolated mouse monocytes were low on platelets

**Figure 1. Platelets regulate the cytokine secretion of human monocytes.**

(A) Concentrations of IL-1β, IL-6, and TNFα released into the cell-free supernatants (CFS) by untouched (StdMo, light blue), platelet-depleted (PdMo, blue), or PdMo that were supplemented with autologous platelets (PdMo + Plts, dark blue) at a 100:1 platelet:monocyte ratio. Cytokine levels secreted by platelets alone (Plts, purple) were measured as control. Cells were stimulated with LPS (2 ng ml$^{-1}$ for 3 h) followed by activation with nigericin (10 μM for 1.5 h). Data is combined from independent experiments ($n = 4$). (B) Intracellular maturation and secretion of IL-1β assessed by WES capillary electrophoresis coupled with Ab-based detection in whole cell lysates (WCL) or CFS of primary human monocytes and platelets stimulated as in (A). (C, D) Representative images of immunostaining and confocal imaging of human ASC (C), and quantification of ASC speck formation (D) in StdMo, PdMo, or PdMo + Plts that were primed with LPS (2 ng ml$^{-1}$, 3 h) and stimulated with Nigericin (10 μM, 90 min) in the presence of VX-765 (50 μM) to prevent ASC speck release. Images are representative of 3 independent experiments, pooled in (D). Isotype controls, unstimulated conditions are shown in Appendix Fig. S1. Data is combined from independent experiments ($n = 3$). (E) Fluorometric assessment of caspase-1 activity in the CFS of primary human monocytes stimulated as in (A). Data is combined from independent experiments ($n = 3$). (F, G) IL-1β concentrations in CFS of LPS-primed StdMo, PdMo, or PdMo + Plts that were transfected with Lipofectamin (LF) containing (E) double-strand DNA (dsDNA, 0.5 μg ml$^{-1}$) for the activation of the AIM2 inflammasome. In (F), cells were treated with PrgI (100 ng ml$^{-1}$) and protective antigen (PA, 1 μg ml$^{-1}$) for 90 min, for the activation of the NLRC4 inflammasome. Floating bars display the max/min values with indication for the mean (white bands). Each symbol represents one independent experiment or blood donor. Data is combined from independent experiments ($n = 3$ in E) and ($n = 4$, in F, G). (H) Concentrations of IL-1β assessed in CFS by positively isolated mouse CD45$^+$ CD11b$^+$ LyC6$^+$ blood monocytes that were culture alone (Monocytes) supplemented with autologous platelets (Monocytes + Plts) at a 100:1 platelet:monocyte ratio. Cytokine levels secreted by platelets alone (Plts) were measured as control. Cells were stimulated with LPS (1 μg ml$^{-1}$) followed by activation with nigericin (10 μM). Graphs with floating bars depict maximum/minimum values relative to the mean (white bands). Each symbol represents pooled data from 3 mice, $n = 12$ in total. (I) Radar plots displaying the 19 significantly ($p < 0.05$, two-way ANOVA, Tukey's multiple comparison test) altered cytokines detected by Cytokine Luminex in the CFS of human monocytes and platelets stimulated as in (A). Protein concentrations are represented by the spread from inner (0 ng ml$^{-1}$) to outer circles (≥10 ng ml$^{-1}$). Colours represent stimuli (Gray: Unstim, Red: LPS, dark blue: LPS + Nig). Each symbol represents one independent experiment or blood donor ($n = 5$ for monocytes, and $n = 2$ for platelets alone). Graphs in (A, D–G) display floating bars depicting the maximum/minimum and the mean (white bands). $P$ values are from two-way ANOVA with Tukey's or Sidaks multiple comparison tests and are shown in the figure. Each symbol represents one blood donor. For better visualization, cytokine concentrations ($y$ axis) are displayed with a cut off at 8 ng ml$^{-1}$. Levels of IL-1RA and CCL4 reached 20 ng ml$^{-1}$. See also Figs. EV1, EV2.

(Fig. EV2E) and responded poorly to LPS + Nig stimulation (Fig. 1H). Akin to human monocytes, addition of freshly isolated mouse platelets to monocytes boosted their IL-1 response towards inflammasome activation (Fig. 1H). These findings conclusively demonstrate that platelets directly impact the inflammasome activation of monocytes.

In human monocyte-derived macrophages (hMDMs), platelets license NLRP3 mRNA and protein expression, specifically boosting the activation of the NLRP3 inflammasome (Rolfes et al, 2020). However, in human monocytes, we found that platelet removal additionally modulated the activity of the AIM2 and NLRC4 inflammasomes (Fig. 1F,G) which, differently than NLRP3 do not require LPS-priming. Similar findings were observed when we assessed IL-18, another member of the IL-1 family also cleaved by active caspase-1 (Appendix Fig. S1). As AIM2, NLRC4, ASC, and IL-18 expression are independent of TLR priming, these findings point to a broader influence of platelets on monocyte inflammasome function (Fig. 1F,G) and that platelets may influence additional innate immune sensors in monocytes. Supporting this conclusion, platelet removal impacted the monocyte production of inflammasome-independent cytokines IL-6 and TNFα upon exposure to LPS (Figs. 1A and EV1E), Pam3CysK4, a TLR2 ligand, and Resiquimod (R848), which activates TLR7 and TLR8 (Fig. EV1J). Platelet depletion further affected the production of numerous cytokines, chemokines, and growth factors upon TLR4 and NLRP3 inflammasome activation (Fig. 1I) or TLR1/2 and TLR7/8 activation (Fig. EV1K). While StdMo released copious amounts of IL-1β, IL-6, CCL2, CCL4, IL-8, PDGF, and IL-1RA (> 10 ng ml$^{-1}$), the overall cytokine response of PdMo was blunted. However, the PdMo's impaired cytokine response was reverted by reintroducing autologous platelets (50:1 platelet:monocytes ratio) to PdMo (Figs. 1I and EV1K). Hence, the disruption of cytokine responses of PdMo was not limited to a specific TLR or the NLRP3 inflammasome but portrayed a widespread impairment of monocyte inflammation towards pattern recognition receptors (PRRs). These findings underscore the paramount role of platelets in the PRR-induced cytokine response of human monocytes.

## Platelet supplementation reverts monocyte immunoparalysis in immune thrombocytopenia

To assess the impact of platelets on the effector functions of monocytes in a clinical setting where blood platelet numbers are inherently low, we isolated monocytes from patients with immune thrombocytopenia (ITP). ITP is an autoimmune disorder characterized by the destruction of platelets and impaired platelet production in the absence of infections or other causes of thrombocytopenia (Cooper and Ghanima, 2019). Patients in our ITP cohort had low platelet counts despite showing average blood leukocyte counts and plasma C-reactive protein concentrations (Fig. 2A–D). Importantly, ITP patients were asymptomatic, free of infections, and were not treated with glucocorticoids (Fig. 2A). We found that untouched monocytes taken from ITP patients (referred to as "ITPMo") were inherently impaired in their capacity to produce cytokines in response to LPS and LPS + Nig stimulation compared to monocytes from healthy volunteers (StdMo) (Fig. 2E–G). Remarkably, when healthy fresh platelets were supplemented to ITPMo, their cytokine production was revitalized to levels equivalent to stimulated healthy monocytes (StdMo), demonstrating that platelets can revert monocyte immunoparalysis. These findings confirm the relevance of platelets as checkpoints for monocyte-driven immune responses in a clinical setting and highlight the potential of platelet supplementation in counteracting monocyte immunoparalysis in ITP.

## Platelet depletion induces transcriptional silencing of inflammatory genes in human monocytes

To gain insights into the transcriptional changes in human monocytes upon platelet depletion/supplementation, we analyzed the expression of 770 genes comprising the myeloid innate immune response. We compared untouched StdMo, PdMo, and PdMo replenished with autologous platelets (PdMo + Plts) under resting conditions or upon ex vivo stimulation with LPS (Fig. 3).

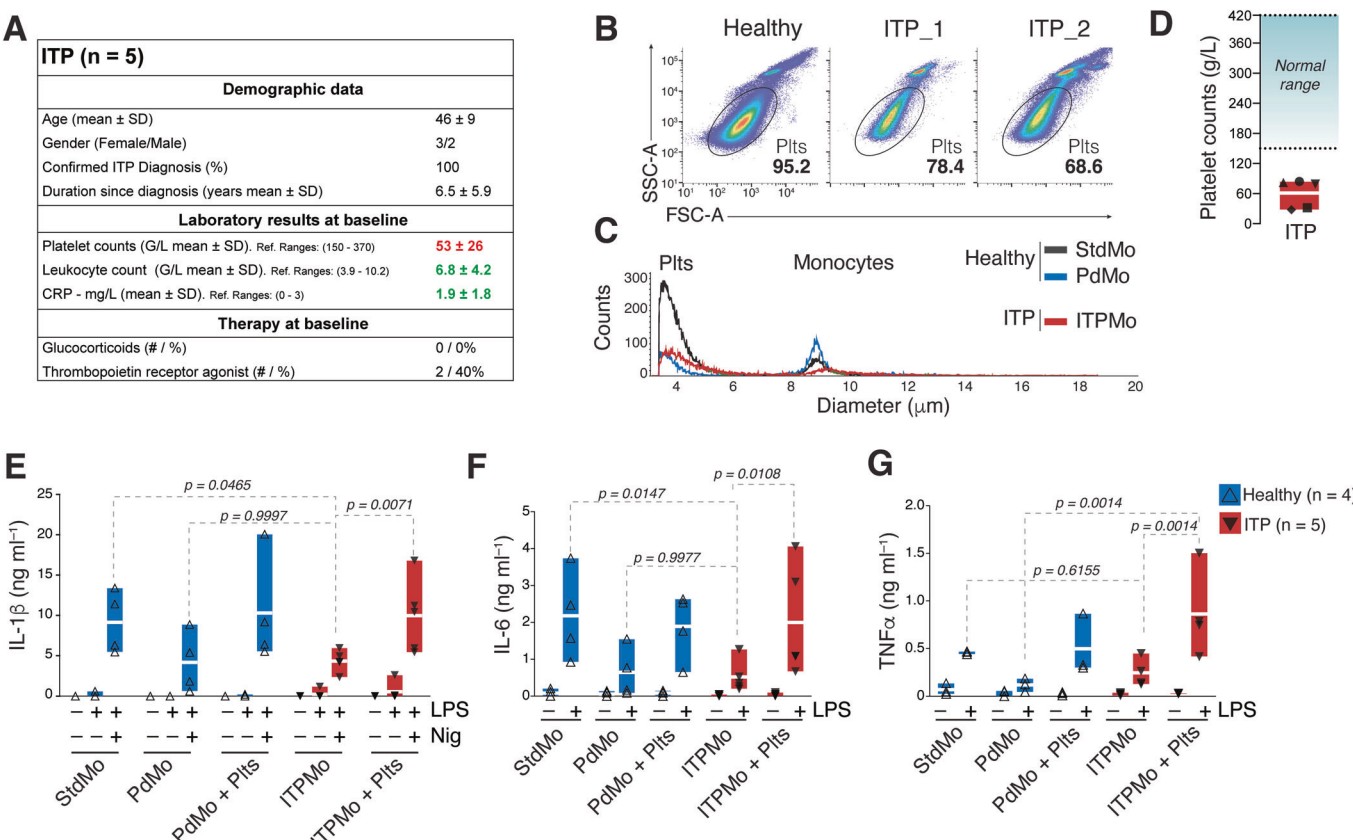

**Figure 2. Platelet supplementation reverts monocyte immunoparalysis in immune thrombocytopenia (ITP).**

(A) Demographic and clinical characteristics of ITP patients ($n = 5$). (B, C) Representative flow cytometry percentages and (C) CASY automated cell quantification of monocytes and platelets in preparations of untouched or platelet-depleted monocytes from healthy donors (±platelet-depletion) or untouched monocytes from ITP patients (ITPMo). (D) Clinical laboratory quantification of platelets in the peripheral blood of ITP patients. Blue shaded area displays the healthy reference ranges ($n = 5$). (E–G) Concentrations of IL-1β, IL-6, and TNFα released into the CFS by untouched (StdMo), and PdMo isolated from healthy volunteers ($n = 4$), or untouched monocytes from ITP patients (ITPMo, $n = 5$). Healthy platelet-depleted monocytes (PdMo + Plts), or ITPMo were supplemented with platelets (ITPMo + Plts). Cells were stimulated with LPS (2 ng ml$^{-1}$) for IL-6 and TNFα measurements, or LPS + Nig (10 μM) for IL-1β measurements. Data is displayed as floating bars with the max/min values and mean (white bands). P values were calculated with two-way ANOVA, Tukey's multiple comparison test, and are displayed in the Fig. Each symbol represents one independent experiment or blood donor.

Firstly, we examined the transcriptional effects of LPS stimulation in StdMo, PdMo, PdMo + Plts, and platelets alone (Plts). Principal component analysis (PCA) revealed that platelet transcripts formed distinct clusters separated from monocytes, indicating a negligible contribution of platelet transcripts to the overall mRNA pool. The transcriptional response of stimulated monocytes was clearly distinguished from that of unstimulated StdMo (Fig. 3A), consistent with a typical LPS-induced gene expression (with 100 upregulated and 54 down-regulated genes) compared to unstimulated conditions (Fig. 3B, left panel). Strikingly, stimulated PdMo showed a less distinct transcriptional profile, clustering closer to unstimulated StdMo (Fig. 3A), indicating a loss of transcriptional response to stimulation, and consistent with diminished expression of LPS-induced genes (Fig. 3D). Indeed, only four genes were significantly induced by LPS in PdMo (Fig. 3B, middle panel and Fig. 3C), indicating a general suppression of inflammatory gene expression. Notably, the reintroduction of autologous platelets to PdMo (50:1 Platelet:Monocyte ratio) re-approximated their PCA-clustering towards StdMo (Fig. 3A) and

restored the expression of 86% of LPS-induced genes in PdMo, resembling the profile of StdMo (Fig. 3B, right panel, and Fig. 3C). These results were consistent with the multiplex cytokine analysis (Figs. 1I and EV1K).

Next, to specifically address the effects of platelet depletion/replenishment on monocytes, we separately compared StdMo vs. PdMo in unstimulated and stimulated conditions (Fig. 3E,F). Platelet depletion alone modified 87 genes in steady-state monocytes, notably suppressing genes like MAPK14 (p38α) and BTK involved in pro-inflammatory signaling. Meanwhile, numerous transcription factors (TFs) related to monocyte differentiation and other processes were induced (Appendix Fig. S2). We identified several monocyte cytokine function regulators among the most highly differentially expressed genes (DEGs). For example, genes involved in sensing chemokines or external stimuli (e.g., CCR2, FCGR3A, and CD14) were downregulated in PdMo. In contrast, several TFs (e.g., EGR2, PPARG) and ERK-MAPK repressor genes (e.g., ATF3 and SPRY2) were upregulated in PdMo (Fig. 3E,F; Appendix Fig. S2). These findings demonstrate

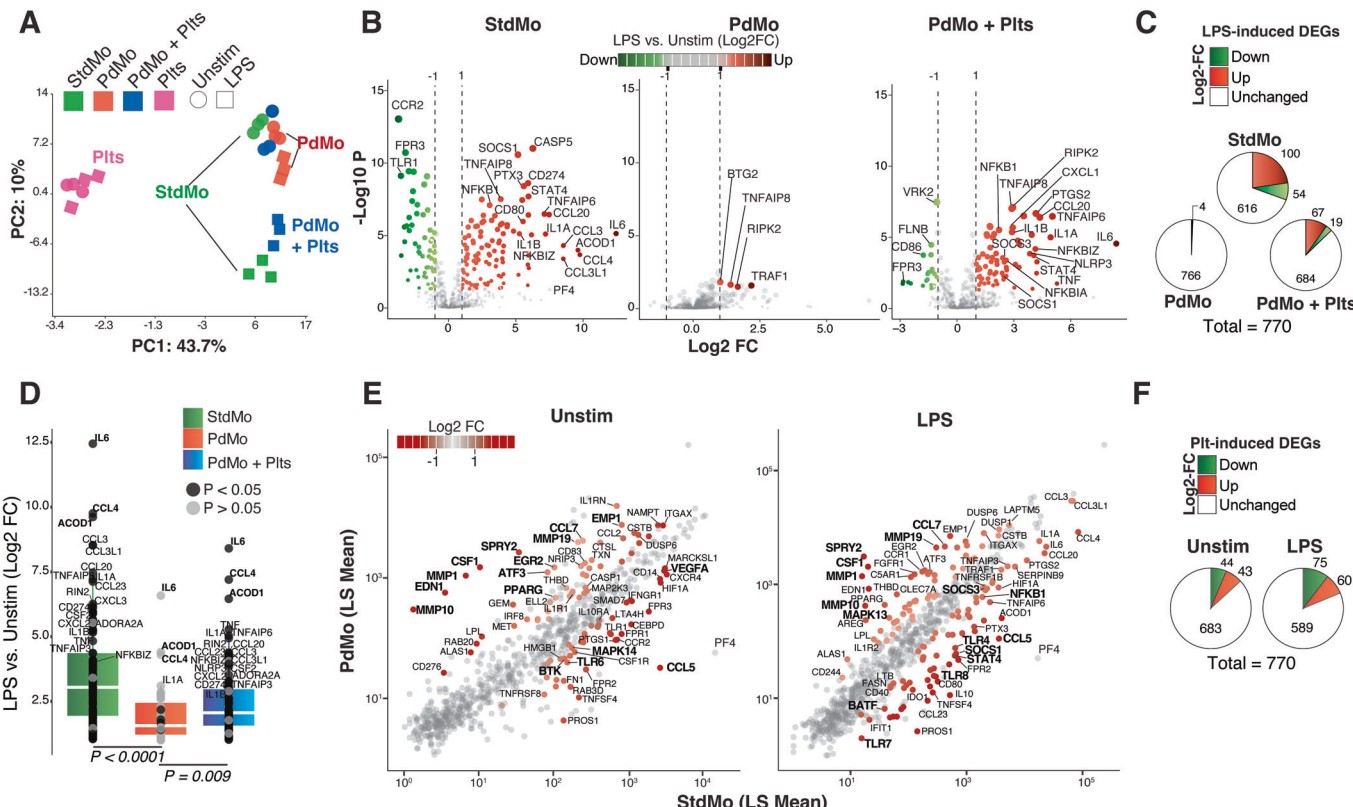

**Figure 3. Platelet depletion causes transcriptional shutdown of myeloid innate immune genes in human monocytes.**

(**A**) Principal Component Analyses (PCA) of gene expression in StdMo (green), platelet-depleted monocytes (PdMo, red), platelet-depleted monocytes reconstituted with platelets (PdMo + Plts, blue), and platelets alone (Plts, pink). Each symbol represents one donor. Stimulations are represented with shapes (ellipse = unstimulated, square = LPS). (**B**) Volcano plots showing log2 fold change (x-axis) and significance (−log10 *p-value; y-axis) of genes differentially expressed comparing LPS-stimulated vs. unstimulated (Unstim) StdMo, PdMo, or PdMo + Plts. P values are from ANOVA multiple comparisons with FDR correction calculated in Partek® Genomics Suite®. (**C**) Pie charts indicate the number of up- and downregulated DEGs in each group upon LPS stimulation. (**D**) Quantitative expression (Log2 Fold Change) of the LPS-induced (>= 2-Fold Change) genes in StdMo (green), PdMo (red), or PdMo + Plts (blue) (n = 724). P values are from Krustal–Wallis test calculated with the ggplot2 R package. (**E**) Scatter plots displaying the LSMean (x-axis) of StdMo vs PdMo (y-axis) from DEGs comparing the effects of platelet removal in Unstim and LPS-stimulated conditions. DEGs with 2-Fold change are highlighted in red. (**F**) Pie charts representing the DEGs of the comparisons in (**D**), representing DEGs induced by platelet depletion in unstimulated or LPS-stimulated monocytes. See also Appendix Fig. S2.

that transcriptional reprogramming underlies the functional effects of platelet depletion in primary human monocytes. Furthermore, in line with the cytokine secretion (Figs. 1I and EV1K), the transcriptional shutdown of pro-inflammatory gene expression in PdMos can be reversed by their replenishment with fresh platelets.

## Platelet-depletion in vivo alters monocyte's transcriptional host defense programs

Our findings show that monocytes from thrombocytopenic individuals have impaired cytokine responses to TLR and NLR activation. To recapitulate the ITP scenario in vivo, we induced thrombocytopenia in mice via i.v. injection of 2 mg kg$^{-1}$ of anti-GPIbα mAb or control IgG (Sreeramkumar et al, 2014; Xiang et al, 2013) (Fig. 4A). We then challenged mice with i.v. LPS (2 mg kg$^{-1}$) or PBS. Anti-GPIbα effectively reduced platelet counts in treated mice (Fig. 4B). Platelet depletion did not significantly affect systemic IL-1β, TNFα, and IL-6 levels in mice (Appendix Fig. S3A), consistent with previous findings in LPS challenge and

bacterial infection (Carestia et al, 2019; Claushuis et al, 2016; de Stoppelaar et al, 2014; Xiang et al, 2013).

Given IL-1β impairment in isolated mouse monocytes restored by platelet supplementation (Fig. 1H), and to identify the monocyte contributions to the systemic cytokine levels, we analyzed platelet depletion effects on blood monocytes isolated by FACS-sorting of Ly6G$^-$ IA/IE$^-$ CD45$^+$ CD11b$^+$ CD115$^+$ mouse PBMCs. Platelet transcripts (e.g., *Pf4, Gp9, Itga2b, Ppbp, Tubb1, Treml1, Clu*) were downregulated in monocytes from platelet-depleted mice (PBS group) (Appendix Fig. S3B). However, genes involved in the complement cascade (e.g., *C1qa, C1qb, C1qc*) and cell activation (e.g., *Htra3, Mertk*) were upregulated, with enrichment in pathways related to coagulation, hemostasis, platelet activation, complement, and wound healing (Appendix Fig. S3C). These findings indicate that anti-GPIbα treatment resulted in platelet activation.

We compared the effects of platelets on the monocyte transcriptional response to LPS challenge in vivo. Expression profiling (Fig. 4C) and gene ontology (GO) (Fig. 4E,F) showed that monocytes from IgG-treated mice had a typical LPS-induced transcription profile (Alasoo et al, 2015) with upregulation of

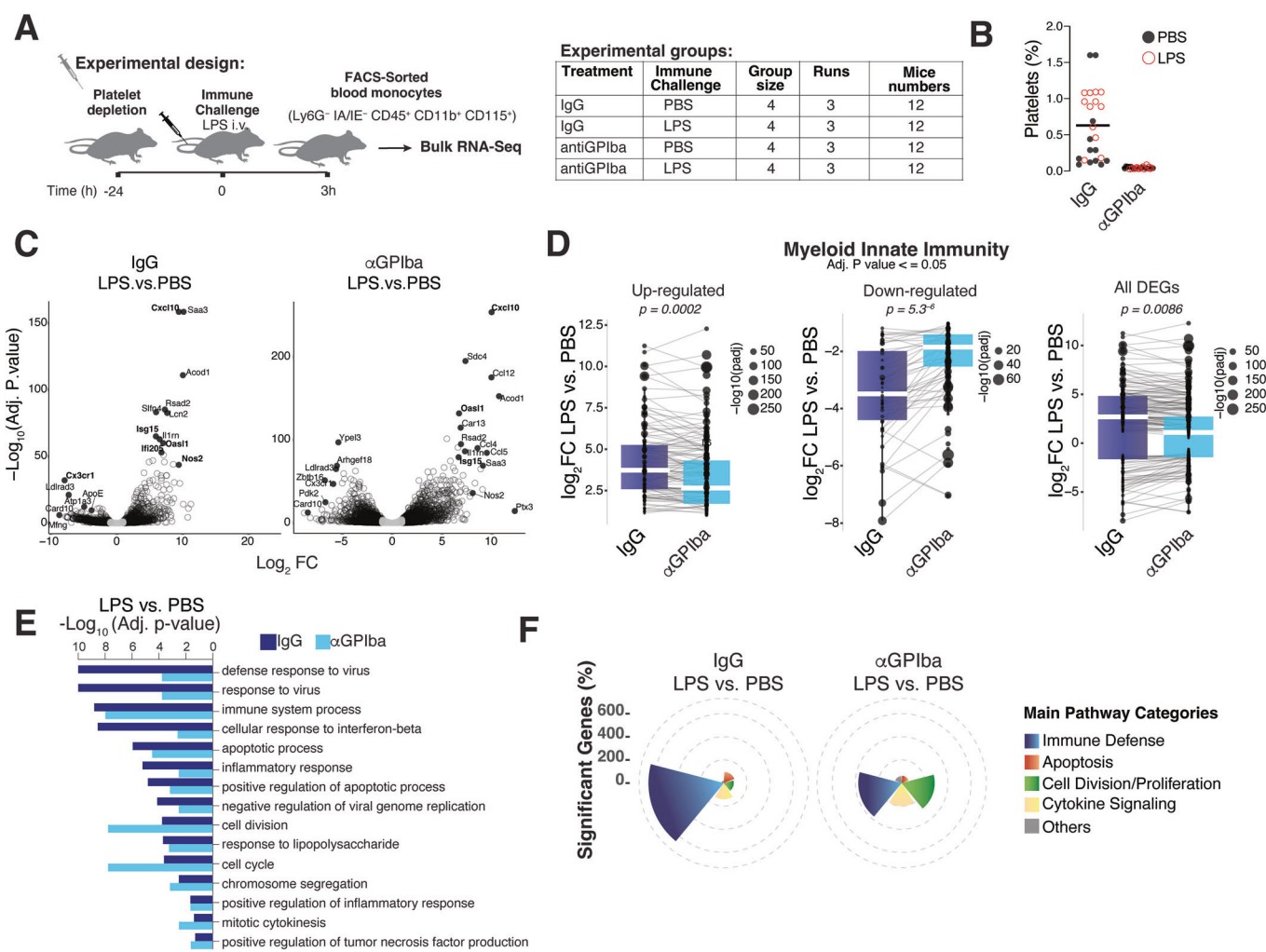

**Figure 4. Thrombocytopenia in vivo alters monocyte's host defense transcriptional program.**

(A) Schematic representation of the experimental setting for platelet depletion in vivo followed by intravenous challenge with LPS, FACS-sorting and bulk-RNASeq of blood monocytes. Monocytes were isolated from PBMCs from pooled blood of 4 mice per group (IgG- or anti-GPIbα-treated) each challenged either with LPS or PBS in 3 different experiments. (B) Flow cytometric quantification of platelets in IgG-treated and anti-GPIba treated mice ($n = 24$ per condition). (C) Volcano plots (log2 fold change vs. $-\log_{10}$ adjusted $P$-value) of the significant DEGs (adjusted $P$-value < 0.05, Fold Change ≤ −2 or ≥ 2) comparing PBS vs. LPS-challenge in IgG (left) or anti-GPIbα-treated mice (right). Wald test was used to generate $p$-values and log2 fold changes (details in RNASeq_report, submitted to BioStudies E-MTAB-14126). (D) Fold change of the DEGs that compose the myeloid innate immunity that were modulated by LPS in monocytes from IgG vs. anti-GPIbα-treated mice. A total of 556 mouse genes with matching hortologs in the human myeloid panel and were significantly changed are shown. $P$ values are from Wilcoxon rank-sum test, calculated by the ggplot2 R package. (E) Pathway enrichment analysis of the LPS-induced DEGs comparing monocytes from IgG (dark blue) or anti-GPIba-treated mice (light blue). A gene ontology analysis was performed on the statistically significant set of genes by implementing the software GeneSCF v.1.1-p2. The mgi GO list was used to cluster the set of genes based on their biological processes and determine their statistical significance. (F) Main represented categories of pathways (GO analysis) from the DEGs induced by LPS on monocytes from IgG-treated vs. platelet-depleted mice (See also Appendix Fig. S4).

immune defense genes (e.g., *Cxcl10, Ccl12, Il1a, Ly6c, ISGs, Il12a*) and downregulation of *Card10* and *Cx3cr1* (Pachot et al, 2008) (Fig. 4C). LPS altered more transcripts in monocytes from platelet-depleted (anti-GPIbα-treated) than IgG-treated mice (4088 vs. 1282 transcripts, Appendix Fig. S3D). However, 80% of the LPS-induced genes were shared between the groups. Monocytes from platelet-depleted mice had 3008 unique DEGs, while those from IgG-treated mice had 202 (Appendix Fig. S3D).

Focusing on 770 myeloid innate immunity genes (Fig. 3), monocytes from thrombopenic mice showed reduced expression of myeloid immune response genes after LPS challenge (Fig. 4D). IgG-treated mice had higher enrichment scores for pathways in host

immune defense (e.g., immune response to virus, IFNβ response, inflammation) (Fig. 4E,F). These pathways were present but less enriched in platelet-depleted mice, whose gene profile skewed towards apoptosis and cell cycle regulation (Fig. 4E,F). Thus, despite causing platelet activation, which may influence the systemic inflammatory status, platelet depletion reprograms monocyte transcription towards cell cycle and division, compromising immune defense. These findings confirm previous observations of compromised immunity (Carestia et al, 2019; Claushuis et al, 2016; van den Boogaard et al, 2015; Xiang et al, 2013), and higher susceptibility to LPS- or bacteremia-induced sepsis in thrombopenic mice, despite normal systemic cytokine levels.

## Platelets regulate cytokine secretion of monocytes in trans, and independently of classical platelet-monocyte crosstalk mechanisms

We next investigated various mechanisms underlying the cellular cross-talk between platelets and monocytes in regulating monocyte cytokine secretion (Appendix Figs. S4, S5). Despite evidence that innate immune cells can engulf platelets (Lang et al, 2002; Maugeri et al, 2009; Rolfes et al, 2020; Senzel and Chang, 2013), we did not observe distinguishable platelet internalization by human monocytes. However, blocking actin polymerization in platelet-depleted monocytes abolished the platelets' ability to restore faulty cytokine secretion, indicating the significance of monocyte's phagocytic machinery in this process. We also found that cell-free supernatants (or releasates) of platelets efficiently restored cytokine production in monocytes, supporting that the platelet effect does not depend on cellular contacts. Furthermore, using a series of experiments with recombinant proteins and blocking antibodies we ruled out the involvement of numerous classical mechanisms of platelet-monocyte crosstalk from the effects we report in this study. That included: numerous ligand-receptor interactions and integrins, and platelet released molecules, such as RANTES, CXCL12, sialic acids, and signaling receptors such as Siglec-7, and CD40-CD40L. Additionally, supplementation of platelet-depleted monocytes with recombinant cytokines or activation of the CD40-CD40L axis did not mimic platelet-mediated rescue of cytokine secretion. Combined, these experiments (Appendix Figs. S4, S5) demonstrated that the platelet effect on monocytes is contact-independent, caused by factors present in the supernatants of platelets, rather than classical platelet-monocyte crosstalk mechanisms.

## Platelets are cellular sources of pro-inflammatory transcription factors (TFs)

Our transcriptomic approach revealed that platelets influence the expression of genes controlled by key transcription regulators known to orchestrate pro-inflammatory cytokine production in monocytes, such as NF-κB, BTK, and MAPKs. We, therefore, profiled the activation of serine/threonine (STK) and protein tyrosine (PTK) kinase networks on StdMo, PdMo, or PdMo + Plts as well as on platelets alone (Plts) and observed that platelet depletion markedly impacted the kinome network of LPS-stimulated monocytes (Figs. 5A–D and EV3A). With few exceptions, the LPS-induced phosphorylation of numerous PTK and STK subtracts was significantly lower in PdMos compared to StdMo (Fig. 5C). Substantiating the results of mRNA expression (Fig. 3), platelet depletion decreased the activity of BTK, p38 MAPK, MAPK-activated protein kinases (MAPKAPK), IKK kinases, erythropoietin-producing human hepatocellular receptors (Eph), Cyclin-depended kinases (CDK), protein kinase A, (PKA) and protein kinase C (PKC) (Fig. EV3A). In contrast, platelet-depleted monocytes displayed increased activation of Janus kinases (JAKs) and IKKs, which can regulate NF-κB signaling (Solt and May, 2008) (Fig. 5A). In line with the altered function of these groups of kinases, pathway analysis revealed that platelet depletion predominantly changed the activity of kinases involved in signal transduction associated with NF-κB activation, immune responses via CD40 signaling, MAPK, and JAK/STAT signaling pathways (Fig. 5D). Combined, the transcriptome (Fig. 3) and kinome

(Fig. 5) of human monocytes subjected to platelet depletion/re-addition revealed dysregulation of key cytokine-regulatory pathways in PdMo.

Confirming the dynamic changes in NF-κB activation caused by platelet depletion/reconstitution in human monocytes, the LPS-induced phosphorylation of the NF-κB subunits p55 (RelB) and p65 (RelA) was impaired in PdMo, as measured by HTRF (Fig. EV3B) and immunoblotting (Fig. 5E), and it was restored by the re-addition of platelets. Supporting that platelets regulate the NF-κB activity on monocytes, increasing concentrations of platelets enhanced TLR2/1-triggered NF-κB activity in monocytic THP-1 Dual reporter cells encoding an NF-κB-inducible promoter (Fig. 5F). Together, these findings highlight the involvement of the NF-κB and MAPK signaling pathways as candidate mechanisms by which platelets regulate the cytokine production of human monocytes.

Interestingly, throughout our kinase and immunoblotting assays, platelets consistently showed high basal kinase activity (Fig. 5B,C), and high protein levels of NF-κB subunits (RelA, p65) (Figs. 5E and EV3C), and p38α MAPK (Fig. EV3D), supporting prior findings that platelets are cellular sources of these signaling molecules (Lannan et al, 2015). Indeed, by assessing the native and phosphorylated form of these proteins in the lysates or supernatants of human platelets, we demonstrated that platelets express and release NF-κB and confirmed previous observations that they release p38α (Fig. EV3D), the major isoform expressed on human and mouse platelets (Shi et al, 2017). Although phosphorylated p38 (P-p38) was predominantly enriched in platelets, this protein was additionally detected in platelet supernatants (Fig. EV3D). These findings support that human platelets are abundant sources of pro-inflammatory transcriptional regulators.

## Platelet-derived TFs are enriched in human monocytes

Given the abundance of these critical transcriptional regulators in platelets (Beaulieu and Freedman, 2009; Ezumi et al, 1995; Lannan et al, 2015; Poli et al, 2022) and the corresponding dynamic activation of these pathways on platelet-depleted/replenished monocytes (Figs. 3, 5), we speculated that platelets could transfer TFs or signaling molecules to monocytes. Cell-to-cell propagation of TF signaling has been reported to occur through gap junctions between connecting cells, with biological functions demonstrated in recipient cells (Ablasser et al, 2013; Kasper et al, 2010). However, little is known about the trans-cellular propagation of signaling molecules in blood circulating cells.

To precisely delineate the repertoire of platelet proteins transferred to monocytes, we employed an unbiased proteomic approach combined with Stable Isotope Labeling by Amino acids in Cell culture (SILAC) (Ong et al, 2002). For this, we cultured a human megakaryocytic cell line (MEG-01) for several days in a medium containing stable isotope labeled amino acids [$C^{13}L^{15}$] L-lysine and [$C^{13}L^{15}$] L-Arginine (hereafter referred to as heavy AAs). Under these conditions, newly translated proteins in MEG-01 incorporate heavy AAs and can be distinguished from pre-existing proteins on monocytes by a mass shift (Wolf et al, 2020). Of note, cell-free supernatants from MEG-01 (termed MK-Sups), akin to platelets, were equally capable of restoring the impaired cytokine responses observed in PdMos (Fig. 6A). Subsequently, PdMos were incubated

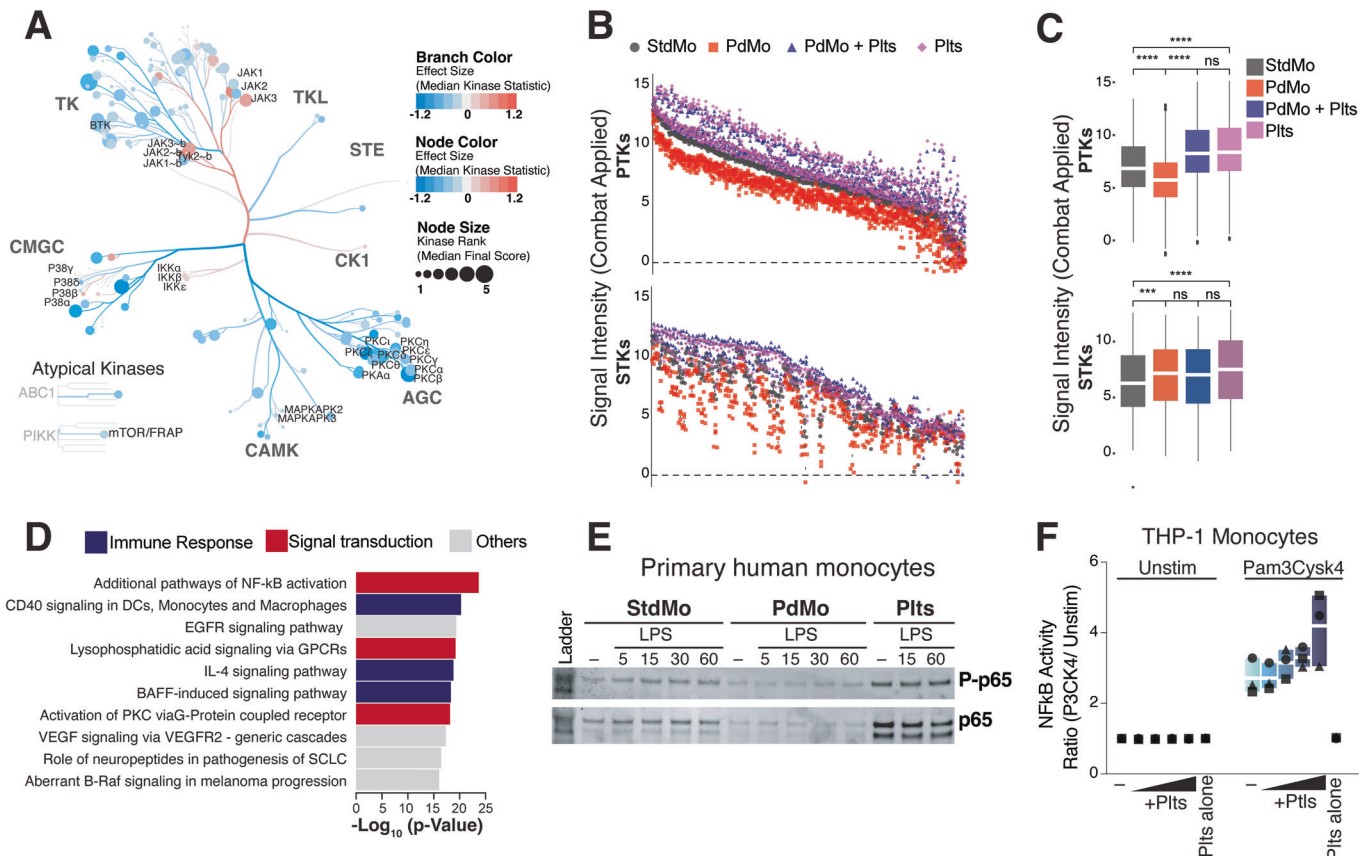

**Figure 5. Platelets are cellular sources of active kinases and transcription factors.**

(A) Coral trees display the activity of Protein Tyrosine (upper branches) and Serine/Threonine kinases (lower branches) in StdMo, PdMo, PdMo + Plts and platelets from 5 donors, upon LPS stimulation for 15 min. (A) Impact of platelet depletion on the activity of PTK and STK in unstimulated monocytes. (B, C) Scatter plots (B) and box plot (C) showing the signal intensity for the phosphorylation status of phosphosites (x-axis) that are substrates of PTK (top) and STK (bottom) comparing StdMo, PdMo, PdMo + Plts, and platelets alone (Plts). Each symbol shows one donor (*n* = 5) and only include peptides which passed the QC. batch correction with a 2-step Combat correction. *P* values in (C) were calculated by ANOVA multiple comparison and are indicated as **** (<2.22e−16) or ns (*p* > 0.05). (D) Pathway analysis of the most represented signaling pathways dynamically changed upon platelet depletion/re-addition to primary human monocytes. See also Fig. EV3. *P* values were calculated by PamGene and based on Paired T-test (PTT) per Group paired on Run and N. (E) Immunoblot of total RelA (p65), and Phosphorylated-p65 (P-p65) in MoStd, MoPD and Plts that were left untreated (−) or stimulated with LPS (2 ng ml⁻¹) for 5, 15, 30 or 60 min. Representative of three independent experiments. (F) NF-κB activity assay in CFS of unstimulated (Unstim), or Pam3CysK4-stimulated (100 ng ml⁻¹) NF-κB-SEAP and IRF-Lucia luciferase Reporter Monocytes cultured alone (−) or co-cultured with increasing concentrations of human platelets (1:10, 1:20, 1:50, and 1:100). SEAP-Activity was measured with QuantiBlue Buffer. Graphs show floating bars display max/min values with indication to the mean (white bands). Each symbol represents one independent experiment.

with MK-Sups, followed by a stringent washing protocol to eliminate non-specific associations before subjecting the monocyte lysates to MS analysis (Fig. 6B, C). Notably, we detected 33 proteins harboring heavy AAs enrichment in PdMos exposed to MK-Sups under LPS-stimulated conditions (Fig. 6D) and numerous TFs in unstimulated monocytes (Appendix Fig. S6A). Notable entities figured among these proteins, including the transcriptional activator NF-κB2 (p100/p52), transcriptional regulators (e.g., ANP32B, THOC1), enzymes involved in mRNA translation and stability (ZFP36, EOF2A, and PAIP1), and proteins implicated in the MAPK/ERK signaling cascade (Appendix Fig. S6). Furthermore, 5 out of the 33 heavily labeled proteins enriched in PdMo after LPS were involved in trans-membrane functions, pointing towards a mechanism for the internalization of MK-derived cargo. We confirmed the expression of NF-κB2 on human platelets by WES capillary electrophoresis (Fig. 6E), raising the possibility that platelets donate NF-κB2 to human monocytes.

## Platelets overcome pharmacological and genetic ablation of NF-κB and MAPK signaling in human monocytes

Next, we conducted a series of pharmacological and genetic strategies to target pro-inflammatory pathways in platelets or monocytes and examine their effects in the cytokine outputs. Firstly, we tested whether pharmacological inhibition of NF-κB and p38 MAPKs on platelets interfere with their capacity to reactivate cytokine production in PdMos (Fig. 7A). Pre-treatment of platelets with the irreversible inhibitor of IκB kinase (IKK) phosphorylation BAY 11-7082 (BAY), or the p38 inhibitor SB203580 (Yamashita et al, 2009) extinguished their ability to recover IL-1β secretion from inflammasome-activated PdMo (Fig. 7B). Likewise, intact platelets, but not NF-κB-, or p38-inhibited platelets, restored IL-6 and TNFα secretion of LPS-stimulated PdMo. To account for inhibitor carryover, we directly exposed monocytes to the highest concentrations of BAY or

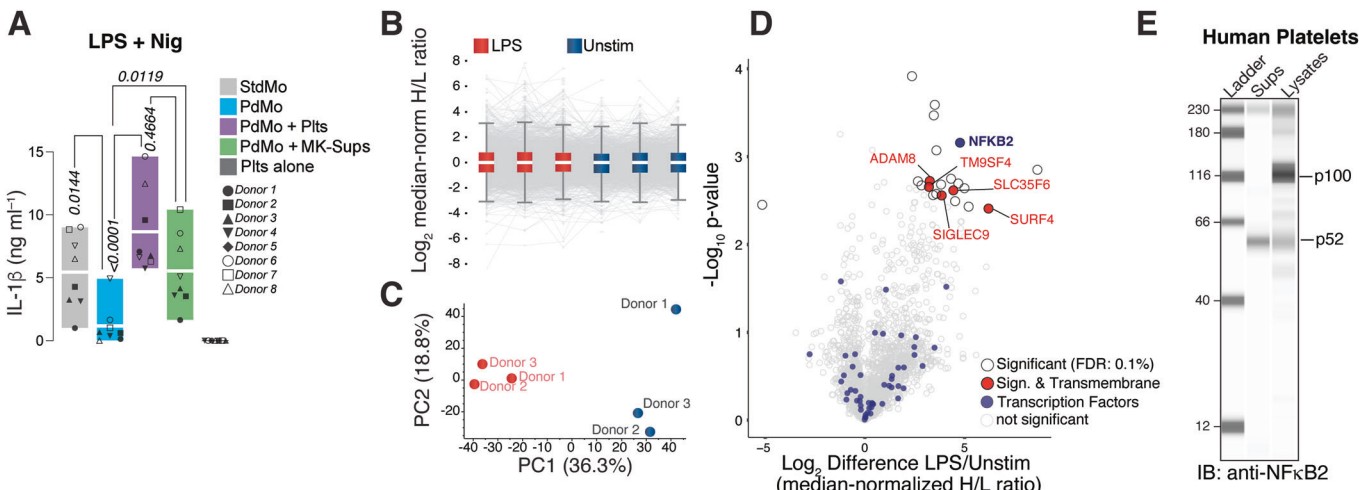

**Figure 6. Enrichment of megakaryocyte-derived NF-κB2 in human monocytes.**

(A) IL-1β released from LPS-primed StdMo, and PdMo that were reconstituted with platelets (PdMo + Plts) or culture supernatants from MEG-01 cells that were cultured in SILAC medium (MK-Sups). Floating bars display the max/min values with indication for the mean (white bands). P values were calculated using one-way ANOVA, Tukey's multiple comparison test and are indicated in the figure. Each symbol represents one independent blood donor (n = 8). (B, C) Analysis of Mass speck proteomics combined with Stable isotope labeling with amino acids in cell culture (SILAC-proteomics) in human PdMo exposed to cell-free supernatants of MEG-01 cells saturated with heavy AAs from 3 donors. (B) Boxplots of quantitative distributions showing the Log2 transformed protein ratios of heavy to light (H/L) labeled proteins calculated with a sample-wise median substitution for normalization. (C) Principal component analysis (PCA) of SILAC proteomics samples from PdMo reconstituted with cell-free supernatants of MEG-01 cells. (D) Volcano plots of proteins with heavy-AAs detected in PdMo exposed to cell-free supernatants of MEG-01 cells through SILAC-proteomics. Cells were stimulated with LPS. Unstimulated conditions are shown in Appendix Fig. S6 1D annotation enrichment (Benjamini-Hochberg FDR = 0.1) revealed that the Uniprot keyword "transmembrane" was enriched in LPS-stimulated samples. Paired Welch's T-test with permutation-based FDR correction (FDR = 0.1, S0 = 0.1), n = 3. (E) WES capillary electrophoresis and immunoblotting of NF-κB2 on whole cell lysates (Lysates) or cell-free supernatants (Sups) from human platelets. Representative of 2 independent experiments.

SB203580 used on platelets before washing (Fig. 7B, last two right bars). BAY has also been described to inhibit NLRP3 (Juliana et al, 2010), and, as expected, it completely abrogated monocyte cytokine responses to TLR and NLRP3 activation, whereas direct incubation of monocytes with SB203580 partially inhibited their response. BAY also blocked platelet activation, assessed by measuring the thrombin-induced P-selectin expression, and the binding of PAC-1, an antibody that binds to the GPIIb/IIIa complex on the surface of activated platelets (Appendix Fig. 7A,B). These findings support that p38 and NF-κB or their downstream signaling in platelets are essential for the regulation of cytokine responses in recipient monocytes.

Next, to address whether platelet-derived TFs function in recipient cells, we tested the capacity of platelets to bypass NF-κB or p38 inhibition in primary human monocytes. To demonstrate this, we pre-incubated untouched monocytes (StdMo) with high doses of BAY (50 μM) and SB203580 (20 μM). After washing the inhibitors away, we supplemented BAY- or SB203580-treated monocytes with growing concentrations of freshly isolated autologous platelets (Fig. 7C). In line with previous observations of toxicity associated with BAY in primary human cells (Rauert-Wunderlich et al, 2013; White and Burchill, 2008), BAY-treated monocytes displayed decreased viability (Figs. 7D and EV4C), precluding us from addressing the importance of platelet-derived NF-κB using this inhibitor. Consistent with dysfunctional p38 signaling (Lee et al, 1994), SB203580-treated monocytes were incapable of secreting cytokines in response to LPS or LPS + Nig (Fig. 7E). Strikingly, the addition of intact platelets dose-dependently bypassed p38 inhibition reactivating cytokine

secretion in SB203580-treated monocytes (Fig. 7E). Hence, platelet supplementation dose-dependently restored dysfunctional p38 activity in p38-inhibited monocytes, indicating that the p38 signaling originated from platelets. Supporting this conclusion, dual blockage of p38 on platelets and on monocytes curbed the platelet's ability to restore cytokine responses in SB203580-treated monocytes (Fig. EV4D).

Next, we aimed to create a simulation where platelets were the only cellular source of functional TFs, and specifically address the function of platelet-derived NF-κB on human monocytes. To achieve this, we used CRISPR-Cas9 gene editing to eliminate the key components of the canonical and non-canonical NF-κB signaling pathway in THP-1 monocytes, respectively RelA (p65) and the small subunit NF-κB2 (p100/52) identified in our SILAC-Mass spec approaches (Fig. 7F). These genetic modifications were confirmed by immunoblotting (Fig. 7G). As a result, monocytes lacking RelA (RelA$^{-/-}$) or NF-κB2 (NF-κB2$^{-/-}$) could not produce IL-1β in response to stimulation with Pam3Cysk4 + Nig (Fig. 7H). However, when platelets were added to these genetically altered monocytes, IL-1β production increased in a dose-dependent manner, indicating a reactivation of the NF-κB signaling pathways. Consistent with our previous findings (Rolfes et al, 2020), the addition of platelets also enhanced IL-1β secretion in genetically unmodified, wild-type THP-1 monocytes (Fig. 7H). Notably, platelets were able to reconstitute functions in cells deficient in both canonical and non-canonical NF-κB signaling, pointing towards additional or redundant mechanisms. These findings indicate that platelet-derived NF-κB signaling can compensate for genetic deficiencies in the NF-κB pathway within monocytes and

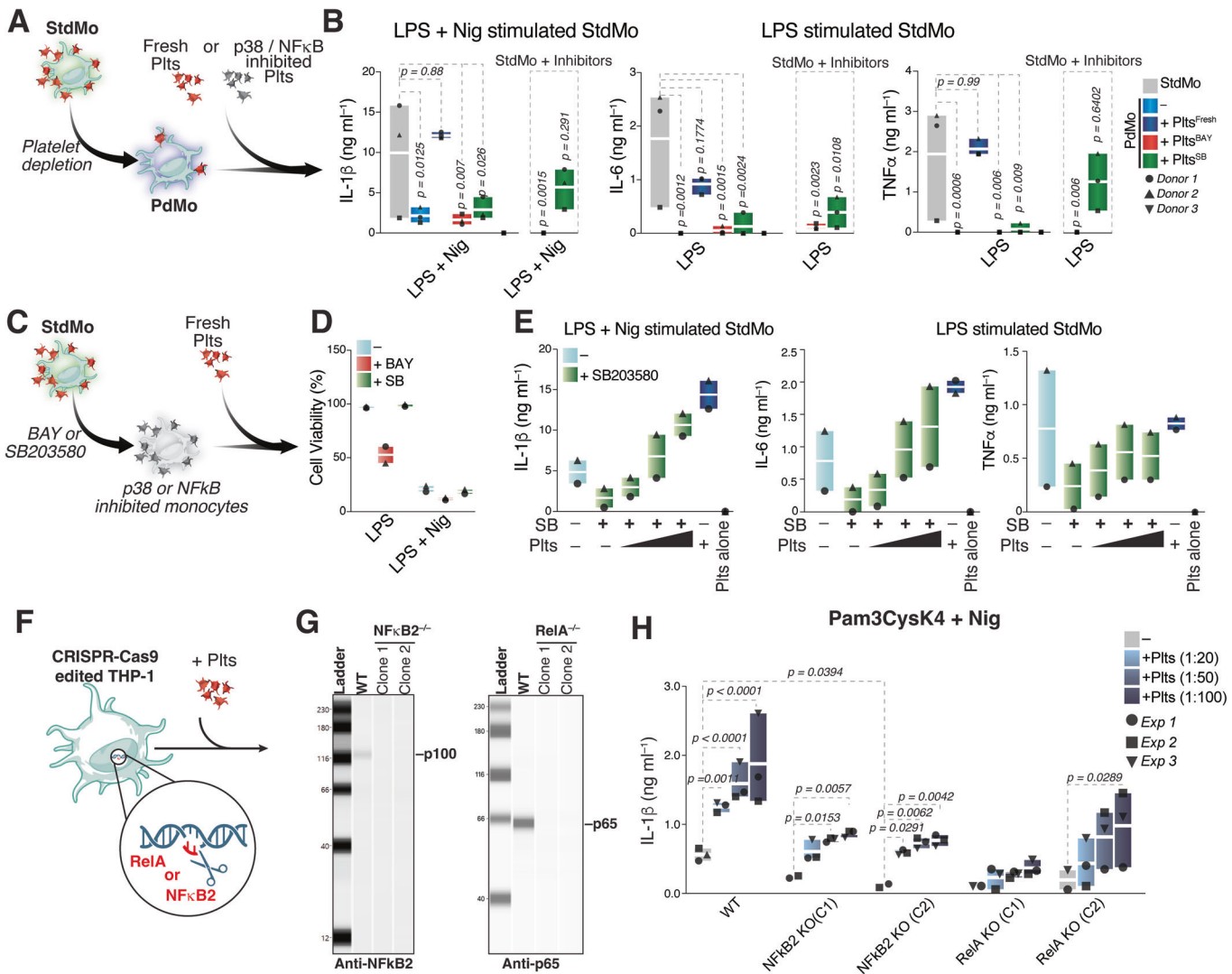

**Figure 7. Platelets restore cytokine function in p38 MAPK and NF-κB2-deficient monocytes.**

(**A**) Schematic representation of the supplementation of platelet-depleted (PdMo) primary human monocytes with autologous platelets that were left untreated (+ Fresh Plts), or pre-incubated for 20 min with 50 μM of BAY 11-7082 to inhibit NF-κB (+ Plts^BAY) or 20 μM of SB203580 to inhibit p38 MAPKs (+ Plts^SB) and washed before being added to PdMo. (**B**) IL-1β, TNFα, and IL-6 concentrations in CFS of StdMo, PdMo, or PdMo that were supplemented with platelets pre-treated with BAY11-7082 (+ Plts^BAY), SB203580 (+ Plts^SB), or left untreated (+ Plts^Fresh). Co-cultures were stimulated with LPS (2 ng ml⁻¹) for 3 h followed by 1.5 h nigericin stimulation (10 μM). Side bars show the effects of direct exposure of monocytes to the inhibitors (StdMo + Inhibitors) (n = 3). (**C**) Schematic representation of the inhibition of NF-κB or MAPK p38 on primary human monocytes (StdMo) followed by their replenishment with fresh autologous platelets. (**D**) Cell viability assay in human monocytes treated with BAY 11-7082 (50 μM, 20 min) or SB203580 (20 μM, 20 min) followed by stimulation with LPS or LPS + Nig, as depicted in (**C**) (n = 3). (**E**) IL-1β, TNFα, and IL-6 concentrations in the CFS of StdMo that were treated with SB203580 before been added with increasing ratios of freshly isolated platelets (1:5, 1:50, and 1:100). Cells were stimulated with LPS or LPS and nigericin (LPS + Nig). Floating bars display max/min values with indication to the mean (white bands). Each symbol represents one donor. See also Appendix Fig. S7. 8. (**F, G**) (**F**) Schematic representation of the genetic ablation of p65 (RelA) or p52/100 (NF-κB2) on human THP-1 monocytes, (**G**) and knockout confirmation at the protein level by immunoblotting for each protein. Two clones are shown. Data is representative of 3 independent experiments. (**H**) IL-1β concentrations in the CFS of stimulated RelA⁻/⁻ or NF-κB2⁻/⁻ THP-1 monocytes that were cultured alone or added with increasing ratios of freshly isolated platelets (1:20, 1:50, and 1:100). Cells were stimulated with Pam3CysK4 (1 μg ml⁻¹) or Pam3CysK4 and nigericin (10 μM) (Pam3 + Nig). Floating bars display max/min values with indication to the mean (white bands). P values were calculated using two-way ANOVA, Tukey's multiple comparison test compared to the control group (− Plts) and are indicated in the figure. Each symbol represents one donor. Data is pooled from 3 independent experiments.

highlight the critical role of platelets in modulating immune responses through intercellular signaling.

Finally, we tested whether inhibition of NF-κB on platelets extinguish their capacity to recover NF-κB signaling in monocytes where NF-κB activity was ablated or made deficient. For this, we pre-treated freshly isolated platelets with BAY 11-7082 before using

them to reconstitute signaling in gene-edited THP-1 clones lacking NF-κB2 and NF-κB p65. However, BAY-treated platelets retain their ability to restore IL-1β production in (Fig. EV4G), indicating that additional players, i.e., other NF-κB subunits, MAPKs, or others (Appendix Fig. S6) may be involved, but also highlighting differences between THP-1s, which are able to secrete IL-1β

without platelets, and primary human monocytes, where platelets play a crucial role.

## Platelet vesicles facilitate TFs transfer to human monocytes

Our findings revealed the presence of numerous TFs in the releasates of platelets (Plt Sups) (Figs. 5D, 6E and EV3C,D) and MKs (Fig. 6A). Notably, we found that a significant fraction of active TFs (i.e., P-p38 and the p52 subunit of NF-κB2) were enriched in the releasates of platelets, suggesting that platelets release the active form of those molecules (Figs. 6E and EV3C,D).

To gain further insights into the mechanisms allowing the transfer of platelet TFs into monocytes, we fractionated platelet releasates (Plt Sups) to enrich small vesicles. We also generated vesicle-free platelet releasates (VF-Plt Sups) through size exclusion purification (Appendix Fig. S8). Using NanoSight NS300 analysis, BCA protein assays, and WES capillary electrophoresis, we confirmed the absence of vesicles in these VF-Plt Sups using different techniques (Appendix Fig. S8). Notably, although NF-κB2 was present in platelet lysatesits active subunit p52 was predominantly detected in the vesicle-enriched fractions, and was absent in vesicle-free platelet releasates. In line with our previous findings (Fig. EV3C) the p65 NF-κB subunit RelA was predominantly present in platelets (Fig. 8A), and absent in Plt Sups or in the vesicle-enriched fractions. Furthermore, vesicle-enriched fractions were as efficient as unfractionated releasates (Plt Sups) to reconstitute PdMos cytokine responses (Fig. 8B; Appendix Fig. S8), whereas depletion of vesicles extinguished the platelet effect in reconstituting PdMos cytokine responses (Fig. 8C). Finally, we investigated the ability of enriched platelet-vesicles in reconstituting the defective cytokine response in NF-κB2 (NF-κB2$^{-/-}$) and NF-κB-p65 (RelA$^{-/-}$) knockout THP-1s. Recapitulating our previous findings (Fig. 7H), deficiency in NF-κB2 and NF-κB-p65 impaired cytokine responses to Pam3CysK4 in THP-1 knockouts. Likewise, platelets and platelet releasates (Plt Sups) were capable of boosting cytokine production in stimulated NF-κB2$^{-/-}$ and RelA$^{-/-}$ cells (Fig. 8D). Corresponding with the presence of NF-κB2 p52 in the vesicle fraction of Plt Sups, enriched vesicles were equally effective in enhancing the cytokine response of knockout cells as full platelet releasates (Fig. 8D). These findings substantiate the vesicle-mediated transfer of platelet-originated pro-inflammatory pathways to monocytes, underscoring a potentially crucial intercellular communication mechanism.

## Discussion

In this study, we reveal that platelets are essential to sustain monocyte inflammatory responses. We demonstrate that the activation of pro-inflammatory MAPK p38 and NF-κB signaling pathways propagate from platelets to human monocytes, sustaining the cytokine response of human monocytes. Removing platelets induces immunoparalysis in monocytes, marked by downregulated pro-inflammatory genes and impaired cytokine responses to PRR activation. This immunoparalysis is naturally present in monocytes from ITP patients but can be reversed by reintroducing fresh platelets. Platelets contain bioactive signaling molecules, including phosphorylated p38 (P-p38), and the active subunits of NF-κB (p65 and p52) which were detected in monocytes exposed to MK releasates, platelets, and platelet vesicles. Notably, platelets

dose-dependently reactivated MAPK and NF-κB signaling even in monocytes in which these pathways were genetically knocked out or pharmacologically inhibited, unveiling a novel mechanism of intercellular communication.

Thrombocytopenia, a life-threatening consequence of trauma and infections, is common in conditions causing immunoparalysis. However, confounding factors like co-infections or low leukocyte counts obscure the platelet contributions to immunoparalysis. We observed monocyte immunoparalysis in ITP patients without infections or immunosuppressive therapies, normal leukocyte counts and C-reactive protein levels. Thrombocytopenia in ITP compromises host defense, increasing risk of infections and morbidity (Birnie et al, 2019; de Stoppelaar et al, 2014; Qu et al, 2018). However, systemic levels of IL-1β, IL-6, and TNFα in ITP patients often match those of healthy individuals (Andreescu, 2023) and do not reflect clinical manifestations. Despite limitations, the platelet-depletion in mice mirrored the human ITP scenario, indicating weakened monocyte host defense transcriptional programs, without affecting plasma cytokine levels. Nevertheless, thrombocytopenic mouse monocytes stimulated ex vivo displayed diminished IL-1β response to inflammasome activation, which was restored with fresh platelets. It is important to highlight that antibody-based platelet depletion in vivo caused severe thrombocytopenia ($<15 \times 10^3/\mu L$), rarely seen in patients. Moreover, anti-GPIba treatment may have caused artifacts, as indicated by a transcriptional signature of platelet and complement activation in anti-GPIb treated mice. Given the difficulties in removing platelets from a living organism in a controlled manner, and mixed outcomes from previous studies (Carestia et al, 2019; Claushuis et al, 2016; van den Boogaard et al, 2015; Xiang et al, 2013), more research is needed to confirm these results in vivo and determine their clinical relevance.

Our findings reveal new insights into the pro-inflammatory functions of MPAs. We corroborated previous findings that platelets enhance monocyte migration (Chatterjee et al, 2015; Zuchtriegel et al, 2016), as addition of platelets or platelet releasates enhanced monocyte migration towards a CCL2 gradient. However, platelet-depletion did not affect this process (Fig. EV5). Platelet-depleted monocytes also displayed a slight increase in bacterial phagocytosis (Fig. EV5C,D), aligning with more proeminent roles of NF-κB and MAPK signaling in cytokine production.

We also confirmed previous findings that platelets are rich stocks of NF-κB subunits (Beaulieu and Freedman, 2009; Lannan et al, 2015; Spinelli et al, 2010). Furthermore, it has been known that interaction with platelets enhances the nuclear translocation of NF-κB within monocytes (Ueda et al, 1994; Weyrich et al, 1996; Weyrich et al, 1995). However, the possibility that a considerable portion of NF-κB signaling arises from platelets was neither raised nor experimentally demonstrated. We demonstrated that platelets and their vesicles reactivate TLR-induced cytokine secretion in RelA$^{-/-}$ and NF-κB2$^{-/-}$ THP1 monocytes, pinpointing platelets as the primary signal origin. Although platelets could also bypass p38 inhibition and reconstitute cytokines in SB203580-treated monocytes, our mass speck proteomics approach did not identify MEG-01-derived p38α on human monocytes, likely due to the differences between human platelets and MK cell lines, or the rapid turnover of these molecules in recipient cells. Furthermore, as MS is an abundance-based method, we cannot exclude the transfer of low amounts of p38α or downstream signaling contributing to pathway reactivation. Moreover, as NF-κB transcriptional activity requires functional p38 MAPK signaling (Anderson, 2010;

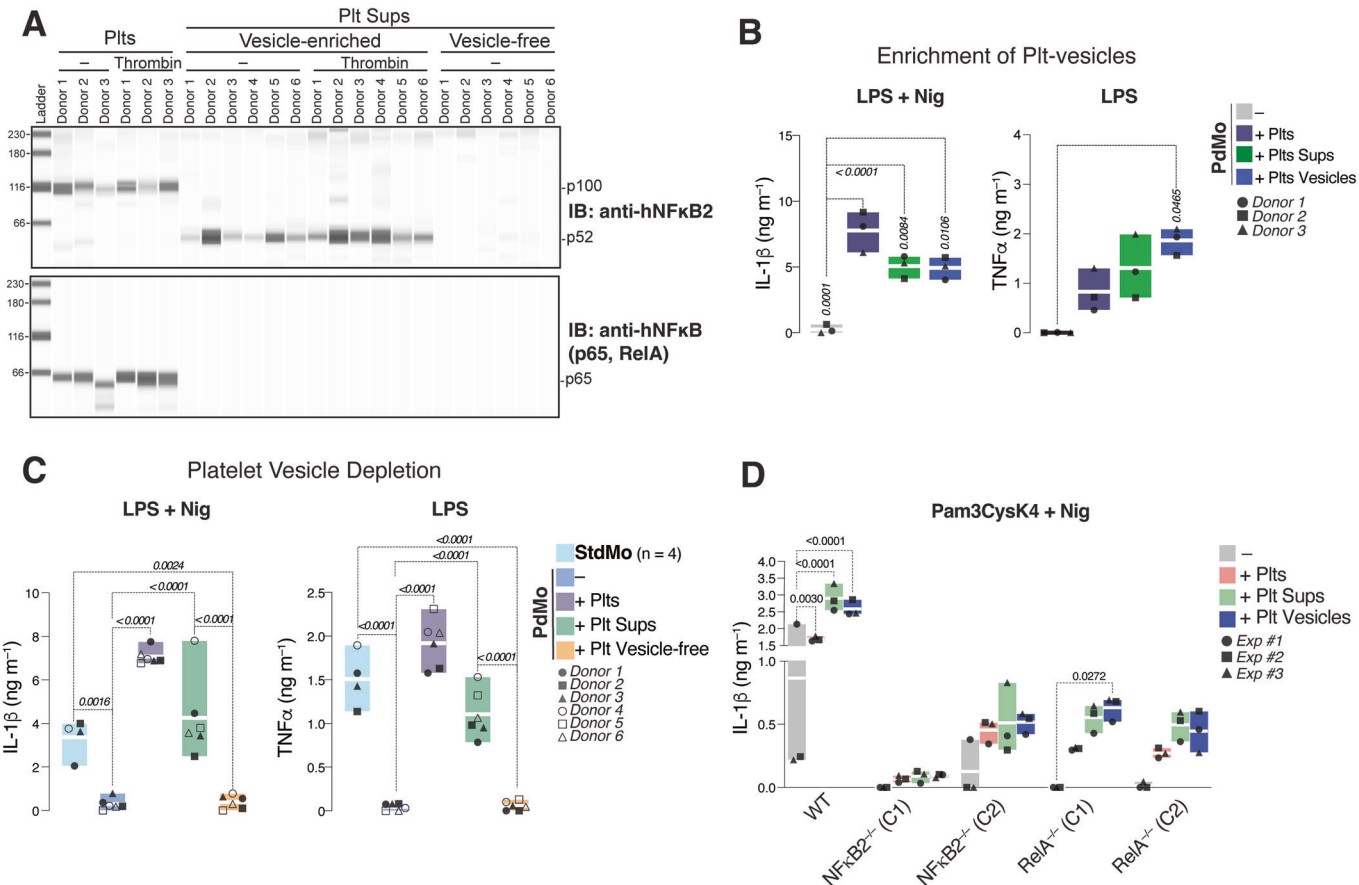

**Figure 8.  Platelet vesicles reconstitute NF-κB signaling in NF-κB⁻/⁻ monocytes.**

(A) WES capillary electrophoresis and immunoblotting of NF-κB2 on lysates, releasate, and vesicle-free releseates of purified platelets from six healthy volunteers. Platelets were left untreated or activated with 1 U ml⁻¹ of Thrombin. (B) Levels of IL-1β and TNFα in the CFS of PdMos that were replenished with platelets (+ Plts), unfractionated platelet releasates (+ Plt Sups), or isolated vesicles, obtained after vesicle-enrichment of platelet releasates (+ Plt vesicles). Graphs with floating bars depict maximum/minimum values relative to the mean (white bands). P values were calculated using one-way ANOVA, Tukey's multiple comparison test, and are indicated in the figure. Each symbol represents one donor (n = 3). (C) Assessment of IL-1β and TNFα levels in the CFS of StdMo, PdMos, and PdMos that were replenished with platelets (PdMo + Plts), unfractionated platelet releasates (+ Plt Sups) or releasates depleted of vesicles (+ Plt Vesicle-free). Vesicles were removed from the supernatant by size exclusion using a 100 kDa filter. Cells were stimulated with LPS (2 ng ml⁻¹) followed by activation with nigericin (10 µM). Graphs with floating bars show maximum/minimum values relative to the mean (white bands). P values were calculated using one-way ANOVA, Tukey's multiple comparison test, and are shown in the figure. Each symbol represents one donor (n = 6). (D) IL-1β levels released into the CFS of stimulated Wild-type WT, RelA⁻/⁻, or NF-κB2⁻/⁻ THP-1 monocytes that were cultured alone or added with platelets, unfractionated platelet releasates (+ Plt Sups) or isolated vesicles, obtained after vesicle-enrichment of platelet releasates (+ Plt vesicles). Clones were stimulated with Pam3CysK4 (1 µg ml⁻¹) or Pam3CysK4 and nigericin (10 µM) (Pam3 + Nig). Graphs with floating bars illustrate maximum/minimum values relative to the mean (white bands). P values were calculated using two-way ANOVA, Tukey's multiple comparison test, and are displayed in the figure. Each symbol represents one independent experiment or blood donor (n = 3). For simplicity, only stimulated conditions (LPS ± Nig and Pam3CysK4 ±Nig) are shown. Additional conditions are represented in Appendix Fig. S8.

Gottschalk et al, 2016; Guma et al, 2011; Saha et al, 2007), platelets may circumvent SB203580 inhibition by supplying NF-κB signaling to monocytes. Supporting this hypothesis, SB203580 also reduced the phosphorylation of p65 (RelA) in treated monocytes (Fig. EV4E). Finally, specific ablation of p38 MAPK in platelets confers protection against inflammatory heart diseases (Shi et al, 2017) and platelet-activating factor (PAF)-induced lethality (Abhilasha et al, 2019), indicating a role for platelet p38 signaling in inflammatory diseases (Anand et al, 2011; Christie et al, 2015; Yong et al, 2009).

Our experiments ruled out common ligand-receptor interactions in platelet-monocyte crosstalk, suggesting that p38 and NF-κB signaling propagate through platelet vesicles. Indeed, platelet vesicles

are known mechanism for the transfer of cellular receptors (Rozmyslo-wicz et al, 2003), bioactive lipids (Barry et al, 1997; Rossaint et al, 2016) intact organelles (Levoux et al, 2021) from platelets to leukocytes.

In our study, we consistently observed that platelet lysates primarily contained the inactive form of NF-κB2 (p100), while the active form (p52) was predominant in releasates, regardless of platelet activation. These results connect the active processing of NF-κB2 at the protein level and their functional restoration in monocytes. Using samples from the same donors, we demonstrated that isolated platelet vesicles effectively restore the impaired cytokine response of PdMos, while vesicle-free supernatants failed to do so.

Our findings raise questions on the evolutionary dependency on platelets for human monocyte inflammation. In lower vertebrates,

hemocytes perform both immune defense and coagulation functions. In mammals, these roles have evolved into separate cells—megakaryocytes, platelets, and monocytes—each with distinct functions. However, some intercellular dependencies may have persisted to prevent unwanted activation, exemplifying the growing intercellular links between coagulation and innate immunity (Burzynski et al, 2019; Wu et al, 2019).

## Study limitations

Our study demonstrates that platelets play a crucial role in the pro-inflammatory cytokine response of monocytes, using primary human monocytes from healthy individuals and patients with inherited platelet abnormalities. However, given the limitations of human studies, most tests were conducted in vitro with human cells stimulated ex vivo. Moreover, antibody-based thrombocytopenia in vivo does not fully represent clinical scenarios.

We also found notable differences in the effects of the IKK phosphorylation inhibitor (BAY 11-7082) on primary human monocytes versus the THP-1 cell line. While platelets enhance IL-1β secretion from THP-1 cells, these cells do not depend on platelets for primary IL-1β responses and rely less on specific NF-κB pathways compared to primary cells. THP-1 cells may activate alternative NF-κB pathways, as suggested by p50 binding to cRel (Rauert-Wunderlich et al, 2013; Shih et al, 2012). The complexity of the NF-κB pathway varies between cell types. For example, RelB promotes dendritic cell activation through RelB-p50 dimers in the canonical NF-κB pathway (Sun, 2012). Platelets restored IL-1β production in THP-1 knockout clones lacking RelA and NF-κB2, key members of the canonical and non-canonical pathways, indicating challenges in identifying specific NF-κB subunits. Furthermore, similar to the p100 subunit of NF-κB2, the p65 RelA subunit was enriched on platelets, but absent in platelet releasates. However, as the small subunit of NF-κB2 (p52) was encapsulated in vesicles, it may be the case for the p50 subunit of RelA, not investigated in this study. Hence, identifying specific NF-κB subunits remains challenging (Kasper et al, 2010), and additional experiments are needed to elucidate the precise mechanisms and course of interactors of each pathway. Our SILAC mass spectrometry identified additional platelet-derived transcription factors (TFs) in monocytes, including STAT1, THYN1, and STAT3 (Appendix Fig. S6). These molecules may influence monocyte cytokine responses, but their significance requires further investigation.

In summary, our study introduces a novel mechanism of platelet regulation on monocyte immunity, highlighting the propagation of platelet-derived NF-κB and p38 signaling as trans-cellular regulators of cytokine inflammation. Our research points towards new ways for cells to communicate, potentially reshaping treatments for inflammatory diseases and improving outcomes in conditions involving aberrant of dysfunction monocyte activity. Additionally, this discovery could refine platelet transfusion techniques, which are generally safe but occasionally cause inflammatory reactions.

# Methods

## Reagents

Cell culture reagents (e.g., PBS, Fetal calf serum, GlutaMax, and RPMI) were from Gibco, Thermo Fisher Scientific. Stimuli were as follows: LPS (Ultrapure from E. coli O111:B4), Pam3CysK4, and

R848 (Resiquimod) were from Invivogen. Nigericin acid was from Thermo Fisher Scientific. Inhibitors used were SB203580 (Tocris), Dynasore (Merck), Cytochalasin D, BAY 11-7082 (Sigma-Aldrich). The CellTiter-BlueTM Cell Viability Assay and Caspase-Glo. 1 Inflammasome Assay was from Promega. HTRF kits to detect IL-1β, IL-6, TNFa, and NF-κB were from Cisbio. ProcartaPlexTM kits for detecting other Cytokine/Chemokine/Growth Factors were from Invitrogen. Antibodies were anti-IL-1β antibody (BAF401, R&D Systems, 1:1000). DRAQ5 was from Thermo Fisher Scientific. For FACS, we used FcR Blocking Reagent, human and mouse (Miltenyi Biotec). Anti-human CD41 directly conjugated to Alexa Fluor 488 (Clone: HIP8), Anti-human CD14-A647 (Clone: HCD14), and isotype Mouse IgG1, k Isotype Ctrl (Clone: MOPC-21) Antibody were all from BioLegend.

## Study subjects

Peripheral blood was collected by venipuncture from healthy volunteers after the signature of informed consent and approval of the study by the Ethics Committee of the University of Bonn (Protocol #282/17), following the Declaration of Helsinki. Patients with immuno-thrombocytopenia were recruited from a tertiary hematology referral hospital. We recruited five patients (3 females and two males, average age of 46 ± 9) from December 2019 to May 2020. All patients were diagnosed with ITP, and comorbidities/infections were ruled out. Written informed consent was obtained from all subjects according to the Declaration of Helsinki, and the experiments are in conformity with the principles set out by the Department of Health and Human Services Belmont Report. Our experiments received approval by the Institutional Review Board of the University of Bonn (313/21). All participants were instructed about the study's objectives and signed an informed consent following guidelines for human research.

## Primary cells

Primary human monocytes were isolated from peripheral blood collected from healthy volunteers in S-Monovette K3EDTA tubes. Blood was diluted 1:3 in PBS, and peripheral blood mononuclear cells (PBMCs) were isolated using Ficoll® Paque PLUS density gradient centrifugation at $700 \times g$, 20 min. Primary human CD14$^+$ monocytes were isolated from PBMCs using positive magnetic separation with the Classical Monocyte Isolation Kit (Miltenyi Biotech) or negative selection with the EasySep™ Human Monocyte Isolation Kit (STEMCELL™ Technologies) following manufacturer instructions. Platelet-depleted monocytes were generated by adding a Platelet Removal Component (50 μl ml$^{-1}$) supplied with the isolation kit (STEMCELL™ Technologies). The purity of monocyte populations was assessed by flow cytometry. Cells were incubated with FcR Blocking Reagent for 15 min at 37 °C, followed by staining with fluorochrome-conjugated monoclonal antibodies against anti-CD14 APC (eBioscience), anti-CD45 PE (eBioscience), and anti-CD41a FITC (eBioscience) for 30 min at RT. Cells were washed with PBS, and fluorescence was measured on a MACSQuant® Analyzer 10.

## Platelet isolation from human blood

Human platelets were isolated from venous blood from healthy volunteers collected as previously described (Alard et al, 2015).

Briefly, whole blood was centrifuged at $330 \times g$ for 15 min, and platelet-rich plasma (PRP) was collected. To prevent platelet aggregation during isolation, PRP was treated with 200 nM Prostaglandin E1 (PGE1), diluted 1:2 with PBS, and centrifuged at $240 \times g$ for 10 min to pellet leukocytes. Platelet suspensions were collected and washed by centrifugation at $430 \times g$ for 15 min in the presence of PBS with PGE1. Washed platelets were pellet and resuspended in pre-warmed RPMI. Human platelets were stimulated with $1 \ U \ ml^{-1}$ thrombin from human plasma (Sigma-Aldrich) or left unstimulated for 30 min to assess the purity and activation of isolated cells. Platelets were incubated with FcR blocking reagent (Miltenyi) followed by staining with the leukocyte marker anti-CD45 PE (eBioscience), and the platelet marker anti-CD41a FITC (eBioscience) and the platelet activation marker anti-CD62P APC (eBioscience).

## Platelet isolation from mouse blood

Wild-type C57BL/6 mice were euthanized using $CO_2$, and death was confirmed by the absence of the hindlimb reflex. Following this, the intrathoracic cavity was exposed, and cardiac blood was taken from the still-beating hearts and slowly collected using pre-coated citrate syringes with a 21G needles (BD). The isolated blood from three to four mice was pooled in a citrate tube (Sarstedt) and diluted 1:2 with pre-warmed HBSS (Gibco). Samples were pelleted by centrifugation at $250 \times g$, 10 min at 22 °C (acceleration 1, brake 0) to obtain platelet-rich plasma (PRP). The PRP was transferred to a new tube and treated with 100 nM PGE1 (SIGMA). Platelets were then pelleted by centrifugation at $1250 \times g$, 20 min at 22 °C (acceleration 1, brake 0) and resuspended in supplement-free RPMI. Platelet count was adjusted to a concentration of $2 \times 10^8 \ ml^{-1}$. Purity was assessed by Flow Cytometry using the Attune NxT flow cytometer from ThermoFisher, after 1:10 mouse FcR blockage (Miltenyi) and immunofluorescence staining with followed by staining with the leukocyte marker 1:200 anti-mouse CD45 FITC (BioLegend, clone: HI30), and the platelet marker 1:50 anti-mouse CD41a eFluor450 (eBioscience, clone: ebioMWReg30) and the platelet activation marker 1:200 anti-mouse CD62P APC (eBioscience, clone: Psel.KO2.3) and analyzed with the software FlowJo (version 10.10.0). For inflammasome assays, $1 \times 10^6$ platelets per well were added to $1 \times 10^5$ monocytes in 96-well tissue culture plates.

## Isolation of mouse blood monocytes

For the isolation of blood monocytes, cardiac blood was isolated from wild-type C57BL/6 mice, as described above. Peripheral blood mononuclear cells (PBMCs) were obtained through Ficoll-Paque® PLUS (VWR) gradient. Briefly, polled blood from three to four mice was placed into a 15 mL falcon tube, totaling 1.5 to 2 mL per tube. Blood was diluted with 5–6 mL PBS (Gibco). Diluted blood was layered on the top of 4.5 mL Ficoll in a new 15 ml falcon tube and centrifuged at $850 \times g$, 20 min at 22 °C (acceleration 1, brake 0). The PBMC layer was harvested and filtered through a 40 µm nylon mesh into another 15 ml falcon tube and diluted with PBS to 15 ml. Cells were centrifuged at $550 \times g$, 10 min at 22 °C, and the pellet resuspended in 90 µL of MACS buffer per $10^7$ total cells. Monocytes were isolated following the manufacturer's instructions (CD11b MicroBeads human and mouse, Miltenyi) except for centrifugation steps at $550 \times g$ instead of $300 \times g$. For isolation and immunophenotyping, 10 µL of CD11b MicroBeads were added per $10^7$ total cells, incubated for 15 min at 4 °C. After washing with 1–2 mL of

MACS buffer per $10^7$ cells and centrifugation at $550 \times g$ for 10 min, up to $10^8$ cells were resuspended in 500 µL of MACS buffer. Each mouse's cell suspension was applied onto a column in a suitable MACS Separator, washed three times with 500 µL MACS buffer, and then eluted into a 50 ml falcon tube. Positive magnetic selection was employed using CD11b as a marker, allowing the isolation of $CD45^+ \ CD11b^+ \ Ly6C^+$ cells (Fig. EV2E). For purity and quality control, the cells were stained with the platelet marker 1:50 anti-mouse CD41a APC (eBioscience, clone: ebioMWReg30), the integrin 1:200 anti-mouse CD11b PE (eBioscience, clone: M1/70), the mouse monocyte marker 1:200 anti-mouse Ly6C eFluor450 (eBioscience, clone: HK1.4) and the leukocyte marker 1:200 anti-mouse CD45 FITC (BioLegend, clone: HI30). Cells were treated with 1:10 mouse FcR block (Milteny), stained with a mix of antibodies, washed, and analyzed using the Attune NxT flow cytometer from ThermoFisher and FlowJo software (version 10.10.0). Cells were counted, adjusted to a concentration of $2 \times 10^6 \ ml^{-1}$, and plated at $1 \times 10^5$ cells per 96-well plate.

## Generation of platelet supernatants

Platelet supernatants were generated from suspensions of $1 \times 10^8$ human platelets incubated in RPMI and left untreated (Plt Sups) or stimulated with $2 \ ng \ ml^{-1}$ LPS (PltsLPS-Sups) for 3 h at 37 °C. Platelets were pelleted by centrifugation at $500 \times g$ for 10 min. Next, supernatant was transferred into a new falcon and centrifuged at $3000 \times g$ for 10 min. The platelet supernatants were harvested and used to stimulate human monocytes.

## Generation of vesicle-enriched platelet supernatant

Platelets were isolated, and the platelet supernatant was generated as described above. After the supernatant was cell-free, 1–1.5 ml of the platelet supernatant was placed in 1.5 ml ultra-centrifuge tubes (Beckman-Coulter). Subsequently, the supernatant was ultra-centrifuged for 2 h at $100,000 \times g$ at 4 °C to concentrate the vesicles in the supernatant (Beckman-Coulter Optima Max-XP, Rotor TLA-55). Then, the supernatant was carefully removed, leaving only 100 µl in the tube. Tubes from the same donor were then pooled, and to each well 50 µl of the concentrated platelet vesicles were added to $1 \times 10^5$ THP-1 or primary monocytes (both in 50 µl volume). Aliquots were taken from each donor and analyzed using the vesicle detection device NanoSight NS300 (Malvern Panalytical). For this, 100 µl of the sample was diluted with 900 µl of PBS (1:10) and then placed into the analysis chamber of the NanoSight NS300. Each sample was recorded three times for 60 s and then analyzed by the NTA 3.1 software. Between each sample, the chamber was washed twice with PBS, and at the beginning, before recording all samples, a background control was taken with PBS.

## Generation of vesicle-free platelet supernatant

Platelets were isolated, and platelet supernatant was generated as described above. After the supernatant was made cell-free, it was transferred and again centrifuged at $3000 \times g$ for 10 min. Subsequently, the supernatant was sterile-filtered through a 0.2 µm filter (Whatman™, VWR). The sterile-filtered supernatant was ultra-filtered using 100 kDa 50 ml Amicon columns (Merck) via centrifugation, following the manufacturer's protocols for the

filter's use. The flow-through was validated for vesicle-free content using NanoSight NS300 (Malvern Panalytical) and NTA 3.1 software, following the procedure described above. 50 µl of the vesicle-free supernatant were used per well.

## Cell lines

The human monocytic THP-1 cell line (ATCC TIB-202) and the THP-1 Dual™ reporter cell (InvivoGen, Thpd-nfis) were culture in RPMI supplemented with 10% heat-inactivated fetal calf serum, 1% Penicillin/Streptomycin and 1% GlutaMax. For THP-1 DualTM cells, the growth medium contained additionally 25 mM HEPES. Cells were either studied in a monocyte-like phenotype (without PMA-differentiation) or after their differentiation into macrophages by PMA (50 µM) treatment for 24 h. Human megakaryocytic cell line MEG-01 (ATCC CRL-2021) were cultured in RPMI-1640 medium supplemented with 10% FBS, 1% Penicillin/Streptomycin and 1% GlutaMAX (1x) at 37 °C with 5% $CO_2$.

## Stimulation Assays

Monocytes ($1 \times 10^5$/well) were cultured alone or co-cultured with 1:50 ($5 \times 10^6$/well) or 1:100 platelets ($1 \times 10^7$/well). For stimulation, cells were treated with agonists of TLR1/2: Pam3CysK4 (1 µg ml$^{-1}$); TLR4: LPS (2 ng ml$^{-1}$); or TLR7/8: Resiquimod R848 (3.5 µg ml$^{-1}$); for 4.5 h. For inflammasome activation, cells were primed with a TLR agonist for 3 h (indicated in Figs), followed by activation with nigericin (10 µM, 90 min) PrgI from *Bacillus anthracis* (100 ng ml$^{-1}$) and protective antigen (PA, 1 µg ml$^{-1}$), for 90 min) or transfection of poly(dA:dT, 0.5 µg ml$^{-1}$, 3 h). After stimulation, CFS was harvested and used for cytokine detection.

## Cytokine measurement

Levels of human, or mouse IL-1β, IL-6, and TNFα in cell-free supernatants or whole cell lysates were quantified using commercially available HTRF (homogeneous time-resolved fluorescence, Cisbio) kits. The HTRF was performed according to manufacturer instructions. A Human ProcartaPlexTM (Invitrogen) immunoassay was additionally used to detect 45 human cytokines, chemokines, and growth factors, according to the manufacturer's instructions, using a MAGPIX® System.

## Cell viability assays

Cell viability was determined by measuring the release of Lactate Dehydrogenase Assay (LDH) in CFS or the CellTiter-Blue Assay (CTB) in stimulated cells. For LDH measurement, CFS from stimulated cells were incubated with LDH buffer for 30 min at 37 °C without $CO_2$, and absorbance was measured at 490 nm using a Spectramax i3 plate reader (ThermoFisher). Cells were incubated with CTB buffer at 37 °C, and the fluorescence was read at different time points in both assays, cells treated with Triton X-100 served as a 100% cell death control. Fluorescence was detected with a SpectraMax.

## Assessment of caspase-1 activity

The specific activity of caspase-1 was assessed in cell-free supernatants using Caspase-Glo® 1 Inflammasome Assay (Promega).

Briefly, CFS was mixed 1:1 with Glo Buffer and incubated for 30 min at RT. Luminescence was measured with SpectraMax i3 (Molecular devices).

## Imaging and quantification of ASC specks

For the quantification of ASC specks, human monocytes and platelets were seeded into a Poly-L-Lysine-coated (Sigma-Aldrich) 96-well microscopy plate (CellCarrier 96 Ultra Microplate, Perkin Elmer) at a density of $5 \times 10^4$ cells per well. Cells were primed for 3 h with LPS (2 ng ml$^{-1}$) and stimulated with Nigericin (10 µM) for 1.5 h in the presence of VX-765 (50 µM, Selleckchem) to avoid the release of ASC specks. Afterwards, the plate was centrifuged ($500 \times g$, 5 min) and the cells were fixed with 4% paraformaldehyde (Thermo Fisher) for 10 min at RT. After two washes with PBS, samples were permeabilized and blocked with Perm/Block Buffer (1% FCS, 10% Goat Serum, 0.5% Triton X-100, in PBS) for 30 min at 37 °C and stained with anti-ASC Antibody (TMS-1, Biolegend, clone HASC-71, 1 µg ml$^{-1}$) or the same amount of IgG1k isotype control antibody (Biolegend, clone MOPC-21) and incubated overnight at 4 °C. On the next day, cells were washed twice with Perm/Block buffer ($500 \times g$, 5 min) before staining with Goat anti-mouse IgG (H + L) Cross-Adsorbed Secondary Antibody (AF488-conjugated, Thermo Fisher, 1 µg ml$^{-1}$) and the DNA dye DRAQ5 (Thermo Fisher, 1:5000 dilution) in Perm/Block buffer for 60 min at 37 °C in the dark. After two subsequent washes with Perm/Block buffer, cells were washed with PBS and imaged using a ZEISS Celldiscoverer 7 microscope, 20× objective (dry, PlanApochromat, NA 0.95 autocorr, Axiocam 506 mono), and ZEN Blue software (ZEISS). The quantification of ASC specks per nuclei was carried out by using an image analysis pipeline constructed in the CellProfiler 3.9.1 software (Open source software: https://cellprofiler.org). Briefly, after automated correction for even illumination of each image, nuclei and ASC specks were quantified per field of view. ASC speck signal was previously enhanced by the EnhanceOrSuppressFeatures tool.

## Confocal imaging

Platelet interactions with human monocytes were imaged in a Leica TCS SP8 SMD confocal system (Leica Microsystem). Cells were seeded in microslide 8-Well IBIDI chamber, fixed with 4% paraformaldehyde (PFA) for 30 min, washed and incubated with staining mAbs in permeabilization/blocking buffer (10% goat serum, 1% FBS, and 0.5% Triton X-100 in PBS) supplemented with human FcR Blocking Reagent for 30 min. Cells were stained overnight at 4 °C with anti-human CD41a AlexaFluor488, anti-human CD14 A647, or Hoechst 34580 (ThermoFisher).

## Bacteria phagocytosis assays

Primary human monocytes (StdMo or PdMo) were exposed to an MOI of 1:25 of an *E. coli* strain (DH5alpha) constitutively expressing mCherry (BW25713, pFCcGi, https://www.addgene.org/59324/), for 2 h. Cells were also left untreated, or pre-treated with cycochalasin D (50 µm) before being exposed to a higher MOI of bacteria (100:1). Samples were fixed with 4% PFA, stained with 1:5000 DRAQ5 (nuclei) and 1:100 WGA AF488 (membranes) and imaged on a confocal and automated widefield microscope. Images (four per condition) were

analyzed by counting the *E. coli* inside five monocytes per image, counted by two independent experimenters. Mean numbers of *E. coli* per cell were plotted.

## Monocyte trans-migration assays

Primary monocytes (StdMo, or PdMo) were supplemented with platelets, platelet (Plt Sups) or Mk releasates (MK Sups) were seeded on the upper chamber of a trans well plate with either 3 μm or 5 μm pore sizes and incubated with CCL2 (40 ng ml$^{-1}$) or left untreated for 4 h. Cell migration was measured by confocal imaging and quantification of cells that migrated to the bottom well and stained with DRAQ5. 16 pictures per well were taken and the nuclei were counted via cell profiler. Log2 fold change was calculated and the CCL2 conditions were normalized to their respective unstimulated condition.

## Immunoblotting

Cells were lysed with RIPA complete lysis buffer supplemented with EDTA-free protease and phosphatase inhibitors (Roche) and 25 U of Benzonase Nuclease. Samples were loaded onto NuPAGETM Novex 4–12% Bis-Tris gels and transferred to Immobilon-FL Polyvinylidene fluoride (PVDF) membranes. After blocking with 3% BSA [w/v] in Tris-buffered saline (TBS), membranes were incubated overnight with primary antibodies diluted in TBS containing 3% BSA [w/v] and 0.1% Tween. After washing steps, membranes were incubated with secondary antibodies conjugated to IRDye680 or IRDye800 (1:25000) for 2 h at RT. The membrane was scanned at Odyssey Imager (Li-Cor Biosciences). The following primary antibodies were used for: β-Actin rabbit mAb (1:5000), Phospho-NF-κB p65 (Ser536) (93H1) Rabbit mAb #3033 (1:1000), NF-κB p65 (C22B4) Rabbit mAb #4764 (1:1000) and IkBa (L35A5) Mouse mAb (Amino-terminal Antigen) #4814 (1:1000).

## Platelet depletion in vivo

In vivo, platelet depletion was performed in female C57BL/6J mice (12 weeks) by intravenous (i.v.) injection of 2 mg kg$^{-1}$ of polyclonal rat anti-mouse GPIbα (R300) or same amounts of a polyclonal rat IgG none-immune (C301) antibody as control (Emfret Analytics). Mice were then challenged 12 h later with i.v. injection of 2 mg kg$^{-1}$ LPS or PBS as control. Blood was collected after 3 h, and PBMCs were isolated by density gradient centrifugation in Ficoll Paque PLUS. Mouse monocytes were separated from PBMCs by FACS-sorting gating on Ly6G$^-$, IA/IE$^-$, CD45$^+$, CD11b$^+$, and CD115$^+$ cells and sorted directly into QIAzol lysis reagent. Lysates were frozen in liquid nitrogen and sent to GENEWIZ Azenta Life Sciences for bulk RNA sequencing. All animal experimentation was approved by the local ethical committee (LANUV-NRW #84-02.04.2016.A487). Mice were purchased from Charles River and housed at the House for Experimental Therapy (HET) of the University Hospital of Bonn.

## Fluorescence-activated cell sorting was used to obtain mouse monocytes

Mouse blood monocytes were isolated from pooled blood of 4 mice per experimental group in 3 different experiments. For this, mouse

PBMCs were isolated through Ficoll gradient and stained with anti-Ly6G BV785 (Biolegend, dilution: 1:40), anti-I-A/I-E BV785 (Biolegend, dilution: 1:80), anti-CD45 FITC (Biolegend, dilution: 1:200), anti-CD11b PE (eBioscience, dilution: 1:166), anti-CD115 APC (Biolegend, dilution: 1:80) and Hoechst 33258 (Abcam, 1 μM) as viability staining. Finally, viable Ly6G I-A/I-E- CD45+ CD11b+ CD115+ monocytes were sorted directly into QIAzol lysis reagent using the BD FACS Aria Fusion (387333827) and BD FACS Aria III (216372545) at the Flow Cytometry Core Facility of the Medical Faculty of the University of Bonn.

## Bulk RNA sequencing of isolated mouse monocytes

For bulk RNA sequencing, samples were processed according to GENEWIZ (Azenta Life Sciences) pipeline. RNA extraction was performed, followed by a library preparation. Ultra-low input RNA-Seq was performed at Illumina HiSeq PE 2 × 150 bp with ~350 M reads. Reads were then mapped to the Mus musculus GRCm38 reference genome (ENSEMBL) using the STAR aligner v.2.5.2b, and unique gene hit counts were generated with the featureCounts from the Subread package v.1.5.2. Finally, standard analysis was performed by GENEWIZ. Differential gene expression analysis was performed using DESeq2, *p*-values, and log2 fold changes were calculated by applying the Wald test, and genes with adjusted *p*-value < 0.05 and absolute fold change >2 were considered significantly altered differentially expressed genes (DEGs). Gene ontology (GO) analysis of the significant DEGs was applied by GENEWIZ (Azenta Life Sciences) using the Fisher exact test. GO terms were assigned using the GeneSCF v.1-p2 software. Data visualization was created using the R package ggplot2.

## Transcriptional profiling of human monocytes and platelets with NanoString

According to the manufacturer protocol, the nCounter® Human Myeloid Innate Immunity Panel v2 (NanoString® Technologies) was used to assess the mRNA expression of 770 human transcripts. Briefly, single or co-cultures of monocytes and platelets were lysed in RLT Buffer (Qiagen) containing β-Mercapto-ethanol (1 × 10$^4$ cells μl$^{-1}$). RNA was isolated with and homogenized with CodeSets and left for hybridization overnight. RNA/CodeSet complexes were immobilized on nCounter cartridges at the nCounter® Prep Station 5 s, and data collection (RCC files) and quality check were performed in the nCounter® Digital Analyzer 5 s. The nSolver™ Analysis and Partek® Genomics Suite® software was used for analysis.

## Kinase activity profiling microarray

Primary monocytes and platelets were isolated from 6 healthy donors. Monocytes were stimulated ex vivo before or after the removal of platelets or after the re-addition of platelets to PdMo (biological replicates with *n* = 6). As before, platelets alone (*n* = 6) were assessed. Stimulated cultures were lysed in M-PER mammalian protein extraction reagent supplemented with PhosphoSTOP (Roche) and cOmplete Tablets (Roche) incubated on ice for 15 min. After centrifugation at 20,000 × g for 15 min at 4 °C, the lysate was

aliquoted, snap-frozen, and stored at −80 °C. Only non-thawed aliquots were used for the kinase activity assay.

Ser/Thr Kinase (STK) activity profiling assays based on measuring peptide phosphorylation by protein kinases (PamGene International BV, The Netherlands) were performed as instructed by the manufacturer. In summary, samples with 1 µg protein were applied on PamChip 4 arrays containing 144 (STK) or 196 (PTK) peptides immobilized on a porous aluminum oxide membrane. The peptide sequences (13 amino acids long) harbor phosphorylation sites and are correlated with one or multiple upstream kinases. Fluorescently labeled anti-phospho antibodies detect phosphorylation activity of kinases present in the sample. Instrument operation and imaging are controlled by the EVOLVE 2.0 software and quantified using BioNavigator 6.3 (BN6; PamGene International BV, The Netherlands). Signal intensities at multiple exposure times were integrated by linear regression (S100), Log2-transformed, and normalized using a Combat correction model for batch correction where the scaling parameters (mean/SD) are estimated using an empirical Bayes approach, which effectively reduces the noise associated with applying the correction.

The normalized values were used to perform statistics comparing groups or the upstream kinase analysis (UKA) tool (BN6; PamGene International BV). The following statistical test is used to generate a list of differentially significant phosphorylated peptides: Paired T-test for LPS effect/comparisons and ANOVA-PostHoc Test for multiple treatments versus control for comparing each group to StdMo unstimulated in both unstimulated and LPS-15 min conditions. For Kinase interpretation (differential): To generate a ranked list of putative kinases responsible for differences in the peptide phosphorylation, PamGene's in-house method called Upstream Kinase Analysis (UKA) was used. For Pathway interpretation (differential): To generate a ranked list of possible canonical pathways (and networks) responsible for differences in the peptide phosphorylation, there are many open-source tools to perform this.

The phylogenetic kinome tree, applicable to group the kinases into sequence families, is plotted using the online portal CORAL: http://phanstiel-lab.med.unc.edu/CORAL/. The upstream kinase analysis functional scoring tool (PamGene International) rank-orders the top kinases differential between the two groups, the ranking factor being the final (median) kinase score (represented by node size). This score is based on a combined sensitivity score (difference between treatment and control groups, expressed as node color) and specificity score for a set of peptides to kinase relationships derived from existing databases. An arbitrary threshold of a final score 1.2 was applied based on the specificity scores. Significant peptides (t-tests, $p$-value < 0.05) or kinases (UKA, final scores >1.2) were imported to the MetaCore pathway analysis tool (Clarivate Analytics), where an enrichment analysis was performed for pathways and networks. It matches the kinases or substrates in the kinome array data with functional ontologies in MetaCore. The probability of a random intersection between a set of IDs in the target list with ontology entities is estimated in the $p$ value of hypergeometric intersection. The lower p-value means higher relevance of the entity to the dataset, which shows a higher rating for the entity. Direct interaction network algorithms were used to build interconnected networks within each comparison. The "Add interactions" feature was used to add the interaction between RIPK1 and the data in the MetaCore™ database after it was built.

# Mass spec proteomics combined with stable isotope labeling by amino acids in cell culture (SILAC)

### Sample preparation for LC-MS analysis

The megakaryocytic cell line MEG-01 was cultured in SILAC RPMI 1640 Flex medium without D-glucose, L-arginine, D-lysine and phenol red (Gibco). The medium was supplemented with 10% dialyzed FBS (Thermo Fisher Scientific), "heavy" $^{13}C_6$ $^{15}N_4$ L-arginine and $^{13}C_6$ $^{15}N_2$ L-lysine (Silantes). They were cultured for five passages on heavy isotope amino acids to ensure the complete incorporation of heavy isotopes into the proteins (Ong et al, 2002). SILAC-labeled MEG-01 cells were tested by LC-MS proteomics analysis to ensure a heavy isotope amino acid incorporation of at least 95%. For experimental use, the cells were washed twice with PBS and then resuspended in RPMI without supplements, adjusting the concentration to $1 \times 10^6$ cells ml$^{-1}$. The cells were then incubated at 37 °C for 3 h. To collect the cell-free supernatant, the cells were pelleted at $170 \times g$ for 7 min, and the supernatant was transferred and centrifuged again at $3000 \times g$ for 10 min to remove any cellular debris. The transferred supernatant was added to $1 \times 10^5$ primary human monocytes at a volume of 100 µl per well. Subsequently, the cells were either stimulated with LPS at a concentration of 2 ng ml$^{-1}$, additionally with nigericin at 10 µM, or left untreated, as previously described. Following stimulation, the plates were centrifuged at $500 \times g$ for 5 min. 60 µl of supernatant were transferred into new 96-well plates for HTRF validation of IL-1β and TNFα, if the donors responded. The remaining supernatant was also aspirated, and the cells were carefully washed twice with PBS. Subsequently, the primary cells were lysed in 50 µl of 1% sodium deoxycholate (SDC, Merck) in Tris-HCl (100 mM, pH 8.5) supplemented with Benzonase (7.5 U ml$^{-1}$, Merck). Samples were incubated at room temperature for 30 min. Following, protein disulfide bridges were reduced and alkylated with 55 mM Bond-Breaker TCEP solution (Thermo Fisher Scientific) and 10 mM chloroacetamide (CAA, Sigma, Merck) by incubation at 95 °C for 10 min. Protein concentrations were determined with the Pierce 660 nm protein assay (Thermo Fisher Scientific) and 20 µg of protein was used for digestion. Trypsin/Lys-C digestive enzyme (Promega) was added in a 1:25 enzyme-to-protein ratio and digested overnight at 37 °C and 800 rpm shaking. On the next day, digestion was stopped by adding stop buffer (1% formic acid in isopropanol) in a 5:1 buffer-to-sample volume ratio. After centrifugation at full speed for 5 min, samples were desalted using the stage tip cleanup protocol (Rappsilber et al, 2007). Briefly, in-house-made stage tips filled with Styrol-divinylbenzene Reversed Phase Sulfonate (SDB-RPS, Affinisep) were activated with 20 µl methanol, washed once with 20 µl of buffer B (80% acetonitrile, 0.1% formic acid, LC-MS-grade, VWR) and equilibrated twice with 20 µl buffer A (0.1% formic acid, LC-MS-grade VWR). Samples were loaded onto the stage tips and were desalted by washing the tips once with 100 µl buffer A and twice with 100 µl buffer B. Peptides were eluted with 60 µl of freshly prepared buffer X (5% ammonia, 80% acetonitrile in LC-MS-grade water). After each step, tips were centrifuged at ~800 × g for 20 s or until all liquid passes through the SDB-RPS material. Eluted peptides were vacuum-concentrated and following resuspended in buffer R (2% acetonitrile, 0.1% formic acid in LC-MS-grade water). Peptide concentrations were determined and 150 ng per sample used for LC-MS analysis.

## Liquid chromatography-mass spectrometry

Proteomics samples were measured in data-dependent acquisition (DDA) mode with a liquid chromatography-tandem mass spectrometry system consisting of a Vanquish Neo chromatographic system (Thermo Fisher Scientific) and an Orbitrap Exploris 480 mass spectrometer (Thermo Fisher Scientific). An in-house packed analytical column filled with 1.9 μm ReproSil-Pur 120 C18-AQ material (Dr. Maisch) and a length of 30 cm was used to separate peptides based on their hydrophobicity. Peptide separation was achieved with a binary buffer system (buffer A: 0.1% formic acid in LC-MS-grade water, buffer B: 80% ACN, 0.1% formic acid in LC-MS-grade water) over a 120 min chromatographic gradient at a constant flow rate of 350 nl/min. The amount of buffer B was linearly increased from 6% to 32% over 92 min followed by a linear increase of B to 55% over 14 min. Finally, buffer B was increased to 95% over 4 min and the analytical column was washed at 95% for 10 min.

Eluting peptides were on-line transferred to the mass spectrometer and ionized by nano-electrospray ionization operated at a constant spray voltage of 2.4 kV. Precursor MS were recorded at a resolution of 60,000 with an AGC target of 300%, a maximum injection time of 25 ms, and a scan range of 350–1750 $m/z$. The top 20 precursor ions of each MS scan with a minimum intensity of 2e4 were selected for fragmentation and subsequent MS/MS measurement. Fragment MS/MS scans were recorded at a resolution of 15,000, an AGC target of 100%, and a maximum injection time of 22 ms. The quadrupole isolation window was set to 1.4 $m/z$ and ions were fragmented with an HCD collision energy of 30%. Selected precursor ions were dynamically excluded from MS/MS measurement for 30 s. The mass spectrometry proteomics data have been deposited to the ProteomeXchange Consortium (http://proteomecentral.proteomexchange.org) via the PRIDE partner repository with the dataset identifier PXD052113.

## Statistical analysis of mass spec proteomics data

Mass spectrometric raw data were processed with the MaxQuant software suite (version 2.0.3.0, https://rdcu.be/dhiOz) and its implemented Andromeda scoring algorithm (Cox et al, 2011). The SwissProt human proteome was used as the sequence database for peptide identification (downloaded 2023-02-15). Lys-8 and Arg-10 were selected as heavy labels with a maximum of five modifications (fixed cysteine carbamidomethylation, variable methionine oxidation, variable N-terminal acetylation) per peptide. Trypsin/P was selected as the digesting enzyme with a maximum of two missed cleavages per peptide and a minimum length of seven amino acids. Match between runs was activated with the default options. Identified precursor peptides and protein groups were filtered at an FDR < 1% using the unique + razor strategy to assign protein group intensities and quantification.

Statistical analysis of the data was performed with the Perseus software suite (v. 1.6.15, (Tyanova et al, 2016). SILAC H/L protein ratios were log2 transformed and sample-wise normalized by median subtraction. Protein groups were filtered for ratios identified in at least two replicates of one condition. Missing values were imputed by random value drawing from 1.8 standard deviations downshifted, and 0.3 standard deviations broad normal distributions. Significantly changed SILAC H/L ratios in the LPS-stimulated condition were identified by paired Welch's t-testing

with permutation-based FDR correction (significance cutoff: FDR < 0.1, S0 = 0.1). Systematically enriched or depleted gene ontology (GO) terms or UniProt keywords were identified by 1D annotation enrichment analyses (significance cutoff: Benjamini-Hochberg FDR > 0.1.

## Statistical analysis

Statistical analyses were performed with GraphPad Prism Version 9.0 f. Unless indicated otherwise, all graphs are built from pooled data from a minimum of two independent experiments (biological replicates) performed in triplicates (technical replicates). All statistical analyses were preceded by normality and lognormality tests, followed by the recommended parametric or nonparametric tests. For most experiments with several groups (StdMo, PdMo, PdMo + Ptls, and Plts, stimulated vs. unstimulated), P values were determined by two-way ANOVA with Tukey's or Sidak's or multiple comparison tests. No outliers were detected or removed from the analysis. Additional statistical details are given in the respective Fig. legends when appropriate.

## Data presentation

Unless indicated otherwise (in Fig. legends), all graphs are represented as Floating Bars (with mean and minimum to maximum values) and are built from pooled data from a minimum of two independent experiments (biological replicates) performed

**The paper explained**

**Problem**

Monocytes secrete highly inflammatory cytokines that are critical for a proper immune response but can also lead to excessive inflammation when dysregulated. Similarly, dysfunctions in monocyte activity are associated with immunoparalysis, a state of hypo-responsiveness that halts the immune system's ability to fend off invaders. How monocytes mediate these diverse functions remains a mystery. This study investigates the role of platelets in regulating monocyte inflammation, with a focus on human monocytes.

**Results**

We discovered that platelets are essential for the monocyte pro-inflammatory cytokine response. Platelet depletion in monocytes from healthy individuals and low blood platelet counts in patients with immune thrombocytopenia (ITP) caused monocyte immunoparalysis impairing their response to viral and bacterial ligands. Supplementation with fresh or healthy platelets reversed this condition and restored the monocyte's capacity to produce cytokines. Mechanistically, we found that pro-inflammatory signaling, including NF-κB and p38 MAPK, propagates from platelets to monocytes sustaining their inflammatory capacity. Platelet vesicles mediate this intercellular communication.

**Impact**

These findings highlight a novel intercellular communication mechanism by which platelets regulate monocyte function. Clinically, this suggests potential therapeutic strategies involving platelet supplementation to counteract monocyte immunoparalysis in conditions such as ITP and other inflammatory diseases. Understanding the platelet-monocyte interaction may improve treatments for immune dysregulation and associated diseases.

in triplicates (technical replicates) with monocytes or platelets from different donors. Each symbol represents the average from 3 technical replicates per donor or experiment in the case of cell lines. Symbols are coded (●, ▼, ■, ●, etc.) to indicate donors so readers can track the internal variability between different donors or experiments. Dots are semi-transparent, with darker symbols showing overlapping points.

## Data availability

The datasets produced in this study are available in the following databases: Source data: BioStudies (https://www.ebi.ac.uk/biostudies/studies/S-BSST1437; Accession Number: S-BSST1437). NanoString-Transcriptional profiling of LPS-primed and inflammasome-activated untouched, or platelet-depleted human CD14+ monocytes. BioStudies (https://www.ebi.ac.uk/biostudies/arrayexpress/studies/E-MTAB-14072; Accession Number: E-MTAB-14072). RNA-Seq Transcriptomic Profiling of Ly6G− IA/IE − CD45 + CD11b + CD115+ Mouse Monocytes: Impact of In Vivo Platelet Depletion and LPS Challenge. BioStudies (https://www.ebi.ac.uk/biostudies/arrayexpress/studies/E-MTAB-14126?key=b14d057c-eff2-478e-a070-1a0a8e1e209a; Accession Number: E-MTAB-14126). The mass spectrometry proteomics data have been deposited to the ProteomeXchange Consortium via the PRIDE partner repository with the dataset identifier PXD052113.

The source data of this paper are collected in the following database record: biostudies:S-SCDT-10_1038-S44321-024-00093-3.

## Peer review information

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

## Acknowledgements

This study was funded by the European Research Council (EC | H2020 | H2020 Priority Excellent Science | H2020 European Research Council (ERC) PLAT-IL-1, 714175). Bernardo S Franklin is further supported by the HORIZON Lump Sum Grant (ERC PoC UNBIAS, 101123144), (BEATSep, 101137484), as well as

Germany's Excellence Strategy (EXC 2151 – 390873048, SFB TRR259) from the Deutsche Forschungsgemeinschaft (DFG, German Research Foundation). We also thank Marco A Ataide, Florian I Schmidt, and Dagmar Wachten for scientific discussions and input. We thank Cornelia Rohland and Maximilian Rothe for technical help and organization and maintenance of mouse lines, respectively. We thank Jonathan Schmid-Burgk for their advice in generating THP-1 KO cells using CRISPR-Cas9. We thank Dr. Savithri Rangarajan and Dr. Rik de Wijn (PamGene, Diagnostic Assay Services, 's-Hertogenbosch, The Netherlands) for their assistance in the peptide quality control and bioinformatics analysis of the Protein Kinase Assay. We thank Andreas Dolf, Peter Wurst, Gabor Horvath as well as the Microscopy and Flow Cytometry Core Facilities of the Medical Faculty at the University of Bonn for providing help, services, and devices funded by the Deutsche Forschungsgemeinschaft (DFG, German Research Foundation, project number 388158066, 216372545 and 388159768). Finally, we thank Andrea Schlichting and Leonie Verwohlt for administrative support.

## Author contributions

**Ibrahim Hawwari**: Data curation; Formal analysis; Validation; Investigation; Visualization; Methodology; Writing—review and editing. **Lukas Rossnagel**: Data curation; Investigation; Methodology; Writing—review and editing. **Nathalia Rosero**: Investigation; Methodology. **Salie Maasewerd**: Investigation; Performed the imaging and quantification of ASC specks and helped in the generation of THP1 knockout cell lines. **Matilde B Vasconcelos**: Investigation; Assisted with monocyte and platelet vesicle isolation experiments Main Fig. 8B,D. **Marius Jentzsch**: Resources; Investigation; Developed the protocols for CRISPR-Cas9 gene editing of THP1 monocytes and helped to generate the THP1 lines used in our study. **Agnieszka Demczuk**: Investigation; Produced the data presented in Fig. 1F,G (AIM2 and NLRC4 inflammasome activation). **Lino L Teichmann**: Resources; Coordinated ITP patient recruitment and sample collection. **Lisa Meffert**: Resources; Methodology; Organized the ITP patient recruitment and sample collection. **Damien Bertheloot**: Helped by giving valuable input on experimental details, discussions, and intellectual contributions. **Lucas S Ribeiro**: Methodology; Assisted with the animal experiments involving platelet depletion, injections and blood collection for Monocyte isolation (Fig. 4). **Sebastian Kallabis**: Data curation; Performed the mass spec proteomics and the analysis. **Felix Meissner**: Resources; Help in the desing and analysis of the Mass Spec Proteomics. **Moshe Arditi**: Resources; Investigation. **Asli E Atici**: Investigation. **Magali Noval Rivas**: Resources; Investigation. **Bernardo S Franklin**: Conceptualization; Resources; Data curation; Formal analysis; Supervision; Funding acquisition; Validation; Investigation; Visualization; Methodology; Writing—original draft; Project administration; Writing—review and editing.

Source data underlying figure panels in this paper may have individual authorship assigned. Where available, figure panel/source data authorship is listed in the following database record: biostudies:S-SCDT-10_1038-S44321-024-00093-3.

## Funding

## Disclosure and competing interests statement

The authors declare no competing interests.

# Expanded View Figures

**Figure EV1.   Platelets are critical checkpoints for the cytokine production of primary human monocytes.**

(A) Representative flow cytometry analysis of human PBMCs, and isolated untouched (StdMo), or platelet-depleted (PdMo) primary CD14$^+$ monocytes stained with CD14 (monocyte marker) and CD41a (platelet marker), or corresponding isotype controls. Gating strategy to identify platelet-free monocytes (CD14$^+$ CD41a$^-$, Monos), monocyte-platelet aggregates (CD14$^+$ CD41a$^+$, MPAs), and free platelets (CD14$^-$ CD41a$^+$, Plts). Plots on the right shows immunophenotyping of PBMCs, StdMo and PdMo based on the surface expression of CD14 and CD16. Data is from one representative of several independent experiments. (B) Comparative quantification of Monos, MPAs and Plts populations based on their frequency determined by flow cytometry in monocyte isolations ($n = 50$) as shown in A. Error bars show the SD of each group (Monos, MPAs and Plts). (C) Quantification of platelet and monocyte counts through a CASY cell counter and analyzer. (D) Confocal imaging of StdMo showing CD14$^+$ monocytes (blue) and CD61$^+$ platelets (red), and the formation of MPAs. Scale bars: 10 µm. (E) IL-1β and IL-6 concentrations in cell-free supernatants (CFS, left) or whole cell lysates (WCL, right) from untouched (StdMo), platelet-depleted (PdMo), or PdMo that were supplemented with autologous platelets (100:1 platelet:monocyte ratio). Cytokine levels secreted by platelets alone (Plts) were measured as control. Cells were stimulated with LPS (2 ng ml$^{-1}$ for 3 h) followed by activation with nigericin (10 µM for 1.5 h, for IL-1β), or directly with LPS (2 ng ml$^{-1}$ for 4.5 h, for IL-6). Floating bars display the max/min values with indications of the mean (white bands). Each symbol represents one independent experiment/blood donor. (F–H) Flow Cytometry assessment of surface expression of CD14 and TLR4 in StdMo, and PdMo, showing (F) gating strategy, (G) kinetics and (H) comparative quantification of CD14 and TLR4 expression over time (0.5, 1, 2, or 3 h) upon LPS (2 ng ml$^{-1}$) stimulation. Data display the percentage of TLR4$^+$ or CD14$^+$ StdMo or PdMo in each group. Each symbol represents one of $n = 5$ different donors. (I) Cell viability assessed every 15 min in StdMo, PdMo, PdMo + Plts, and Plts treated as in (E). (J) IL-1β concentrations in CFS of LPS-primed StdMo, PdMo, or PdMo + Ptls that were stimulated with Pam3CSK4 (1 µg ml$^{-1}$), Resiquimod R848 (10 µM), or LPS (2 ng ml$^{-1}$) for 4.5 h and left untreated, or further activated with nigericin (10 µM, for 90 min). Floating bars display the max/min values with indications of the mean (white bands). Each symbol represents one independent experiment/blood donor. *P* values are from ANOVA multiple comparison test. (K) Radar plots displaying all 45 cytokines, chemokines and growth factors measured by Cytokine Luminex in the CFS of StdMo, PdMo, or PdMo + Ptls and platelets stimulated with Pam3CSK4 (1 µg ml$^{-1}$) or Resiquimod R848 (10 µM). Protein concentrations are represented by the spread from inner (0 ng ml$^{-1}$) to outer circles (> 15 ng ml$^{-1}$). Colours represent stimuli (Unstim, dark gray; Pam3CSK4, blue; and R848, red). Each symbol represents one independent experiment/blood donor ($n = 4$ donors).

▶

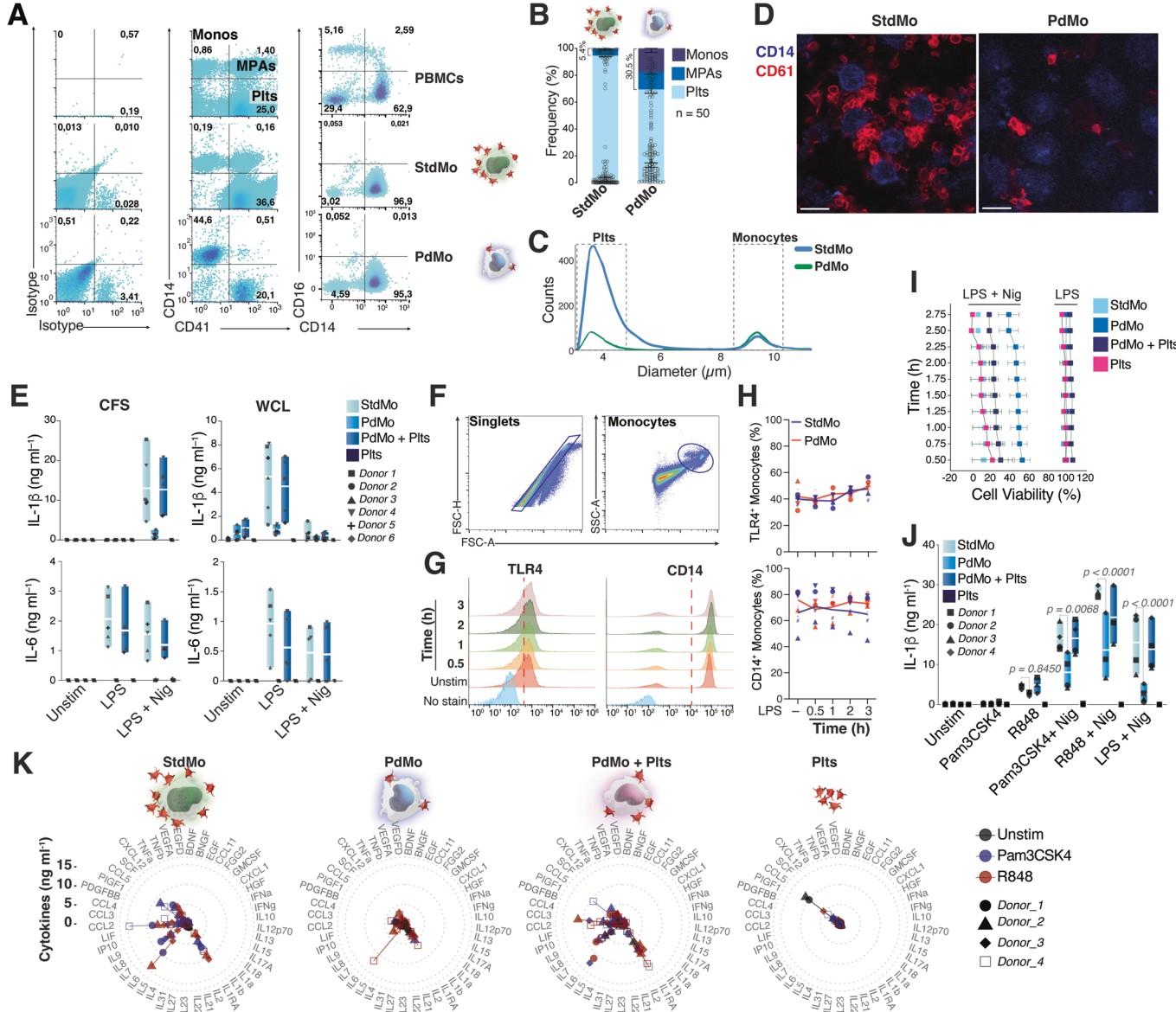

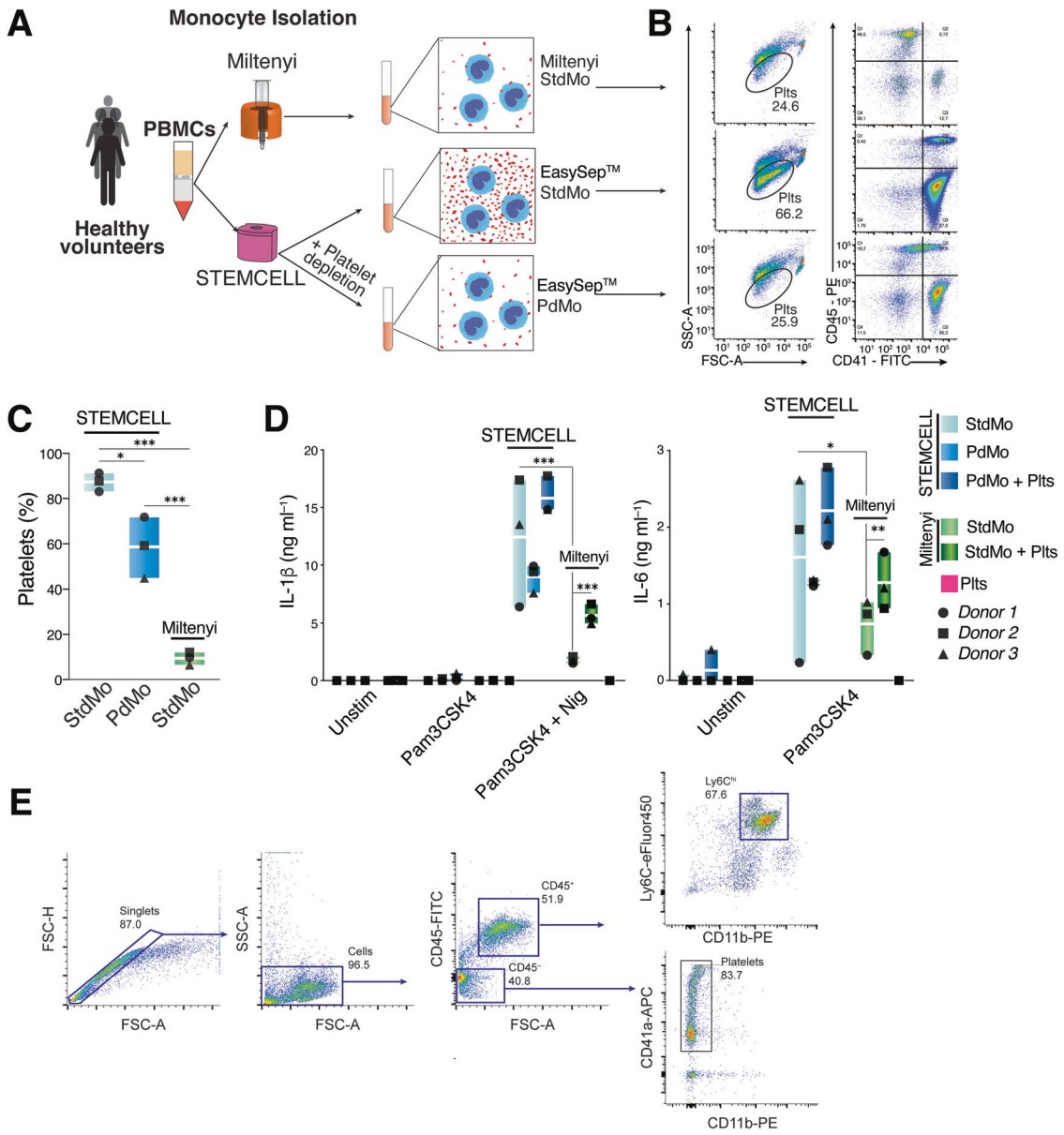

**Figure EV2. Impact of platelet removal from primary human monocytes.**

(A) Schematic presentation of the immune-magnetic isolation of primary human monocytes from peripheral blood comparing the Miltenyi vs. the EasySep™ monocyte isolation kits. The EasySep™ kit was further supplemented with (PdMo) or without (StdMo) a platelet-depletion cocktail. (B) Representative flow cytometry analysis of the human primary monocyte populations isolated as in (A). Gating shows the populations of platelet-free monocytes (CD14+ CD41a−) and platelets (CD41a+), or corresponding isotype controls. Data is from one representative of 3 independent experiments. (C) Frequencies of free platelets in populations of monocytes isolated as in (A), comparing StdMo (light blue bars), PdMo (blue bars) isolated with the EasySep™ kit, or StdMo isolated with the Miltenyi kit (green bars). Floating bars display the max/min values with indications of the mean (white bands). Each symbol represents one independent experiment/blood donor (n = 3). Each symbol represents one independent experiment/blood donor (n = 3). P values are from two-way ANOVA with Tukey's multiple comparison test with 95% confidence interval, and are indicated as * (<0.05), ** (<0.01), and *** (<0.001). (D) Concentrations of IL-1β, and IL-6 released by untouched (StdMo), platelet-depleted (PdMo), or PdMo that were supplemented with autologous platelets (PdMo + Plts, 100:1 platelet:monocyte ratio), using the EasySep™ kit (blue bars), or untouched isolated with the Miltenyi kit cultured alone (StdMo, green bars) or co-cultured with platelets (StdMo + Pts). Cytokine levels secreted by platelets alone (Plts) were measured as control. Cells were stimulated with LPS (2 ng ml⁻¹ for 4.5 h, for IL-6) or with LPS (3 h) followed by activation with nigericin (10 µM for 1.5 h, for IL-1β). Floating bars display the max/min values with indications of the mean (white bands). Each symbol represents one independent experiment/blood donor (n = 3). P values are from two-way ANOVA with Tukey's multiple comparison test with 95% confidence interval, and are indicated as * (<0.05), ** (<0.01), and *** (<0.001). (E) Representative Flow Cytometry and gating strategy to assess the purity of murine monocytes isolated from mouse blood. Cells were gated based on surface expression of CD45, Ly6C, and CD41a. Images are from one representative of independent experiments with n = 4 mice.

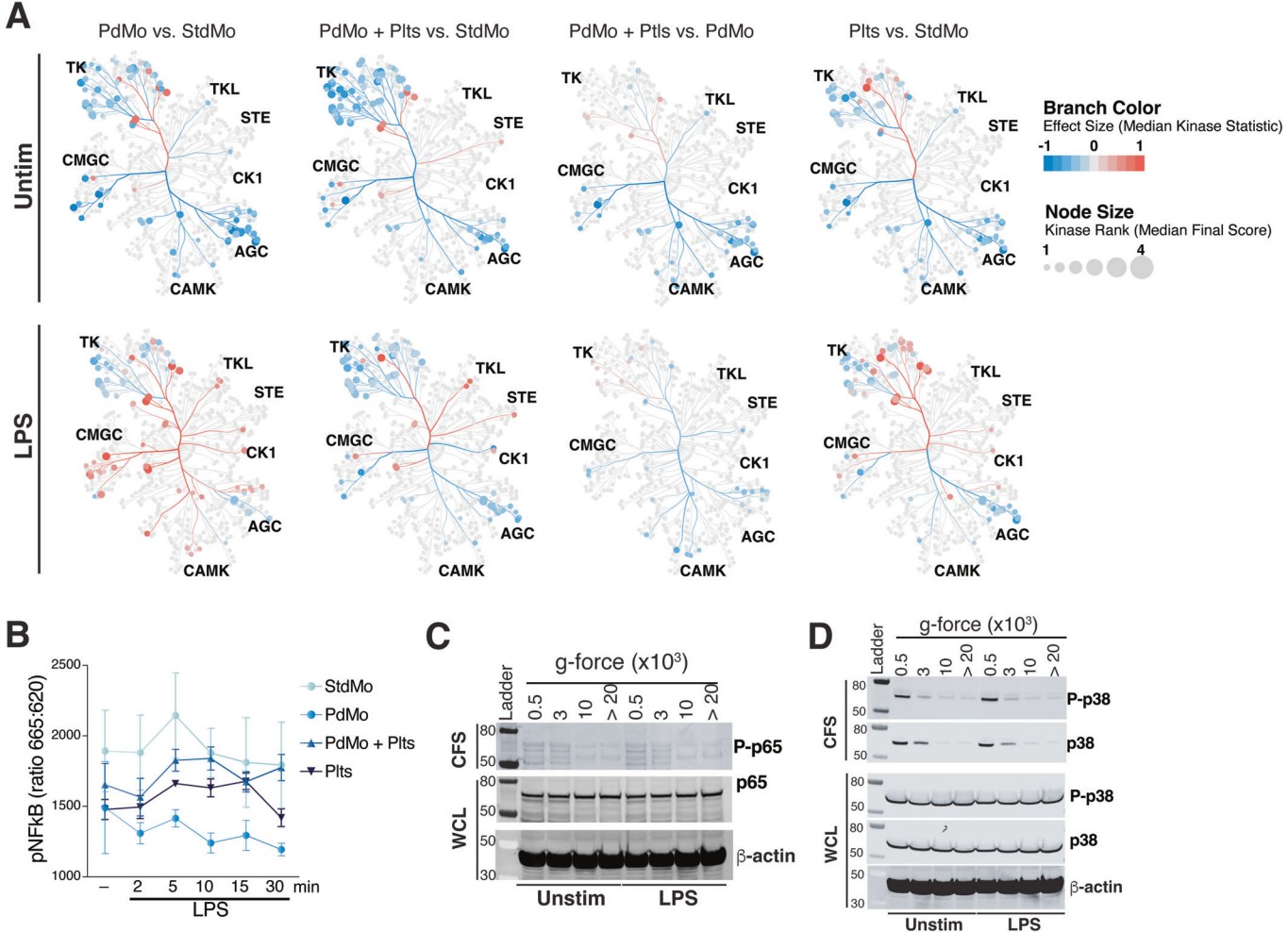

**Figure EV3. Intrinsic kinase activity in platelets and their effects on human monocytes.**

(A) Coral trees displaying the activity of Protein Tyrosine and Serine/Threonine kinases in unstimulated (Unstim) or LPS-treated (LPS; 2 ng ml⁻¹) primary human monocytes comparing the effects of platelet-depletion/supplementation, measured with a PamStation12 (PamGene). (B) NF-κB activity assay in CFS of unstimulated (Unstim), or LPS-stimulated (2 ng ml⁻¹, for the indicated times) primary human monocytes (StdMo, PdMo, PdMo + Plts, and Plts alone. Cells were lysed/incubated with lysis buffer, supplied Cisbio, and used to assess phosphorylated NFkB by HTRF (Cisbio). Error bars display the SD ($n = 3$). (C, D) Immunoblot of RelA (p65 and p-p65) (C) or MAPK (p-38 and P-p38) (D) in resting (Unstim) or LPS-activated (LPS; 2 ng ml⁻¹) human platelets. Platelets were submitted to centrifugation at 500, 3000, 10,000, or 20,000 × $g$ and the levels of proteins were assessed in the pellets (WCL) or supernatants (CSF) after centrifugation. Results are representative of two independent experiments.

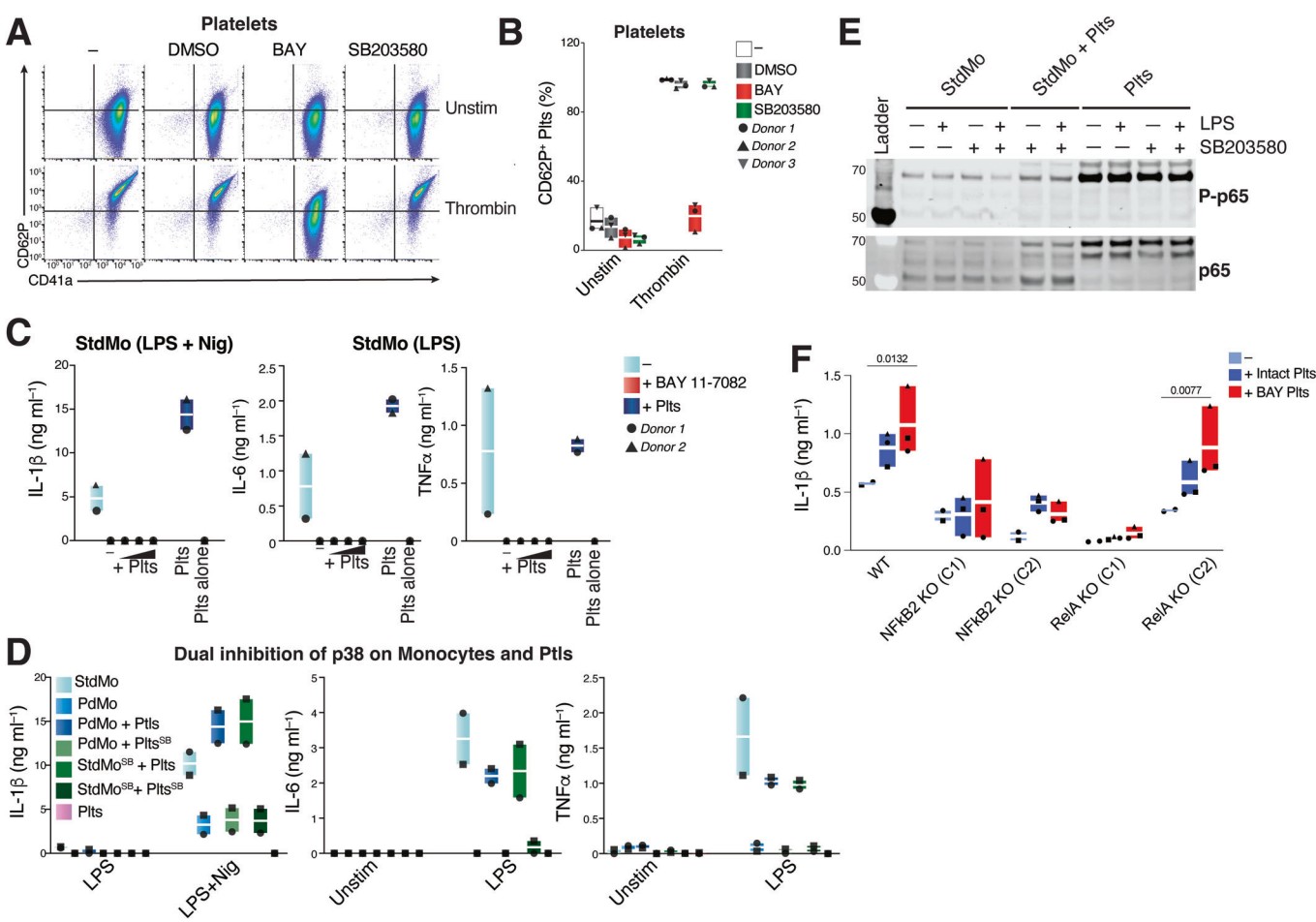

**Figure EV4. Dual p38 and NF-kB inhibition on platelets and monocytes.**

(A, B) Representative flow cytometry assessment and (B) quantification (%, $n = 3$) of CD41a and P-selectin (CD62P) expression on human platelets that were pre-treated with BAY (50 µM), or SB203580 (20 µM) 20 min before being activated with Thrombin (1 U ml$^{-1}$) for 30 min. (C) IL-1β, TNFα, and IL-6 concentrations in the CFS of StdMo that were treated with BAY before been added with increasing ratios of freshly isolated platelets (1:5, 1:50, and 1:100; StdMo:Plts). Cells were stimulated with LPS or LPS and nigericin (LPS + Nig). Floating bars display max/min values with indication to the mean (white bands). Each symbol represents one donor. (D) Concentrations of IL-1β, IL-6, and TNFα in CFS of StdMo, PdMo, or PdMo that were supplemented with platelets pre-treated with 20 µM SB203580 (Plts + SB), or left untreated (+ Plts), or in StdMo pre-treated with SB203580 (StdMo + SB) that were supplemented with intact platelets (StdMo + SB + Plts) or with SB203580-treated platelets (StdMo + SB + Plts + SB). Co-cultures were stimulated with LPS (2 ng ml$^{-1}$) followed by nigericin stimulation (10 µM) ($n = 2$). (E) Immunoblotting for phospho-p65 and total p65 (RelA) on unstimulated and LPS-stimulated StdMo primary human monocytes treated with 20 µM of SB203580. (F) IL-1β concentrations in CFS of stimulated RelA$^{-/-}$, NFkB2$^{-/-}$, or the parental WT THP-1 monocytes supplemented with platelets pre-treated with BAY 11-7082 (50 µM, 30 min) before addition to the clones. Cells were stimulated with Pam3CysK4 (1 µg ml$^{-1}$) or Pam3CysK4 and nigericin (10 µM) (Pam3 + Nig). All Graphs with floating bars depict maximum/minimum values relative to the mean (white bands). P values were calculated using two-way ANOVA, Tukey's multiple comparison test, and are indicated in the figure. Each symbol represents one independent experiment or blood donor ($n = 3$).

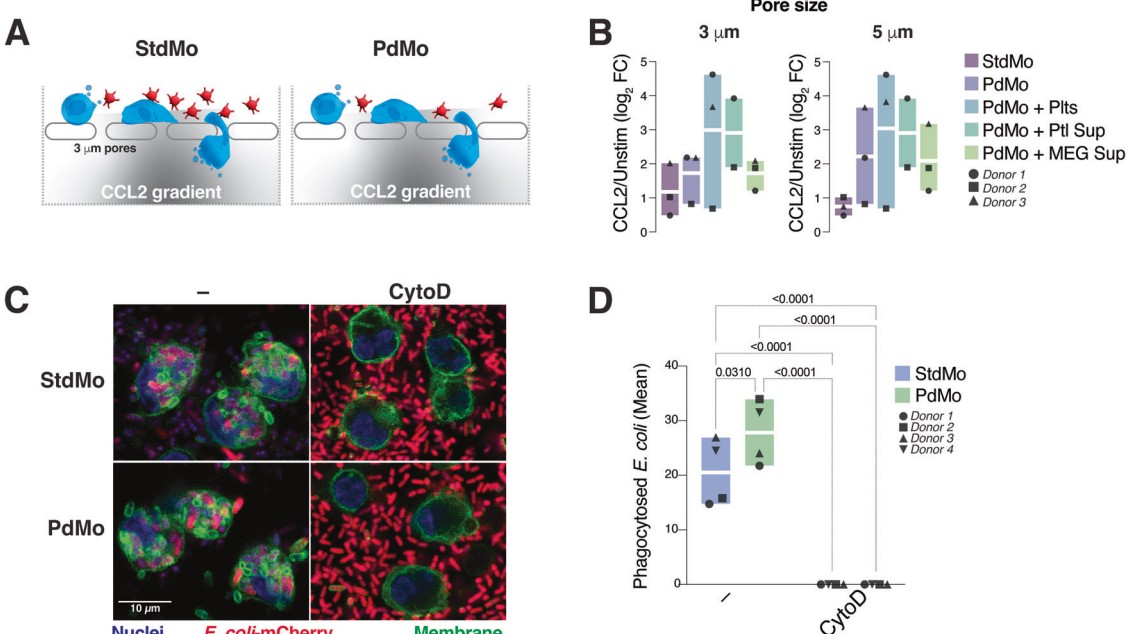

**Figure EV5.  Effects of platelets on monocyte trans-migration and phagocytosis.**

(A, B) Primary human monocytes (StdMo, or PdMo) were supplemented with platelets, platelet (Plt Sups) or Mk releasates (MK Sups). Cells were then seeded on the upper chamber of a trans well plate with either 3 μm or 5 μm pore sizes and incubated with CCL2 (40 ng ml⁻¹) or left untreated for 4 h. Monocyte trans-migration was measured by confocal imaging and quantification of cells that migrated to the bottom wells. 16 pictures per well were taken and the nuclei (stained with DRAQ5) were counted via Cell Profiler. Log2 fold change was calculated and the CCL2 conditions were normalized to their respective unstimulated condition ($n = 3$). (C, D) Confocal Imaging (C) and quantification (D) of StdMo and PdMo exposed to an mCherry fluorescent *E. coli* strain (1:25 MOI, for 2 h). Monocytes were either left untreated (−) or pre-treaded with the phagocytosis inhibitor cycochalasin D (CytoD) and exposed to 100:1 MOI of *E. coli*. Cells were stained with DRAQ5 (blue, nuclei) and WGA-AF488 (green, membranes). Images were acquired by confocal and automated widefield microscopy. Images (four per condition) were analyzed by counting the *E. coli* inside five monocytes per image, by two independent experimenters. Mean numbers of *E. coli* per cell were plotted. Graphs with floating bars depict maximum/minimum values relative to the mean (white bands). *P* values were calculated using two-way ANOVA, Tukey's multiple comparison test. Each symbol represents one donor ($n = 4$).

