## [Peer Review File · EMBO Molecular Medicine]

Platelet transcription factors license the pro-inflammatory cytokine response of human monocytes.

Ibrahim Hawwari, Lukas Roßnagel, Nathalia Rosero, Agnieszka Demczuk, Salie Maasewerd, Marius Jentsch, Matilde Vasconcelos, Lino Teichmann, Lisa Meffert, Lucas Ribeiro, Sebastian Kallabis, Felix Meißner, Damien Bertheloot, Asli Atici, Magali Noval-Rivas, Moshe Arditi, and Bernardo Franklin

Corresponding author(s): Bernardo Franklin (franklin@uni-bonn.de) , Ibrahim Hawwari (ibrahim.hawwari@uni-bonn.de)

Review Timeline:

Submission Date:	14th Nov 23
Editorial Decision:	20th Dec 23
Revision Received:	30th Apr 24
Editorial Decision:	28th May 24
Revision Received:	5th Jun 24
Accepted:	6th Jun 24

Editor: Zeljko Durdevic

Transaction Report:

20th Dec 2023

Dear Prof. Franklin,

Thank you for the submission of your manuscript to EMBO Molecular Medicine, and please accept my apologies for the delay in getting back to you. We have now received feedback from two of the three reviewers who agreed to evaluate your manuscript. As the referee #3 will unfortunately not be able to return his/her report in a timely manner, and given that both reviewers provide very similar recommendations, we prefer to make a decision now in order to avoid further delay in the process.

As you will see from their reports pasted below, both referees recognize potential interest of the study but also raise important concerns that should be addressed in a major revision. If you would like to discuss further the points raised by the referees, I am available to do so via email or video. Let me know if you are interested in this option.

We would welcome the submission of a revised version within three months for further consideration. Please let us know if you require longer to complete the revision.

I look forward to receiving your revised manuscript.

Yours sincerely,

Zeljko Durdevic

We require:

- 1) A .docx formatted version of the manuscript text (including legends for main figures, EV figures and tables). Please make sure that the changes are highlighted to be clearly visible.
- 2) Individual production quality figure files as .eps, .tif, .jpg (one file per figure). For guidance, download the 'Figure Guide PDF': (<https://www.embopress.org/page/journal/17574684/authorguide#figureformat>).
- 3) A .docx formatted letter INCLUDING the reviewers' reports and your detailed point-by-point responses to their comments. As part of the EMBO Press transparent editorial process, the point-by-point response is part of the Review Process File (RPF), which will be published alongside your paper.
- 4) A complete author checklist, which you can download from our author guidelines (<https://www.embopress.org/page/journal/17574684/authorguide#submissionofrevisions>). Please insert information in the checklist that is also reflected in the manuscript. The completed author checklist will also be part of the RPF.
- 5) Please note that all corresponding authors are required to supply an ORCID ID for their name upon submission of a revised

manuscript.

6) It is mandatory to include a 'Data Availability' section after the Materials and Methods. Before submitting your revision, primary datasets produced in this study need to be deposited in an appropriate public database, and the accession numbers and database listed under 'Data Availability'. Please remember to provide a reviewer password if the datasets are not yet public (see <https://www.embopress.org/page/journal/17574684/authorguide#dataavailability>).

13) Author contributions: You will be asked to provide CRediT (Contributor Role Taxonomy) terms in the submission system. These replace a narrative author contribution section in the manuscript.

14) A Conflict of Interest statement should be provided in the main text.

15) Every published paper now includes a 'Synopsis' to further enhance discoverability. Synopses are displayed on the journal webpage and are freely accessible to all readers. They include a short stand first (maximum of 300 characters, including space) as well as 2-5 one-sentence bullet points that summarize the paper. Please write the bullet points to summarize the key NEW findings. They should be designed to be complementary to the abstract - i.e. not repeat the same text. We encourage inclusion of key acronyms and quantitative information (maximum of 30 words / bullet point). Please use the passive voice. Please attach these in a separate file or send them by email, we will incorporate them accordingly.

Please also suggest a striking image or visual abstract to illustrate your article as a PNG file 550 px wide x 300-800 px high.

***** Reviewer's comments *****

Referee #1 (Remarks for Author):

Summary of the key findings

In the manuscript by Hawwari et al, the authors propose that in human blood, platelets boost pro-inflammatory responses of monocytes to pathogenic challenge downstream of TLR and inflammasome stimulation.

Importance of the key findings

These results are highly relevant for the human population as authors demonstrate clearly that in people (and mice) with low platelet counts (immune thrombocytopenia) or upon platelet depletion, there is hypo-responsiveness of monocytes, characterised by reduced pro-inflammatory gene expression and weakened cytokine responses to immune challenge. This hyporesponsive state can be reversed very simply, by adding fresh platelets. The functional characterisation and results to support this part of the model are clear and compelling.

The mechanism proposed is quite unusual. Authors propose that, in a process of intercellular transfer, platelets transfer, via extracellular vesicles, their own pro-inflammatory transcription factors (e.g NFkB2, p38) to boost NFkB and p38-driven inflammatory responses in monocytes, even when monocytes are inhibited or made deficient in NFkB or p38 signalling. The transfer is shown using proteomics but the final mechanism of trans-signalling is still quite challenging to prove.

Open questions:

1. In Fig 1A-B inflammatory responses are compared by ELISA and western blot between monocyte pure cultures (platelet depleted) versus monocytes+platelets mixed cultures. To provide quantitative per-cell measurement of inflammatory signalling in monocytes, can pro-IL1b levels be measured using flow cytometry on gated CD14+ monocytes within these cultures, over a range of LPS priming concentrations?
2. To uncouple transcription-dependent re-programming of monocytes (model proposed here) from general signalling unresponsiveness, can ASC speck formation downstream of inflammasome activation be quantified per cell basis, by comparing transcription/NFkB-dependent inflammasome activation protocol (long priming) to transcription-independent inflammasome activation protocol (30 min priming)? How would transcription-independent IL-18 secretion look in monocyte pure cultures (platelet depleted) versus monocytes+platelets mixed cultures?
3. To show that LPS sensing is intact in platelet-free monocytes, can levels of CD14 and CD11b in these cultures be shown, and the rate of LPS-induced TLR4 internalisation?
4. The model that platelets are not phagocytosed by monocytes but rather that platelet-derived vesicles are engulfed to reprogram monocytes needs to be tested a bit more rigorously. Can phagocytosis of platelet-derived vesicles before and after LPS be quantified? Can it be tested over a time course using flow cytometry e.g. using platelets labelled with pH-sensitive dye so that the signal is only detected when platelets or their vesicles reach a low pH phagosome compartment? In Fig 5C can platelet supernatant (which restores monocyte responsiveness) be further fractionated to remove vesicles, or can vesicles be purified? Do vesicles need to be coming from platelets or they can come from any other cell type? Does LPS treatment of co-cultures increase vesicle release by platelets? How do transcription factors gain access to the cytosol or the nucleus of monocytes after vesicles are engulfed? If monocytes are fed vesicles expressing tagged versions of transcription factors can these tagged versions be shown inside the nucleus of LPS-stimulated monocytes to prove that they drive de-novo transcription?

Maybe authors can use their nice heavy amino acid SILAC approach described in Fig 7 to address the last question.

5. It was clear that the addition of platelets increases NFkB reporter activity in THP reporter monocytes stimulated with Pam3Cysk4. Would platelet supernatant transfer (+/- vesicles) or transwell culture do the same when transferred into NFkB KO THP cells?
6. How long do platelets need to be present in the culture to functionally reprogram monocytes?
7. Is monocyte metabolism affected by platelet presence?
8. Direct ex vivo analysis of monocytes from platelet-depleted mice suggests their transcriptional reprogramming towards cell cycle and division at the expense of immune defence programs. Can authors comment on this? Do blood monocytes cycle normally, or does platelet depletion cause some kind of differentiation state change?
9. Inhibitor pre-treatment of platelets before addition to monocytes does not prove that the transcription factor transfer and trans-signalling in monocytes are the mechanism, as there could be NFkB and/or p38-induced factors(s) inside platelets responsible for monocyte reactivation when transferred via vesicles.
10. Why do platelet-free monocytes have increased Jak signalling? Can authors comment on this? Is Jak activation key to monocyte reprogramming or unresponsiveness?

Minor points

11. Fig1B should include a loading control for the whole cell lysate
12. Fig S1 compared responses of monocytes isolated by positive (Miltenyi) and negative (StemCell) selection protocols from two manufacturers. StemCell also provides positive selection protocol/beads, so ideally, those should be included in this figure for comparison of monocytes obtained by positive versus negative selection protocols by the same manufacturer.

Referee #2 (Comments on Novelty/Model System for Author):

As indicated in my comments to the authors, the paper presents the first direct demonstration of transfer of functional transcriptional factors from platelets into monocytes, and illustrates that they are functionally competent to elicit measurable immune mediators. The study is all done with human samples which, while still pre-clinical, has the potential of translation to the clinic in the future. The limitations and drawbacks of some experiments are detailed in the response to the authors.

Referee #2 (Remarks for Author):

The paper by Hawwari, Franklin et al. introduces the notion that transcription factors can be exogenously delivered to monocytes to enable effective responses to microbial products. This is an exciting concept that, while not new, is robustly demonstrated in human monocytes in vitro using a plethora of elegant and diverse strategies. An obvious drawback from the studies is the in vitro nature of all the assays used here, while the in vivo confirmation in a mouse model is non convincing given the challenge of removing platelets from a living organism in a controlled manner. Nonetheless, the value of the paper is the demonstration that immune cells can be licensed in trans and acquire active mechanisms of activation through interactions with other cells. The limitations of the study need to be precisely described in the text to avoid confusion.

Below are some comments and suggestions to improve this otherwise complete study:

- In addition to the analysis of monocytes isolated using the different platforms, I suggest assessing the differential response of monocytes alone vs MPA isolated from blood in cytokine production assays. The idea is to establish whether specific populations of monocytes already existing in the human circulation (not only in the test tube) are more prone to immune activity.
- It would be interesting to test other features of monocytes in the presence or absence of platelets, including migration (chemotaxis), phagocytosis (beads) and microbicidal activity. The point would be to assess not only specificity or not for cytokine production but more generally the extent to which this platelet-monocyte axis influences immune function.
- Fig.4: The data on thrombocytopenic mice are difficult to interpret and at best the data suggest modest changes when compared with the dramatic effect seen in vivo with human monocytes. This is interesting but of some concern for the extension of the proposed model from human to mice, while at the same time highlighting that experimental depletion in vivo or other systemic effects induced by platelet depletion are not simple to explain. While this validates the in vitro strategy for human cells, I am missing experiments in murine monocytes showing that at least ex vivo the trend is the same. In sum, either this data with mouse cells is added or it would be advisable to remove this section altogether.
- Fig.5A, it is impossible to evaluate from these images where platelets are or not engulfed by monocytes. Images should be clearer and this quantified in some form if the authors wish to make this point.
- EV5 and other figures, clearly illustrate how variable the response of human monocytes is in response to stimulation, which raises the question of whether the low number of patients used in these experiments yields a reliable conclusion. For example,

the inhibition by blocking Siglec7 in DVFH is clear yet the authors conclude lack of an effect based on poorly reliable statistics. This can be also easily taken by intended bias towards a model favored by the reviewers. This is a nice and meritorious study which can easily lose credibility by this type of loose analyses and interpretation.

- A relevant control for the experiments in Fig8G is whether the NFkB inhibitors will fully block the increase in IL1b secretion in KO monocytes supplemented with platelets. This is important to validate that no other alternative pathways are stimulated by platelet-derived factors.

- In the discussion, I miss ideas from the authors on how and where the "platelet-licensed" monocytes function, given that interactions must take place in the blood stream but antimicrobial activities, for example, take place mostly in the tissue parenchyma. I also miss discussion on the differential activity of MPA vs. free monocytes, which would suggest that priming for a subset of monocytes occurs in blood but the duration of such priming was not tested here.

Minor:

- Fig.4B: platelet counts should be given as absolute counts, not percentages.

- Line 264: the statement that "Next, we blocked integrins..." is strange as CD11b is an integrin.

- some explanation and interpretation of the nice kinome dataset in Fig.6A-B would greatly enhance the value and clarity of this figure and results. For example, it would seem that PdMo have massive increase in the activity of these kinases. The interpretation of Fig.6C in contrast is more intuitive and seems to go in the opposite direction as what is shown in Fig.6B.

- Fig8B, the panels for IL6 and TNFa secretion are identical; I wonder if the panel was duplicated by mistake?

30th of April, 2024

Manuscript: EMM-2023-18988

Platelet transcription factors license the pro-inflammatory cytokine response of human monocytes.

Point-by-point response to the Reviewer's comments:

Dear Dr. Durdevic, dear Reviewers,

Firstly, we would like to thank you for considering our manuscript entitled "**Platelet-derived transcription factors license human monocyte inflammation.**" We also thank the reviewers for their time and efforts in judging our manuscript.

We are delighted to share that we have performed experiments to address all the questions raised, even when experiments were not explicitly requested. We are confident that the extensive new data sets and answers address the points raised by the reviewers and substantially improve the manuscript.

In summary, our revised manuscript now includes:

- 1) Immunofluorescence and confocal assessment and quantification of ASC specks, demonstrating:
 - Reduction in ASC speck formation in primary human monocytes depleted of platelets (PdMo) and subsequent restoration of ASC speck formation upon platelet supplementation.
- 2) Fractionation of platelet releasates to enrich or deplete vesicles and correspondent gain-vs. loss-of-function assays, demonstrating that:
 - Purified platelet vesicles are as efficient as platelet releasates (Plt Sups) in restoring cytokine functions of PdMos;
 - Depletion of vesicles from platelet releasates extinguishes their effects on reconstituting faulty cytokine secretion in human platelet-depleted monocytes.
 - Discovery of NF- κ B2 enrichment in the vesicle fraction of platelet releasates and its corresponding absence in vesicle-free fractions.
 - Demonstration that platelet vesicles can reactivate genetic dysfunctional NF- κ B-signaling in NF- κ B2 or NF- κ B-p65 (RelA) deficient human monocytic cell lines (THP1).
- 3) Surface staining of TLR4 and CD14 in primary human monocytes in time course experiments comparing the kinetics of these LPS receptors in untouched (StdMo), PdMos, and PdMos replenished with platelets.
- 4) Demonstration that in mice, platelet depletion from blood monocytes also results in impaired cytokine responses upon inflammasome-activation *ex vivo* and that this dysfunction is ameliorated, akin to human monocytes, by platelet re-supplementation.
- 5) Additional experiments aiming to address specific reviewers' questions, such as phagocytosis and migration assays, metabolism, and others, are detailed in this *point-by-point response letter*.
- 6) A more precise and balanced description of our study's main findings and limitations.

Our new findings expand on our original manuscript demonstrating that platelet-derived pro-inflammatory pathways, i.e: NF- κ B and p38 license, the pro-inflammatory cytokine production of human monocytes. In the following pages, we provide a **point-by-point response** to the Reviewer's comments.

Reviewer 1:

These results are highly relevant for the human population as authors demonstrate clearly that in people (and mice) with low platelet counts (immune thrombocytopenia) or upon platelet depletion, there is hypo-responsiveness of monocytes, characterized by reduced pro-inflammatory gene expression and weakened cytokine responses to immune challenge. This hyporesponsive state can be reversed very simply by adding fresh platelets. The functional characterization and results to support this part of the model are clear and compelling.

Response: We thank this Reviewer for the kind words about our work and highlighting its potential implications. We did our best to fully address all the points raised by this Reviewer.

The mechanism proposed is quite unusual. Authors propose that, in a process of intercellular transfer, platelets transfer, via extracellular vesicles, their own pro-inflammatory transcription factors (e.g., NF- κ B, p38) to boost NF- κ B and p38-driven inflammatory responses in monocytes, even when monocytes are inhibited or made deficient in NF- κ B or p38 signaling. The transfer is shown using proteomics but the final mechanism of trans-signaling is still quite challenging to prove.

Response: Although the proposed mechanism for the transfer in our study is rather "unusual," as pointed out by this Reviewer, it is not entirely unexpected, also pointed out by **Reviewer 2**. A growing literature supports the abundance of NF- κ B and other transcription factors (TFs) in platelets (Lannan et al, 2015; Poli et al, 2021). Since platelets are non-nucleated cells, the possibility that they would "donate" their TFs to other cells has been speculated (Lannan et al., 2015), though never experimentally demonstrated. Our study provides the first experimental demonstration of this phenomenon, and most importantly, in human cells.

As pointed out by this Reviewer, the original results in the first version of our manuscript support the propagation of different pro-inflammatory pathways and signaling molecules from platelets to monocytes. This conclusion was corroborated by SILAC proteomics analyses of monocytes exposed to cell-free supernatants containing proteins labeled exclusively in megakaryocytes and the functional reactivation of key cytokine driving pathways, i.e., p38 MAPK, NF- κ B p65 (RelA), and NF- κ B2 in cells that were otherwise genetically deficient or pharmacologically inhibited.

Although the cytokines we measured are the outputs of NF- κ B and MAPK activation, it is indeed challenging to demonstrate that these platelet-derived molecules/TFs are functionally incorporated within the monocyte signaling pathways. Reviewer 2, for example, agreed with that.

Nevertheless, during the revision of our manuscript, we gained new insights that further support our original findings. In particular, we found that NF- κ B2 is enriched in platelet vesicles regardless of platelet stimulation. The pro-form of NF- κ B2 (p100) is present in platelet lysates, while the active subunit (p52) is predominantly enriched in vesicles, suggesting an intentional "packaging for delivery." Notably, the p52 subunit vanishes when vesicles are removed from the platelet releasates using size exclusion columns, concomitantly with the loss of platelet releasate's ability to restore cytokine production in PdMos. Vesicles are well-known mechanisms of inter-cellular communications, and these new findings strengthen the hypothesis that the platelet-derived cargo is purposely encapsulated in vesicles and delivered to monocytes.

Question 1: In Fig 1A-B, inflammatory responses are compared by ELISA and western blot between monocyte pure cultures (platelet depleted) versus monocytes+platelets mixed cultures. To provide quantitative per-cell measurement of inflammatory signaling in monocytes, can pro-

IL1b levels be measured using flow cytometry on gated CD14+ monocytes within these cultures, over a range of LPS priming concentrations?

Response: In our study, we employed HTRF, Western Blotting, Caspase-1 assays, and transcriptomics, and assessed intracellular and released IL-1 β protein levels. As requested, the assessment of IL-1 β expression on a single-cell level is intriguing. However, we have consistently demonstrated in all of our co-culture assays (and all figures of our manuscript) and our previous study (Rolfes et al., 2020) that platelets do not express IL-1 β , inflammasome components, and the cytokines we reported here, strongly indicating that they originate from monocytes. Hence, we are confident that the inflammasome-driven response in bulk is predominantly (not to say exclusively) produced by monocytes and is strikingly diminished in the absence of platelets. The Reviewer will also see from the new datasets in our revised manuscript that other priming-independent readouts (NLRC4, AIM2, IL-18, and ASC specks) support these conclusions.

Question 2: To uncouple transcription-dependent reprogramming of monocytes (model proposed here) from general signaling unresponsiveness, can ASC speck formation downstream of inflammasome activation be quantified per cell basis, by comparing transcription/NF- κ B-dependent inflammasome activation protocol (long priming) to transcription-independent inflammasome activation protocol (30 min priming)?

Response: The experiments presented in Fig. EV1E of our original manuscript demonstrated that intracellular levels of pro-IL-1 β , measured in whole cell lysates of human monocytes, as well as the activity of Caspase-1 (Main Fig. 1D), and the activity of two other inflammasomes (AIM2 and NLRC4, Main Fig. 1E-F) which are not influenced by TLR-priming follow the same trend as other inflammatory readouts. Meaning they are decreased in PdMos compared to StdMo, but reconstituted upon platelet replenishment. We believe these findings should suffice to uncouple transcription-dependent reprogramming of monocytes (model proposed here) from general signaling unresponsiveness.

Nevertheless, ASC speck formation is a reliable readout for inflammasome activation and expertise of our lab (Beilharz et al, 2016; Bertheloot et al, 2022; Franklin et al, 2014; Hoss et al, 2017; Stutz et al, 2013) and would indeed provide further evidence for a transcription-dependent reprogramming of monocytes. Therefore, we quantified and compared ASC speck formation in StdMo, PdMo, and PdMo supplemented with platelets. Supporting our previous observations, we observed a reduced rate of ASC speck formation in inflammasome-activated PdMos. Again, supplementation with platelets reconstituted ASC speck formation in PdMos.

New Fig. 1C-D - Immunostaining and confocal imaging of human ASC. Representative images (C), and quantification of ASC speck formation (D) in StdMo, PdMo, or PdMo + Plts that were primed with LPS (2 ng ml⁻¹, 3 h) and stimulated with Nigericin (10 μ M, 90 min) in the presence of VX-765 (50 μ M) to prevent ASC speck release. Images are representative of 3 independent experiments, pooled in (D). Isotype controls, unstimulated conditions are shown in Appendix Fig. S1A-B.

Confirming our previous findings (Rolfes et al, 2020), no ASC inflammasomes were visualized in human platelets.

This new dataset is included in the revised manuscript and presented in Main Fig. **1C-D**. The main figure will show the stimulated condition (LPS + Nig). Staining controls and LPS conditions are included in **Appendix Fig S1A-B**. These results are now described in the revised manuscript (Lines **113:127**).

We thank this Reviewer for this contribution to our study.

How would transcription-independent IL-18 secretion look in monocyte pure cultures (platelet depleted) versus monocytes+platelets mixed cultures?

Response: This Reviewer raises an important point, which we have already tried to address in the original manuscript. Our Luminex assay also included human IL-18 (see original **Fig 1G** and **EV1G**); however, the levels detected in culture supernatants of human monocytes were negligible and close to the detection limits (New Appendix Fig. **S1C**, below). Note the red dotted line at the mean IL-18 levels detected in the unstimulated StdMo. Still, akin to most other cytokines we measured, we found a similar trend for IL-18, which is reduced on primary human monocytes upon platelet depletion (PdMo) and reconstituted upon re-addition of platelets (PdMo + Plts) (New Appendix Fig. **S1C**, below).

As requested by this Reviewer, we analyzed IL-18 levels by WES capillary electrophoresis using a highly specific antibody against cleaved IL-18 (New Appendix Fig. **S1D**, below). We confirmed that IL-18 is matured upon stimulation of StdMo, reduced upon platelet depletion, and reconstituted upon platelet re-addition of PdMos. The WES confirms our previous observations that platelets do not contain IL-18 (Rolfes et al., 2020).

Together, this new dataset and the data already contained in our manuscript convey that platelets globally govern the monocyte pro-inflammatory cytokine responses. Platelets are required to license the function of monocytes on several levels, and their depletion leads to "immune paralysis," as proposed in our study.

The IL-18 data is mentioned in **lines 146-150** of our manuscript, and the experimental results are included in **Appendix Fig. S1C-D**.

Question 3: To show that LPS sensing is intact in platelet-free monocytes, can levels of CD14 and CD11b in these cultures be shown, and the rate of LPS-induced TLR4 internalization?

Response: This Reviewer raised a highly relevant point. To address this question, we performed FACS staining of CD14 and TLR4 in StdMo and PdMo upon stimulation with LPS for 0.5, 1, 2, and 3 h. We did not stain for CD11b as requested, as this integrin is a marker in murine monocytes. Additionally, our study shows that the role of CD11b (inhibition of MAC-1) specifically and the class of integrins generally (using RGDs) are not involved in the monocyte-platelet cross-talk in the human system.

Fig. EV1F-H. (F) Gating strategy and (G) schematic representation of the staining of surface CD14 and TLR4 in primary human monocytes left untreated (Unstim) or stimulated with LPS for the indicated periods. (H) Comparison of CD14 and TLR4 fluorescence in StdMo and PdMo stimulated as in G. Data is from 4 biological replicates.

Our new dataset examined the dynamic changes in TLR4 and CD14 protein expression over time after LPS stimulation (New Fig. EV1F-H). Notably, the expression level of each surface molecule is comparable between StdMo and PdMo. These findings highlight that platelet-depletion-induced monocyte immunoparalysis does not underlay an aberrant LPS sensing by TLR4/CD14 or their internalization.

This dataset was included in the revised version of our manuscript and is now represented in Fig. EV1G-I.

The textual description of this dataset is included in the revised manuscript (Lines 106-112).

Question 4: The model that platelets are not phagocytosed by monocytes but rather that platelet-derived vesicles are engulfed to reprogram monocytes needs to be tested a bit more rigorously. Can phagocytosis of platelet-derived vesicles before and after LPS be quantified? Can it be tested over a time course using flow cytometry e.g., using platelets labelled with pH-sensitive dye so that the signal is only detected when platelets or their vesicles reach a low pH phagosome compartment?

Response: We thank the Reviewer for this valuable input. Firstly, we would like to highlight that proving the successful transportation of cargo is consistently challenging, as explained above. However, we show with Mass spec proteomics coupled with SILAC labeling of amino acids that proteins and peptides labeled exclusively in the donor cells (megakaryocytes) can be detected in the recipient cells (monocytes) even after extensive washings, and using cell-free culture supernatants of MKs. Moreover, our experiments enhancing NF- κ B-driven pro-inflammatory cytokine secretion in genetically deficient monocytes, lacking these key cytokine-regulatory TFs, indicate that platelet-derived TFs function within recipient monocytes.

Following this Reviewer's recommendation regarding the phagocytosis of platelets/vesicles, we employed pHrodo as a pH-sensitive dye. We stained human platelets with different concentrations of pHrodo and applied flow cytometry and confocal microscopy to assess the phagocytosis of pHrodo particles by StdMo or PdMo. Unfortunately, flow cytometry did not allow us to detect a shift in the fluorescence of pHrodo, even in our positive controls (pHrodo-stained platelets in neutral vs. low pH) (Rebuttal Fig. 1).

We then employed confocal microscopy to examine whether pHrodo⁺ platelets are engulfed by primary human monocytes and, thereby, shift from pHrodo^{dim} to pHrodo^{bright}. Despite these efforts, we observed no shifts in fluorescence in pHrodo⁺ platelets or pHrodo⁺ particles inside StdMo or PdMo. Another technical issue was that platelets themselves can acidify pHrodo, creating high fluorescence backgrounds that clouded our imaging and our ability to conclude this specific question (Rebuttal Fig. 1).

**Rebuttal Fig. 1.**

(A - B) Flow Cytometry (A) and Confocal imaging (B) of human platelets stained with pHrodo (1 μ M, red). Platelets left unstained, or were stained with pHrodo and maintained in neutral or low pH, and assessed at different timepoints (from 0 - 3 hours). Images are representative of three independent experiments and show the time point 0h. In A, flow cytometry data presented as a histogram displaying the staining with pHrodo (10 μ M) as well as the unstained control.

(C) Confocal imaging StdMo or PdMos incubated with pHrodo-stained platelets. Nuclei were stained with DRAQ5 (blue). Monocytes were directly stained with pHrodo (as positive control, top panels), or incubated with pHrodo-labeled platelets.

Although platelets can independently acidify pHrodo and increase its fluorescence, we could not observe pHrodo⁺ platelets nor pHrodo⁺ particles in StdMo or PdMo. This suggests that the uptake of particles or EVs is not directed into the phagolysosomal machinery, highlighting that the transferred cargo is not degraded and consistent with the mechanisms of vesicle-delivered cargo, though it still needs experimental validation.

Finally, our new dataset (as described below in this letter) demonstrates that removing small vesicles from platelet supernatants extinguishes the platelet effect, supporting that platelet-derived cargo is probably too small for imaging.

- Does LPS treatment of co-culture increase vesicle release by platelets? In Fig 5C, can platelet supernatant (which restores monocyte responsiveness) be further fractionated to remove vesicles, or can vesicles be purified?

Response: In response to the Reviewer's inquiry regarding the influence of LPS treatment on platelet vesicle release and the potential for further vesicle purification from platelet supernatant, we conducted a comprehensive investigation, as outlined below:

We isolated platelets from healthy volunteers and left them unstimulated or stimulated with LPS for 3 hours, mirroring our standard priming protocol in our study. Subsequently, platelets were pelleted at (430 x g for 10 minutes) to obtain platelet releasates (Plt Sups). The Plt Sups were utilized directly or submitted to ultracentrifugation (100,000 x g for 3 hours) to pellet and enrich small vesicles (Vesicle-Enriched Fraction, VE in the figure).

Appendix Fig. S8. Fractionation of platelet releasates and isolation of platelet vesicles.

- (A) Protein concentrations of platelet lysates, platelet releasates (Plt Sups), or platelet vesicles measured by Pierce™ BCA Protein Assay Kit (Thermo Fisher Scientific). Conditions included treatment with LPS (2 ng ml⁻¹) for 3 hours or no treatment. Each symbol represents one donor.
- (B) Vesicle analysis of the different platelet supernatants using NanoSight3000. Each sample was diluted 1:10 with PBS and 1 ml total volume was used and recorded three times for 60 seconds before analysis. On the left are three individual runs of the same condition (marked by different colors: yellow, orange, and red). On the right is the calculated mean. The Y-axis represents the concentration of particles detected per milliliter, and the X-axis shows the size (nm) of the vesicles/particles. Each graph represents one independent experiment or donor.
- (C) WES-capillary electrophoresis and immunodetection of NF-κB2 from vesicle-enriched fractions of platelet releasates (VE) from 4 healthy subjects. Platelets were left untreated (Unstim), or stimulated or restimulated with 2 ng/ml LPS for 3 hours before the assessment.
- (D) IL-1β and TNFα levels in the cell-free supernatant (CFS) of PdMo and PdMo supplemented with autologous platelets, platelet releasates (Plt Sups), or concentrated platelet vesicles. Platelets, Plt Sups, or vesicles were pre-treated with LPS (2 ng ml⁻¹) for 3 hours before addition to PdMo. Cells were stimulated with LPS (2 ng ml⁻¹) followed by activation with nigericin (10 μM). Graphs with floating bars depict maximum/minimum values relative to the mean (white bands). P values were calculated using 1-way ANOVA, Tukey's multiple comparison test, and are indicated in the figure. Each symbol represents one donor.

Our analysis comprised BCA protein assays and WES capillary electrophoresis to assess changes in protein abundance in releasates (Plt Sups) and vesicles following LPS stimulation (New Appendix Fig. S8) and to detect NF-κB2 expression across these fractions (New Fig. 8A). Our findings indicated no significant increase in vesicle production or NF-κB2 expression in response to LPS treatment (Appendix Fig. S8 A and C), consistent with the capacity of unstimulated platelets to mitigate impaired cytokine production in PdMos (reported in the original manuscript). Furthermore, we were able to demonstrate that LPS stimulation of platelets, supernatants, or vesicles have no additional boosting effect on the primary monocytes (Appendix Fig. S8 D).

Next, to specifically investigate the vesicle-mediated protein transfer from platelets to monocytes, we employed size exclusion chromatography and depleted vesicles from platelet releasates, vesicle-free platelet fractions (VF, in Figures) and evaluated their impact on monocyte cytokine production. Notably, the absence of vesicles did not enhance monocyte

Fig. 8. Platelet vesicles underlie the platelet's ability to reconstitute NF- κ B signaling in NF- κ B $^{-/-}$ monocytes.

- (A) WES capillary electrophoresis and immunoblotting of NF- κ B2 on lysates, releasate, and vesicle-free releasates of purified platelets from six healthy volunteers. Platelets were left untreated, or activated with 1 U/ml of Thrombin.
- (B) Levels of IL-1 β and TNF α in the CFS of PdMos that were replenished with platelets (+ Plts), unfractionated platelet releasate (+ Plt Sups), or isolated vesicles, obtained after vesicle-enrichment of platelet releasates (+ Plt vesicles). Pooled from 3 independent experiments. Symbols represent different donors.
- (C) Assessment of IL-1 β and TNF α levels in the CFS of StdMo, PdMos, and PdMos that were replenished with platelets (PdMo + Plts), unfractionated platelet releasates (+ Plt Sups) or releasates depleted of vesicles (+ Plt Vesicle-free). Pooled from 6 independent experiments. Symbols represent different donors.
- (D) IL-1 β levels released into the CFS of stimulated Wild-type WT, RelA $^{-/-}$, or NF- κ B2 $^{-/-}$ THP-1 monocytes that were cultured alone or added with platelets, unfractionated platelet releasates (+ Plt Sups) or isolated vesicles, obtained after vesicle-enrichment of platelet releasates (+ Plt vesicles). All data are displayed as floating bars with max/min values and indication to the mean (white bands). Each symbol represents one donor. For simplicity, LPS or LPS + Nig conditions are shown. Additional conditions are shown in Appendix Fig. S8D.

cytokine response (Fig. 8C), highlighting the importance of vesicles in platelet-monocyte interactions. This was further corroborated by the NanoSight3000 analysis, which demonstrated effective vesicle removal (Appendix Fig. S8B).

Our comprehensive approach thus indicates that platelets transfer proteins to monocytes via vesicles. This addresses the query regarding the role of LPS in vesicle release and the potential for vesicle purification from platelet supernatant.:

- Do vesicles need to be coming from platelets, or they can come from any other cell type?

Response: Vesicles are released by several cell types. However, platelets are of the highest significance as these are the primary source of vesicles in the stream (80% are platelet-derived). Still, here we show that co-incubation of monocytic THP-1 cells with Megakaryocytes, but not HEK cells boosts the IL-1 β production (Rebuttal Fig. 2). Furthermore, as reported before (Rolfes et al., 2020) MKs did not secrete IL-1 β when cultured alone, excluding that the increase in IL-1 levels was not due to their contribution to the pool of cytokines in the supernatants. These

Rebuttal Fig. 2. THP1 cells were cultured alone and mixed with MEG-01 cells or HEK 293-T cells before being stimulated with LPS or LPS + Nigericin. IL-1b levels were measured in culture supernatants. As a control, MEG-01 and HEK293T cells were cultured alone or combined in the absence of THP1s.

results demonstrate that despite numerous cells releasing vesicles, the effect we describe comes from platelets/MKs.

• Does LPS treatment of co-cultures increase vesicle release by platelets?

Response: Please see the response above.

Question 5. It was clear that the addition of platelets increases NF-κB reporter activity in THP reporter monocytes stimulated with Pam3CysK4. Would platelet supernatant transfer (+/- vesicles) or transwell culture do the same when transferred into NF-κB KO THP cells?

Response: Akin to the experiments performed on primary human monocytes (above), we next aimed to investigate the ability of enriched platelet vesicles to reconstitute the defective cytokine

response in NF-κB2 (NF-κB2^{-/-}) and NF-κB-p65 (RelA^{-/-}) knockout THP1s. To examine this, we subjected four THP-1 clones and the parental WT strain (used in the main Fig 8G, of our original manuscript) to various treatments, including platelets, platelet supernatant, and concentrated platelet vesicles, pelleted from releasates at 100,000 x g for 4 hours.

THP-1 cells were primed with Pam3CysK4 for 3 hours, followed by nigericin stimulation. Recapitulating our previous findings, deficiency in NF-κB2 and NF-κB-p65 impaired cytokine responses to Pam3CysK4. Likewise, platelets and platelet releasates (Plt Sups) could boost cytokine production in stimulated NF-κB2^{-/-} and RelA^{-/-} cells (NEW Fig 8D, above). We also observed that enriched platelet vesicles were equally effective in enhancing the cytokine response of knockout cells. These findings indicate that vesicles serve as the primary transport system through which platelets reconstitute these signaling pathways in deficient monocytes.

Question 6. How long do platelets need to be present in the culture to functionally reprogram monocytes?

Response: experimentally, platelets are re-added to the PdMo 45-60 minutes after platelet depletion, and all groups are then directly stimulated with different TLR agonists. We also tested

Rebuttal Fig. 3. To assess whether the platelet-induced effect depends on the timing of the LPS-stimulation after the co-incubation with platelets, StdMo, PdMo and PdMo + Plts were seeded and either directly stimulated with LPS (2 ng ml⁻¹) or rested for 2 h and then stimulated with LPS for 4.5 h for IL-6 and TNFα secretion, or 3 h followed by 1.5 Nigericin (10 μM) for inflammasome activation.

two other scenarios: i) a delayed re-addition of platelets for 2 h to let the monocytes rest longer (**Rebuttal Fig. 3**), and ii) a prolonged co-culture and LPS stimulation (up to 17 h). All scenarios showed the same results. These data highlight a rapid cross-talk event: once monocytes are separated or co-cultured with platelets, the impact on the monocytic immune response is fast.

However, the alternative scenario of co-incubating platelets to PdMo and removing them again is experimentally and technically impossible. In co-culture, monocytes are adherent, and platelets quickly adhere to monocytes, or release vesicles, which cannot be washed out (tested in our laboratories).

Question 7. Is monocyte metabolism affected by platelet presence?

Response: We present to his reviewer our preliminary findings indicating that platelets affect the metabolism of primary human monocyte-derived macrophages (hMDMs) and monocytes. We undertook RNA sequencing (RNASeq) experiments to investigate the gene profile of LPS-primed human macrophages that were co-incubated with platelets, or platelet-supernatants (**Rebuttal Fig. 4, below**). A Kegg-Pathway analysis of the 1725 differentially expressed genes (DEGs) induced by LPS stimulation of hMDMs indicated most of the commonly expected signalling pathways triggered upon LPS recognition (ex: NOD-like receptor signaling pathway, Chemokine signaling pathway, infections, among others). Notably, the gene expression profile of macrophages, irrespective of LPS-stimulation, was drastically altered by the addition of platelets, or platelet releasates (Platelet Sups). The vast majority of DEGs from hMDMs added

Figure for referee with unpublished data and its description has been removed upon request by the authors.

Figure for referee with unpublished data and its description has been removed upon request by the authors.

with platelets represented signalling pathways related to sugar and amino acid metabolism, cancer and metastasis, and TGF- β signalling (**Rebuttal Fig. 4a-c**). We next, conducted Seahorse Assays on hMDMs (**Rebuttal Fig. 5**) and found that addition of platelets to human macrophages boosted their glycolytic capacity (**Rebuttal Fig. 5**). Similar assays were conducted in primary human monocytes (StdMo, PdMo, PdMo + Plts, and Plts-alone). We included these findings in this rebuttal letter with the intention of addressing this reviewer's question, as it was not clear to us if he/she was asking for additional experiments. However, we believe this topic is beyond our study's scope, and wish therefore, that they are not included in the revised manuscript.

Question 8. Direct ex vivo analysis of monocytes from platelet-depleted mice suggests their transcriptional reprogramming towards cell cycle and division at the expense of immune defense programs. Can authors comment on this? Do blood monocytes cycle normally, or does platelet depletion cause some kind of differentiation state change?

Response: Typically, classical CD14⁺ monocytes are considered non-proliferative cells. However, only a few monocytic subpopulations can exhibit potentially proliferative properties in a specific context (Lari Kitchener & Hamilton, 2009). As most of the detected DEGs indicate alterations in processes such as proliferation, "housekeeping," and "cell maintenance," these genes can also be involved in the differentiation and polarization of a monocyte. This supports other findings highlighting the role of platelets and platelet-derived factors in the polarization and differentiation state of monocytes and macrophages (Chatterjee et al, 2015; Scheuerer et al, 2000).

Question 9. Inhibitor pre-treatment of platelets before addition to monocytes does not prove that the transcription factor transfers and trans-signaling in monocytes are the mechanism, as there could be NF- κ B and p38-induced factors(s) inside platelets responsible for monocyte reactivation when transferred via vesicles.

Response: *"Inhibitor pre-treatment of platelets before addition to monocytes does not prove that the transcription factor transfer and trans-signaling in monocytes are the mechanism..."* The experiments that sustain the transference of TFs between platelets and monocytes are the Mass-spec proteomics where we detected the specific TFs that were labeled on platelets but ended up on monocytes, as well as the reconstitution of cytokine functions in knockout cells lacking NF- κ B subunits, or primary human cells where these key TFs were pharmacologically inhibited.

"as there could be NF- κ B and p38-induced factors(s) inside platelets responsible for monocyte reactivation when transferred via vesicles." Indeed, we acknowledge this limitation in our findings, and throughout our manuscript, we were consistently careful in making statements that align with these limitations.

We are starting with the manuscript title, stating that **platelet-derived transcription factors** (without specifying one) license monocyte inflammation. We now propose the title: **"Platelet-derived transcription factors license pro-inflammatory cytokine secretion by human**

monocytes." to more accurately reflect our findings, given that migration and phagocytosis of bacteria were virtually unaffected (as shown below).

In lines 358-360 of our original manuscript, we stated: "These findings support that **p38 and NF- κ B or their downstream signaling** in platelets are essential for the platelet regulation of cytokine responses of human monocytes." As we could not pinpoint a single platelet-derived molecule driving the cytokine response on monocytes, we were careful enough to make those statements.

Additionally, in lines 379 – 383 of our original manuscript, we stated: "Notably, adding platelets to RelA^{-/-} or NF- κ B2^{-/-} monocytes dose-dependently boosted their production of IL-1 β , demonstrating the **reactivation of these pathways** (instead of transfer of these molecules) in otherwise knockout recipient cells. These findings demonstrate that platelets supply human monocytes with **functional signaling TFs**, which license their optimal cytokine responses and reveal a novel layer of cell-to-cell communication in innate immunity."

We now included "Our study provides the first evidence for the propagation of NF- κ B and p38 signaling from platelets to monocytes, with maintained functions in monocytes.", **line 485** of the revised manuscript.

Finally, our SILAC mass-spec proteomics revealed the presence of additional platelet-derived TFs in monocytes, including **STAT1, THYN1, STAT3**, and others (**Figure EV7 in the original manuscript**). Several of these molecules influence cytokine responses of monocytes, and their relevance remains to be experimentally addressed in follow-up studies.

Question 10. Why do platelet-free monocytes have increased Jak signaling? Can authors comment on this? Is Jak activation key to monocyte reprogramming or unresponsiveness?

Response: Good question, and we don't know. However, it could be related to IFN-related transcription factors (STAT1, for example) detected in our Mass-spec proteomics. It is unclear to us whether the reviewer would like us to include this in the manuscript's discussion.

Minor points

11. Fig1B should include a loading control for the whole cell lysate

Response: We appreciate the Reviewer pointing this out.

We have performed loading controls, including the staining of β -actin, proving the loading of all the capillaries with sample material. However, it is essential to highlight that this is a co-culture experiment, where platelets and monocytes both contribute to the pool of β -actin. Hence, even as the lysates contained an equal number of seeded monocytes, the differences in β -actin intensities between StdMo and PdMo are misleading, as they derive from the presence vs. absence of platelets (which also have β -actin). We have used the WES capillary system, resulting in virtual data that we could revisit. The system also includes internal controls for the loading, which are shown in the **Rebuttal Fig. 6** below. To show that all capillaries were loaded with the same amount of sample material, we adjusted the contrast to display smears with crosslinked proteins in the capillaries (analogous to a "ponceau" staining).

Furthermore, as we show that platelets do not express IL-1 β , excluding their additive effect to the IL-1 pool, the IL-1 β bands measured in StdMo and PdMo are expressed exclusively in the monocytes. They can be compared among all monocyte groups – despite the intensities of the β -actin bands.

Please note that in all conditions where platelets were added to monocytes (Rebuttal Fig. 5C, **Lanes 20-22**), or even platelets alone (**Lanes 23-25**), β -actin is detected in cell-free supernatants (CSF) likely arising from platelet vesicles not pelleted during sample preparation. Indeed, β -actin was not detected in monocytes without the addition of platelets (Lanes 14-19).

Rebuttal Fig. 6. Loading controls and b-actin immunodetection for the WES capillary electrophoresis system.

compared responses of monocytes isolated by positive (Miltenyi) and negative (StemCell) selection protocols from two manufacturers. StemCell also provides positive selection protocol/beads, so ideally, those should be included in this figure to compare monocytes obtained by the same manufacturer's positive versus negative selection protocols.

Response: The Reviewer raises a valid point. Nevertheless, we hope the reviewer appreciates that the new mouse data recapitulating the observations made in human monocytes that were positively selected supports the fact that this is not related to one specific kit or company.

Referee #2 (Comments on Novelty/Model System for Author):

As indicated in my comments to the authors, the paper presents the first direct demonstration of transfer of functional transcriptional factors from platelets into monocytes and illustrates that they are functionally competent to elicit measurable immune mediators. The study is all done with human samples which, while still pre-clinical, has the potential of translation to the clinic in the future. The limitations and drawbacks of some experiments are detailed in the response to the authors.

Referee #2 (Remarks for Author):

The paper by Hawwari, Franklin et al. introduces the notion that transcription factors can be exogenously delivered to monocytes to enable effective responses to microbial products. This is an exciting concept that, while not new, is robustly demonstrated in human monocytes *in vitro* using a plethora of elegant and diverse strategies. An obvious drawback from the studies is the *in vitro* nature of all the assays used here, while the *in vivo* confirmation in a mouse model is non convincing given the challenge of removing platelets from a living organism in a controlled manner. Nonetheless, the value of the paper is the demonstration that immune cells can be licensed *in trans* and acquire active mechanisms of activation through interactions with other cells. The limitations of the study need to be precisely described in the text to avoid confusion.

Response: We thank this reviewer for the positive words on the relevance of our findings. We have given due consideration to the points regarding the limitations of the *in vitro* system, and included a paragraph in the discussion of the revised manuscript. It reads as follow:

"Study Limitations: Our study provides evidence in primary human monocytes from healthy volunteers, and in monocytes from patients with inherited platelet abnormalities, that platelets are essential for the pro-inflammatory cytokine response of monocytes. While our findings point towards a potentially novel mechanism of cell-to-cell communication regulating innate immune responses, most of our experiments were performed *in vitro*, after *ex vivo* stimulation of isolated primary human immune cells. In our efforts to reproduce the phenomenon *in vivo* we found similarities in mouse isolated monocytes. However, the effect of platelets was not pronounced in a systemic level, given the challenges of removing platelets from a living organism in a controlled manner, and could be explained by numerous other factors, not ruled out in the present study. Despite our efforts to address monocyte functions beyond cytokine secretion, such as migration, phagocytosis, and microbicidal activities, technical limitations prevented us from reliably identifying distinguishable effects of platelet-depletion on these processes. Further studies are needed to validate these findings *in vivo* and explore their potential clinical implications."

Below are some comments and suggestions to improve this otherwise complete study:

- In addition to analyzing monocytes isolated using the different platforms, I suggest assessing the differential response of monocytes alone vs MPA isolated from blood in cytokine production assays. The idea is to establish whether specific populations of monocytes already existing in the human circulation (not only in the test tube) are more prone to immune activity.

Response: This is a great idea and a valuable insight into our study.

To address this question, we isolated human PBMCs. We employed FACS-sorting to purify the two populations of classical CD16⁻ CD14⁺ human monocytes: Platelet-free monocytes (CD41⁻ CD14⁺ equivalent to PdMo) and MPAs (CD41⁺ CD14⁺, equivalent to StdMo). As illustrated below:

Schematic representation of the gating strategy and FACS-sorting procedure to obtain platelet-associated (StdMo) vs. platelet-free (PdMo) monocytes, directly from human PBMCs.

Unfortunately, after numerous trials and optimizations, and procedures that took **over 9 hours of FACS-sorting** and large volumes of human blood, we could not achieve sufficient yields of platelet-free monocytes for downstream analysis. We also noticed that the fraction of platelet-free monocytes (CD41⁻ of the CD16⁻ CD14⁺ cells, GR1 gate in **Rebuttal Fig. 7**) gradually disappeared throughout the sorting, accumulating in the MPA gate (CD41⁺ CD14⁺,

Rebuttal Fig. 7 - Flow Cytometry-based sorting of human PBMCs to isolate platelet-associated (StdMo) vs. platelet-free (PdMo) monocytes, directly from human PBMCs. Human PBMCs were isolated from the peripheral blood of healthy volunteers through FICOLL gradient. Cells were stained with mAbs targeting CD14, CD16, and CD41a. Gating strategies are shown. Panels on the right show aggregation of CD14⁺ monocytes towards the CD41a⁺ gate, indicating increased aggregation of platelets to monocytes. Data is from one representative of several independent trials.

GR1 gate, in **Rebuttal Fig. 7**), consistent with the aggregation of platelets to monocytes. Unfortunately, we could not address the specific question without relying on magnetic isolation kits due to these technical difficulties. We included in this re-submission a letter from our Flow Cytometry Core Facility managers ("*Letter from FACS Core Facility - Bonn*"), who tried their best to help us to circumvent these limitations.

- It would be interesting to test other features of monocytes in the presence or absence of platelets, including migration (chemotaxis), phagocytosis (beads) and microbicidal activity. The point would be to assess not only specificity or not for cytokine production but more generally the extent to which this platelet-monocyte axis influences immune function.

Response: Another extremely relevant suggestion. We thank this Reviewer for suggesting further assessment of monocyte functions beyond cytokine production, such as migration, phagocytosis, and microbicidal activity.

We examined monocyte transmigration using a trans-well system (Chatterjee et al., 2015; Hu et al, 2008) with monocytes and platelets and chemokine CCL2 stimulation. We tested different conditions, including titration concentrations of platelets and chemokines CCL2, alternating through plates with different pore sizes, and using platelet supernatant, or MEG-01 supernatants to prevent clogging. Despite these efforts, these assays were not conclusive, and we observed only a slight trend towards increased monocyte migration upon platelet addition, but without significant differences compared to the standard monocytes (StdMo) (New Fig. EV5A-B). These findings supports original observations that platelets enhance monocyte transmigration (Chatterjee et al., 2015; Sreeramkumar et al, 2014; Zuchriegel et al, 2016), but we could not observe differences between platelet-depleted monocytes vs. StdMos in their ability to transmigrate.

Additionally, we assessed the phagocytic capacity of human monocytes by exposing them to live *E. coli* constitutively expressing mCherry. The results showed that both StdMo and platelet-depleted monocytes (PdMo) could ingest bacteria, with a slight increased phagocytic activity in PdMo. We also confirmed active phagocytosis, from bacteria invasion, by inhibiting actin polymerization with cytochalasin D, which effectively blocked bacterial internalization (New Fig. EV5C-D).

While we could not reliably assess monocyte migration upon platelet depletion, we confirmed previous findings that platelets facilitate this process. We also noticed a potential influence of platelets in the phagocytic capacity of monocytes suggesting changes in functional priority—monocytes might decrease cytokine activity when phagocytosis is engaged. Consistent with the

Fig. EV5 – Effects of platelets on monocyte transmigration and phagocytosis.

(A – B) Primary monocytes (StdMo, or PdMo) were supplemented with platelets, platelet (Plt Sup) or Mk releasates (Mk Sup) were seeded on the upper chamber of a transwell plate with either 3 μm or 5 μm pore sizes and incubated with CCL2 (40 ng ml⁻¹) or left untreated for 4 hours. Monocyte transmigration was measured by confocal imaging and quantification of cells that migrated to the bottom well and stained with DRAQ5. 16 pictures per well were taken and the nuclei were counted via Cell Profiler. Log₂ fold change was calculated and the CCL2 conditions were normalized to their respective unstimulated condition.

(C - D) Confocal Imaging (C) and quantification (D) of StdMo and PdMo exposed to an mCherry fluorescent *E. coli* strain (1:25 MOI, for 2 hs). Monocytes were either left untreated (–) or pre-treated with the phagocytosis inhibitor cytochalasin D (CytoD) and exposed to 100:1 MOI of *E. coli*. Cells were stained with DRAQ5 (blue, nuclei) and WGA-AF488 (green, membranes). Images were acquired by confocal and automated widefield microscopy. Images (four per condition) were analyzed by counting the *E. coli* inside five monocytes per image, by two independent experimenters. Mean numbers of *e. coli* per cell were plotted.

Graphs with floating bars depict maximum/minimum values relative to the mean (white bands). P values were calculated using 2-way ANOVA, Tukey's multiple comparison test. Each symbol represents one donor

notion that phagocytosis and the activation of intracellular signaling for inflammation as somewhat separate processes (Swanson, 2008; Underhill & Goodridge, 2012). These findings also align with the predominant actions of NF- κ B and MAPK signaling in controlling cytokine secretion. This functional variation could reflect physiological differences between blood monocytes and those migrating into tissues. More comprehensive studies are needed to understand these dynamics better and verify these preliminary findings.

- Fig.4: The data on thrombocytopenic mice are difficult to interpret and at best the data suggest modest changes when compared with the dramatic effect seen in vivo with human monocytes. This is interesting but of some concern for the extension of the proposed model from human to mice, while at the same time highlighting that experimental depletion in vivo or other systemic effects induced by platelet depletion are not simple to explain. While this validates the in vitro strategy for human cells, I am missing experiments in murine monocytes showing that at least ex vivo the trend is the same. In sum, either this data with mouse cells is added or it would be advisable to remove this section altogether.

Response: We agree with this Reviewer that the mouse data do not entirely recapitulate our observations in the human system. Although additional research is necessary to fully address this issue, we have now included experiments to demonstrate that, *at least ex vivo, the trend is the same in the mouse system:*

We isolated CD11b⁺ Ly6C⁺ monocytes from mouse PBMCs through positive magnetic selection. The predominance of monocytes within the isolated population was confirmed by FACS, where nearly all CD11b⁺ cells expressed the CD45 surface marker, while the residual population comprised platelets. Moreover, a significant proportion of CD45⁺CD11b⁺ cells co-expressed Ly6C, confirming the predominance of monocytes. Mouse monocytes were seeded at a density of 10⁵ cells per well (same as we did in the human system) alone or with 10⁷ platelets (100 platelets per monocyte) (New Fig. EV2E).

Figures EV2E and Main Fig 1H – Gating strategy and assessment of isolated murine monocytes.

- (E) Representative Flow Cytometry and gating strategy to assess the purity of murine monocytes isolated from mouse blood. Cells were gated based on surface expression of CD45, Ly6C and CD41a. Images are from one representative of 3 independent experiments with $n = 4$ mice. **Now as Fig. EV2E in the revised manuscript.**
- (F) Concentrations of IL-1 β assessed in CFS by positively-isolated mouse CD45⁺ CD11b⁺ Ly6C⁺ blood monocytes that were culture alone (Monocytes) supplemented with autologous platelets (Monocytes + Plts) at a 100:1 platelet:monocyte ratio. Cytokine levels secreted by platelets alone (Plts) were measured as control. Cells were stimulated with LPS (1 μ g ml⁻¹) followed by activation with nigericin (10 μ M). Graphs with floating bars depict maximum/minimum values relative to the mean (white bands). P values were calculated using 2-way ANOVA, Tukey's multiple comparison test, and are indicated in the figure. Each symbol represents pooled data from 3 mice, $n = 12$ in total. **Now as Fig. 1H in the revised manuscript.**

Similar to our findings in human monocytes, the cytokine response of positively selected mouse monocytes was also boosted by platelets (**New Fig. 1H**). Conversely, platelets alone exhibited no cytokine response, confirming monocytes as their primary sources.

These results were included in the manuscript as new panel **H** in main Fig 1, and Fig. **EV2**, and are now described in the text (**lines 135:141**).

- Fig.5A, it is impossible to evaluate from these images where platelets are or not engulfed by monocytes. Images should be clearer and this quantified in some form if the authors wish to make this point.

Response: We appreciate the Reviewer's feedback regarding the clarity of the images in Fig. 5A. We would like to clarify that the images presented were intended to assess the phagocytosis of platelets by monocytes. However, it is essential to note that the platelets release small vesicles, now confirmed in our revised study, that are too small to be resolved with conventional confocal microscopy, especially after being internalized in monocytes. This limitation is addressed in the revised manuscript to avoid any misunderstanding about what the images can show.

- EV5 and other figures, clearly illustrate how variable the response of human monocytes is in response to stimulation, which raises the question of whether the low number of patients used in these experiments yields a reliable conclusion. For example, the inhibition by blocking Siglec7 in DVFH is clear yet the authors conclude lack of an effect based on poorly reliable statistics. This can be also easily taken by intended bias towards a model favored by the reviewers. This is a nice and meritorious study which can easily lose credibility by this type of loose analyses and interpretation.

Response: We appreciate this Reviewer's constructive feedback on the variability observed in the response of human monocytes. The initially limited number of biological replicates was due to our focus on corroborative experiments that targeted the desialylation of platelets, supporting the

Rebuttal Fig. 7

IL-1 β and TNF α levels in CFS of StdMo, PdMo, PdMo, or PdMo that were supplemented with untreated platelets or platelet-bound to anti-siglec7 (10 $\mu\text{g}\cdot\text{ml}^{-1}$), and stimulated as in A. N = 7 biological replicates. IL-1 β and TNF α levels in CFS of StdMo, PdMo, PdMo, or PdMo that were supplemented with untreated platelets or platelet-bound to anti-siglec7 (10 $\mu\text{g}\cdot\text{ml}^{-1}$), and stimulated as in A. N = 7 biological replicates. Now presented as Appendix Fig. **S6j**.

exclusion of the Siglec7 pathway.

Acknowledging this Reviewer's concern, and to strengthen the robustness of our dataset, we have now performed additional experiments including four more donors, bringing the total to seven different donors pooled in one graph. These updated results, which combine the original and new data points, reinforce our conclusion that Siglec7 crosslinking does not replicate the effect of platelets, nor does it do so to the extent observed with the IgG control. The expanded dataset indicates that neither Siglec7 on monocytes nor sialic acids on platelets play roles in the mechanisms our study reports.

We are grateful for this reviewer's insights, which have significantly strengthened the credibility and robustness of our study.

- A relevant control for the experiments in Fig8G is whether the NF- κ B inhibitors will fully block the increase in IL1 β secretion in KO monocytes supplemented with platelets. This is important to validate that no other alternative pathways are stimulated by platelet-derived factors.

Response: We fully agree with the Reviewer's suggestion and have explored the proposed experimental control using the NF- κ B inhibitor BAY 11-7082. Initially, our results demonstrated that BAY 11-7082 affected THP-1 cells differently than primary human monocytes, a difference we had noted before conducting knockout (KO) experiments with THP-1 cells. This discrepancy is significant as primary human monocytes have limited availability for KO experiments.

In response to the Reviewer's specific suggestion, we treated freshly isolated platelets with BAY 11-7082 and co-stimulated these with wild-type, or CRISPR-edited THP-1 clones lacking NF- κ B2 and NF- κ B p65. We confirmed that BAY prevented platelet activation in response to thrombin based on the surface expression of P-selectin (CD62P) and PAC1 (Appendix Fig. S7A-B). However, BAY-treated platelets showed similar capacity to enhance IL-1 β secretion by knockout cells, compared to intact platelets, indicating that additional players (i.e. other NF κ B subunits, or MAPKs) may be involved, but also highlighting differences between THP1s and primary human monocytes.

The mammalian NF- κ B family consists of five members (i.e.: NF- κ B1 (p105/p50), NF- κ B2 (p100/p52), p65 (RELA), RELB, and c-REL), who share the conserved Rel homology domain (RHD), enabling them to form homo- or heterodimers (Vallabhapurapu & Karin, 2009). These dimers bind to the inhibitor of κ B (I κ B) proteins, retaining them in an inactive state within the cytoplasm. BAY 11-7082 inhibits NF- κ B activation by targeting I κ B kinase (IKK), and preventing I κ B phosphorylation and its subsequent degradation, crucial steps in both canonical and non-canonical pathways. This hinders NF- κ B from translocating to the nucleus to activate its target

Appendix Fig. S7 - The IKK inhibitor (BAY11-7082) prevents platelet activation.

(A - B) Flow cytometry analysis and (B) quantification of CD62P and PAC-1 expression on purified human platelets. Platelets were left untreated (-), or treated with BAY11-7082 (50 μ M), DMSO (50 μ M) for 20 min, before being stimulated with Thrombin (1 U ml⁻¹) for 30 min. Left: Representative FACS histogram for CD62P expression. (B) Dotplot displays the percentage of CD62P⁺ platelets in n=6 different donors with the indication to the mean.

**Fig. EV4**

(F) IL-1 β concentrations in CFS in wild-type, or two clones of NF- κ B^{-/-}, or RelA^{-/-} THP-1s co-cultured with intact (normal) or pre-treated platelets for 30 mins with 50 μ M BAY 11-7082.

genes. The THP-1 cell line appears to be less reliant on a specific NF- κ B signaling pathway compared to primary cells, as cytokine production persisted despite IKK inhibition on platelets. Notably, inhibition of both canonical and non-canonical pathways with BAY 11-7082 in platelets did not prevent them to restore IL-1b production in RelA clones or in clones deficient in the non-canonical pathway due to NF- κ B2 deletion (New Fig **EV4 F**),

underscoring the complexity of NF- κ B pathways, and the differences between primary monocytes and THP1s. Indeed, although platelets enhance IL-1 β secretion on THP1s (Rolfes et al., 2020), these cells do not rely on platelets for primary IL-1 β responses. It is plausible that alternative NF- κ B pathway activation is involved in THP-1s compared to primary human monocytes, supported by the fact that p50 also tends to bind cRel. Additionally, NF- κ B pathway complexity can vary between cell types, as exemplified by RelB promoting dendritic cell activation regulated by the canonical NF- κ B pathway through a RelB-p50 dimer (Rauert-Wunderlich et al, 2013; Shih et al, 2012), suggesting divergent NF- κ B signaling in THP-1 cells compared to primary blood monocytes. Consequently, pinpointing a single factor, or NF- κ B subunit remains a challenge (Kasper et al, 2010), necessitating further experiments to selectively inhibit various subunits or simultaneously other signaling pathways and elucidate precise mechanisms. We have now acknowledged these limitations in the discussion of our revised manuscript.

- In the discussion, I miss ideas from the authors on how and where the "platelet-licensed" monocytes function, given that interactions must take place in the blood stream but antimicrobial activities, for example, take place mostly in the tissue parenchyma. I also miss discussion on the differential activity of MPA vs. free monocytes, which would suggest that priming for a subset of monocytes occurs in blood but the duration of such priming was not tested here.

Response: We thank the Reviewer for his/her insightful comments regarding the localization of "platelet-licensed" monocyte functions and the differences between monocyte subtypes. Our focus in this study has been primarily on cytokine responses which, as demonstrated in previous research (Bonnet et al, 2021; Fajgenbaum & June, 2020; Ferreira et al, 2021; Jafarzadeh et al, 2020; Junqueira et al, 2022; Schulte-Schrepping et al, 2020; Vanderbeke et al, 2021), predominantly occur in the bloodstream. This aligns with our observations that vesicles mediate platelet-monocyte interactions and the known concept that platelets are the primary sources of vesicles in the blood (described initially as "Platelet dust" by Peter Wolf in 1967 (Hargett & Bauer, 2013) also primarily occurs within the vascular system.

The Reviewer's point regarding the differential activities of monocyte populations (MPA versus free monocytes) is well taken. While our study did not extend to the duration of

monocyte priming effects in blood, we acknowledge the importance of this aspect. Given the rapid differentiation of monocytes into macrophages upon entering tissues— where platelets may have more restricted access — it remains technically challenging to evaluate the lasting effects of platelet priming on monocytes within tissue environments.

We appreciate this Reviewer's suggestions for deepening the discussion on these topics and will consider exploring these dynamics in future work.

Minor:

- Fig.4B: platelet counts should be given as absolute counts, not percentages.

Response: We understand the Reviewer's point and recognize the preference for absolute counts; however, due to the variability in the volume of a "drop" of blood, percentages provide a more consistent measure in our methodology. We determine platelet abundance using flow cytometry, where the non-uniform volume of a blood drop makes absolute counts less reliable. Thus, we use percentages to ensure accuracy and reproducibility in our results.

- Line 264: the statement that "Next, we blocked integrins..." is strange as CD11b is an integrin.

Response: Here, we rephrased to: "Next, we broadly blocked the class of integrins," as we were interested in blocking all integrins using RGDs as "pan inhibitors." Indeed, we know that CD11b is an integrin, which we also wanted to investigate more specifically in the CD11b-GPIIb/IIIa axis as described in the literature.

- some explanation and interpretation of the nice kinome dataset in Fig.6A-B would significantly enhance the value and clarity of this figure and results. For example, it would seem that PdMo has massive increase in the activity of these kinases. The interpretation of Fig.6C in contrast is more intuitive and seems to go in the opposite direction as what is shown in Fig.6B.

Response: We thank this Reviewer for pointing this out. Indeed, it missed our attention that Fig 6B was incorrectly colored. As the dataset is redundant with **Fig. 6C**, which shows the cumulative active kinases and their dynamics within StdMo, PdMo, PdMo + Plts, and Plts, we decided to remove the figure completely, and all raw data is provided in Source Data. Furthermore, in the original dataset (Rebuttal Fig.) generated by the support team at PamGene, and used to produce Fig 6B, it can be appreciated that PdMo have the weakest signal for LPS-induced kinase activities. We hope this clarifies the issue.

Rebuttal Fig.

Heatmap representation of the active kinases in StdMo, PdMo, PdMo + Plts, or Plts. Cells were left untreated (-), or stimulated with LPS. This dataset was used to generate the coral trees in Main Fig. 6-A-B.

- Fig8B, the panels for IL-6 and TNF α secretion are identical; I wonder if the panel was duplicated by mistake?

Response: We apologize for these mistakes and thank **Reviewer 2** for bringing them to our attention. Indeed, the graphs were accidentally duplicated. We have now included the correct dataset, supplemented by the original raw data tables (Source Data) accompanying the published version of this manuscript.

References Mentioned in this Rebuttal Letter.

- Beilharz M, De Nardo D, Latz E, Franklin BS (2016) Measuring NLR Oligomerization II: Detection of ASC Speck Formation by Confocal Microscopy and Immunofluorescence. In: *Methods Mol Biol*, pp. 145-158.
- Bertheloot D, Wanderley CW, Schneider AH, Schiffelers LD, Wuerth JD, Todtmann JM, Maasewerd S, Hawwari I, Duthie F, Rohland C et al (2022) Nanobodies dismantle post-pyroptotic ASC specks and counteract inflammation in vivo. *EMBO Mol Med*: e15415
- Bonnet B, Cosme J, Dupuis C, Coupez E, Adda M, Calvet L, Fabre L, Saint-Sardos P, Bereiziat M, Vidal M et al (2021) Severe COVID-19 is characterized by the co-occurrence of moderate cytokine inflammation and severe monocyte dysregulation. *EBioMedicine* 73: 103622
- Chatterjee M, von Ungern-Sternberg SNI, Seizer P, Schlegel F, Büttcher M, Sindhu NA, Müller S, Mack A, Gawaz M (2015) Platelet-derived CXCL12 regulates monocyte function, survival, differentiation into macrophages and foam cells through differential involvement of CXCR4–CXCR7. *Cell Death and Disease* 6: e1989-e1989
- Fajgenbaum DC, June CH (2020) Cytokine Storm. *N Engl J Med* 383: 2255-2273
- Ferreira AC, Soares VC, de Azevedo-Quintanilha IG, Dias SdSG, Fintelman-Rodrigues N, Sacramento CQ, Mattos M, de Freitas CS, Temerozo JR, Teixeira L et al (2021) SARS-CoV-2 engages inflammasome and pyroptosis in human primary monocytes. *Cell Death Discov* 7: 43
- Franklin BS, Bossaller L, De Nardo D, Ratter JM, Stutz A, Engels G, Brenker C, Nordhoff M, Mirandola SR, Al-Amoudi A et al (2014) The adaptor ASC has extracellular and &prionoid&prionoid activities that propagate inflammation. *Nat Immunol* 15: 727-737
- Hargett LA, Bauer NN (2013) On the origin of microparticles: From "platelet dust" to mediators of intercellular communication. *Pulm Circ* 3: 329-340
- Hoss F, Rolfes V, Davanzo MR, Braga TT, Franklin BS (2017) Detection of ASC Speck Formation by Flow Cytometry and Chemical Cross-linking. In: *Methods in Molecular Biology*, pp. 149-165. Springer New York: New York, NY
- Hu Y, Hu X, Boumsell L, Ivashkiv LB (2008) IFN-gamma and STAT1 arrest monocyte migration and modulate RAC/CDC42 pathways. *J Immunol* 180: 8057-8065
- Jafarzadeh A, Chauhan P, Saha B, Jafarzadeh S, Nemat M (2020) Contribution of monocytes and macrophages to the local tissue inflammation and cytokine storm in COVID-19: Lessons from SARS and MERS, and potential therapeutic interventions. *Life Sci* 257: 118102
- Junqueira C, Crespo Â, Ranjbar S, de Lacerda LB, Lewandrowski M, Ingber J, Parry B, Ravid S, Clark S, Schrimpf MR et al (2022) FcγR-mediated SARS-CoV-2 infection of monocytes activates inflammation. *Nature*
- Kasper CA, Sorg I, Schmutz C, Tschon T, Wischniewski H, Kim ML, Arriemerlou C (2010) Cell-Cell Propagation of NF-κB Transcription Factor and MAP Kinase Activation Amplifies Innate Immunity against Bacterial Infection. *Immunity* 33: 804-816
- Lannan KL, Sahler J, Kim N, Spinelli SL, Maggirwar SB, Garraud O, Cognasse F, Blumberg N, Phipps RP (2015) Breaking the mold: transcription factors in the anucleate platelet and platelet-derived microparticles. *Front Immunol* 6: 48
- Lari R, Kitchener PD, Hamilton JA (2009) The proliferative human monocyte subpopulation contains osteoclast precursors. *Arthritis Res Ther* 11: R23
- Poli V, Di Gioia M, Sola-Visner M, Granucci F, Frelinger AL, 3rd, Michelson AD, Zanoni I (2021) Inhibition of transcription factor NFAT activity in activated platelets enhances their aggregation and exacerbates gram-negative bacterial septicemia. *Immunity*
- Rauert-Wunderlich H, Siegmund D, Maier E, Giner T, Bargou RC, Wajant H, Stuhmer T (2013) The IKK inhibitor Bay 11-7082 induces cell death independent from inhibition of activation of NFκappaB transcription factors. *PLoS ONE* 8: e59292

- Rolfes V, Ribeiro LS, Hawwari I, Bottcher L, Rosero N, Maasewerd S, Santos MLS, Prochnicki T, Silva CMS, Wanderley CWS et al (2020) Platelets Fuel the Inflammasome Activation of Innate Immune Cells. *Cell Rep* 31: 107615
- Scheuerer B, Ernst M, Durrbaum-Landmann I, Fleischer J, Grage-Griebenow E, Brandt E, Flad HD, Petersen F (2000) The CXC-chemokine platelet factor 4 promotes monocyte survival and induces monocyte differentiation into macrophages. *Blood* 95: 1158-1166
- Schulte-Schrepping J, Reusch N, Paclik D, Bassler K, Schlickeiser S, Zhang B, Kramer B, Krammer T, Brumhard S, Bonaguro L et al (2020) Severe COVID-19 Is Marked by a Dysregulated Myeloid Cell Compartment. *Cell* 182: 1419-1440 e1423
- Shih VF, Davis-Turak J, Macal M, Huang JQ, Ponomarenko J, Kearns JD, Yu T, Fagerlund R, Asagiri M, Zuniga EI, Hoffmann A (2012) Control of RelB during dendritic cell activation integrates canonical and noncanonical NF-kappaB pathways. *Nat Immunol* 13: 1162-1170
- Sreeramkumar V, Adrover JM, Ballesteros I, Cuartero MI, Rossaint J, Bilbao I, Nacher M, Pitaval C, Radovanovic I, Fukui Y et al (2014) Neutrophils scan for activated platelets to initiate inflammation. *Science* 346: 1234-1238
- Stutz A, Horvath GL, Monks BG, Latz E (2013) ASC speck formation as a readout for inflammasome activation. In: *Methods Mol Biol*, pp. 91-101.
- Swanson JA (2008) Shaping cups into phagosomes and macropinosomes. *Nat Rev Mol Cell Biol* 9: 639-649
- Underhill DM, Goodridge HS (2012) Information processing during phagocytosis. *Nat Rev Immunol* 12: 492-502
- Vallabhapurapu S, Karin M (2009) Regulation and function of NF-kappaB transcription factors in the immune system. *Annu Rev Immunol* 27: 693-733
- Vanderbeke L, Van Mol P, Van Herck Y, De Smet F, Humblet-Baron S, Martinod K, Antoranz A, Arijs I, Boeckx B, Bosisio FM et al (2021) Monocyte-driven atypical cytokine storm and aberrant neutrophil activation as key mediators of COVID-19 disease severity. *Nat Commun* 12: 4117
- Zuchriegel G, Uhl B, Pühr-Westerheide D, Pörnbacher M, Lauber K, Krombach F, Reichel CA (2016) Platelets Guide Leukocytes to Their Sites of Extravasation. *PLoS Biol* 14: e1002459

28th May 2024

Dear Prof. Franklin,

Thank you for the submission of your revised manuscript to EMBO Molecular Medicine. I am pleased to inform you that we will be able to accept your manuscript pending the following final amendments:

- 1) Please address referee 2 minor comments and consider shortening the Discussion as suggested.
- 2) Authors: We note a discrepancy of author names: Nathalia Sofia Rosero Reyes in the manuscript file vs. Nathalia R Rosero in our system, Matilde B Vasconcelos in the manuscript vs. Matilda B Vasconcelos in the system and Lino L Teichmann in the manuscript file vs. Lino L Teichman in the system. Please correct.
- 3) Author checklist: Please submit completed checklist. Currently there are many fields without selected response.
- 4) In the main manuscript file, please do the following:
 - Please address all comments suggested by our data editors listed below:
 - o Figure legends:
 1. Please note that a separate 'Data Information' section is required in the legends of figures 8b-d; EV 2c-d; EV 4b-d; EV 5b, d.
 2. Please note that the legend for figure 6b-c is incorrectly labelled as 6b-d. This needs to be rectified.
 3. Please note that the legend for figure 7f-g is incorrectly labelled as 7f. This needs to be rectified.
 4. Please note that the legend for figure 7h is incorrectly labelled as 7g. This needs to be rectified.
 5. Please define the annotated p values ******/****** in the legend of figure 5c; as appropriate.
 6. Please indicate the statistical test used for data analysis in the legends of figures 1a, d-g; 3b, d; 4c, e; 5d; 6d; 7b; 8b-d; EV 1j; EV 4f; EV 5d.
 7. Please note that in figures EV 2c-d; there is a mismatch between the annotated p values in the figure legend and the annotated p values in the figure file that should be corrected.
 8. Please note that the box plots need to be defined in terms of minima, maxima, centre, bounds of box and percentile in the legends of figures 1a, d-e; 3d; 4d; 6b; EV 4f.
 9. Please note that the box plots need to be defined in terms of minima, maxima, centre, bounds of box, and whiskers and percentile in the legends of figures 1f-h; 2d-g; 5c.
 10. Please note that the box plots need to be defined in terms of bounds of box, and percentile in the legends of figures 5f; 6a; 7b, d-e, h; 8b-d; EV 1e, j; EV 2c-d; EV 4b-d EV 5b, d.
 11. Please note that information related to n is missing in the legends of figures 1a, e; 2d; 3d; 4d; 6b; 7b, d; EV 4f.
 12. Please note that the error bars are not defined in the legends of figures EV 1b, i; EV 3b.
 - Add callouts for Fig 3C-D, Fig 7F and updated Supplementary Fig. S6 to the correct callout Appendix Fig. S6.
 - Author contributions: Please remove it from the manuscript and specify author contributions in our submission system. CRediT has replaced the traditional author contributions section because it offers a systematic machine-readable author contributions format that allows for more effective research assessment. You are encouraged to use the free text boxes beneath each contributing author's name to add specific details on the author's contribution. More information is available in our guide to authors: <https://www.embopress.org/page/journal/17574684/authorguide#authorshipguidelines>
 - Indicate in legends number and nature of replicates and exact p= values, not a range, along with the statistical test used. To keep the figures "clear" some authors found providing an Appendix table Sx with all exact p-values preferable. You are welcome to do this if you want to.
 - Provide the statement that in addition to the WMA Declaration of Helsinki the experiments also conformed to the principles set out in the Department of Health and Human Services Belmont Report.
 - Statistical paragraph should reflect all information that you have filled in the Authors Checklist, especially regarding randomization, blinding, replication.
 - Remove "Data sharing availability".
 - Please complete the data availability statement. The following sentences should be removed: "All other data supporting the findings of this study are included in this manuscript. Source data are provided with this paper". Use the following format to report the accession number of your deposited data including source data:

[data type]: [full name of the resource] [accession number/identifier] [(doi or URL or identifiers.org/DATABASE:ACCESSION)]

Please check "Author Guidelines" for more information.

<https://www.embopress.org/page/journal/17574684/authorguide#availabilityofpublishedmaterial>

5) Funding: Please fuse it with "Acknowledgement" and make sure that information about all sources of funding are complete in both our submission system and in the manuscript. Currently, DFG's grants: 388158066, 216372545 and 388159768 are missing in our submission system.

6) The Paper Explained: Please provide "The Paper Explained" and add it to the main manuscript text. Please check "Author Guidelines" for more information. <https://www.embopress.org/page/journal/17574684/authorguide#researcharticleguide>

7) Synopsis: Every published paper now includes a 'Synopsis' to further enhance discoverability. Synopses are displayed on the journal webpage and are freely accessible to all readers. They include separate synopsis image and synopsis text.

- Synopsis image: Please provide a striking image or visual abstract as a high-resolution jpeg file 550 px-wide x (250-400)-px high to illustrate your article.
 - Synopsis text: Please provide a short standfirst (maximum of 300 characters, including space) as well as 2-5 one sentence bullet points that summarise the paper as a .doc file. Please write the bullet points to summarise the key NEW findings. They should be designed to be complementary to the abstract - i.e. not repeat the same text. We encourage inclusion of key acronyms and quantitative information (maximum of 30 words / bullet point). Please use the passive voice.
 - Please check your synopsis text and image before submission with your revised manuscript. Please be aware that in the proof stage minor corrections only are allowed (e.g., typos).
- 8) For more information: This space should be used to list relevant web links for further consultation by our readers. Could you identify some relevant ones and provide such information as well? Some examples are patient associations, relevant databases, OMIM/proteins/genes links, author's websites, etc...
- 9) As part of the EMBO Publications transparent editorial process initiative (see our Editorial at <http://embomolmed.embopress.org/content/2/9/329>), EMBO Molecular Medicine will publish online a Review Process File (RPF) to accompany accepted manuscripts. This file will be published in conjunction with your paper and will include the anonymous referee reports, your point-by-point response and all pertinent correspondence relating to the manuscript. Let us know whether you agree with the publication of the RPF and as here, if you want to remove or not any figures from it prior to publication. Please note that the Authors checklist will be published at the end of the RPF.
- 10) Please provide a point-by-point letter INCLUDING my comments as well as the reviewer's reports and your detailed responses (as Word file).

I look forward to reading a new revised version of your manuscript as soon as possible.

Yours sincerely,

Zeljko Durdevic

*** Instructions to submit your revised manuscript ***

To submit your manuscript, please follow this link:

- 1) a .docx formatted version of the manuscript text (including Figure legends and tables)
- 2) Separate figure files*
- 3) supplemental information as Expanded View and/or Appendix. Please carefully check the authors guidelines for formatting Expanded view and Appendix figures and tables at <https://www.embopress.org/page/journal/17574684/authorguide#expandedview>
- 4) a letter INCLUDING the reviewer's reports and your detailed responses to their comments (as Word file).
- 5) The paper explained: EMBO Molecular Medicine articles are accompanied by a summary of the articles to emphasize the

major findings in the paper and their medical implications for the non-specialist reader. Please provide a draft summary of your article highlighting

6) For more information: There is space at the end of each article to list relevant web links for further consultation by our readers. Could you identify some relevant ones and provide such information as well? Some examples are patient associations, relevant databases, OMIM/proteins/genes links, author's websites, etc...

7) Author contributions: the contribution of every author must be detailed in a separate section.

8) EMBO Molecular Medicine now requires a complete author checklist (<https://www.embopress.org/page/journal/17574684/authorguide>) to be submitted with all revised manuscripts. Please use the checklist as guideline for the sort of information we need WITHIN the manuscript. The checklist should only be filled with page numbers where the information can be found. This is particularly important for animal reporting, antibody dilutions (missing) and exact values and n that should be indicated instead of a range.

9) Every published paper now includes a 'Synopsis' to further enhance discoverability. Synopses are displayed on the journal webpage and are freely accessible to all readers. They include a short stand first (maximum of 300 characters, including space) as well as 2-5 one sentence bullet points that summarise the paper. Please write the bullet points to summarise the key NEW findings. They should be designed to be complementary to the abstract - i.e. not repeat the same text. We encourage inclusion of key acronyms and quantitative information (maximum of 30 words / bullet point). Please use the passive voice. Please attach these in a separate file or send them by email, we will incorporate them accordingly.

You are also welcome to suggest a striking image or visual abstract to illustrate your article. If you do please provide a jpeg file 550 px-wide x 300-800px high.

10) A Conflict of Interest statement should be provided in the main text

11) Please note that we now mandate that all corresponding authors list an ORCID digital identifier. This takes <90 seconds to complete. We encourage all authors to supply an ORCID identifier, which will be linked to their name for unambiguous name identification.

Currently, our records indicate that the ORCID for your account is 0000-0003-2591-9833.

Link Not Available

Photos 400-800 DPI

*Additional important information regarding figures and illustrations can be found at

<https://bit.ly/EMBOPressFigurePreparationGuideline>. See also figure legend preparation guidelines:

<https://www.embopress.org/page/journal/17574684/authorguide#figureformat>

***** Reviewer's comments *****

Referee #1 (Comments on Novelty/Model System for Author):

Results are highly relevant for the human population. In people with low platelet counts (thrombocytopenia), monocytes are hypo-responsive and show defective/weakened cytokine responses to immune challenges. A mechanism to explain this biology

is offered in this study.

Referee #1 (Remarks for Author):

The authors have addressed my questions, and the revised manuscript is strong. Even when experiments proved technically challenging, they have provided data and clarified where technical limitations are, in the rebuttal letter.

Referee #2 (Comments on Novelty/Model System for Author):

This is a technical tour de force analysis of the monocyte-platelet interactions with the very exciting finding that platelet products are sufficient to for full activation of monocytes shown in multiple ways. The physiological context is a bit less clear but I feel the findings to be robust and an important contribution to the field.

Referee #2 (Remarks for Author):

The revised paper is significantly improved and is overall a technically remarkable study with a number of very interesting observations, and a clear message on the transfer of signaling molecules from platelets to monocytes. I therefore endorse publication.

A minor comment is that given the limited information on in vivo the platelet depletion in mice, probably the result of technical artifacts introduced by the depletion method, the section should be greatly reduced, and the conclusions should state not only the transcriptional changes but also its limitations. As is, I find that this section only distracts from the message of the paper. Finally, the Discussion is too long but paradoxically does not touch on what I think is the critical question of why such mechanism of factor relay has evolved in parallel with the own cell (the monocyte) being already able to produce the same (or similar) factors; why so much redundancy? Regardless on whether the authors would like to add ideas about this on the paper, I think the Discussion could be significantly shortened.

EMM-2023-18988-V3

Manuscript: EMM-2023-18988-V3 - Platelet transcription factors license the pro-inflammatory cytokine response of human monocytes.

Point-by-point response to the Editors and reviewer comments:

Dear Dr. Durdevic, dear editors,

We are grateful that you have considered our manuscript suitable for publication in EMBO Molecular Medicine. Please find below our point-by-point response to your comments.

Editor Comments:

1) Please address referee 2 minor comments and consider shortening the Discussion as suggested.

We highly appreciated the reviewers' comments and their enthusiasm about our work. We have shortened the refereed sessions, and others where we have found redundancies, as suggested. We have also included a brief speculation on the evolutionary benefits of the phenomenon we describe in our study, as recommended by Reviewer 2.

Other modifications included shortening and or complete elimination of repeated statements that we identified between the introduction and discussion, as detailed below:

In lines 96-99 of the introduction of EMM-2023-18988-V2:

"Immunoparalysis is often observed after sepsis (Arens *et al*, 2016; Frazier & Hall, 2008; Roquilly *et al*, 2020), major visceral surgery (Frazier & Hall, 2008), and has recently been associated with the severity of SARS-CoV-2 infections (Agrati *et al*, 2020; Arunachalam *et al*, 2020)."

This was somewhat repeated in lines 559 of the discussion:

"Immunoparalysis often occurs after trauma, major organ surgery (Haupt *et al*, 1998), hemorrhagic fevers (Vangeti *et al*, 2021), sepsis (Cao *et al*, 2019; Weisheit *et al*, 2020), liver failure (Lin *et al*, 2007; Wasmuth *et al*, 2005), and infections (Agrati *et al*, 2020; Arunachalam *et al*, 2020)."

We removed the mentioning from the discussion in the revised EMM-2023-18988-V3.

In lines 105-109 of the introduction (EMM-2023-18988-V2):

"In the bloodstream, monocytes continuously interact with platelets, forming monocyte-platelet aggregates (MPAs) under physiological conditions (Rinder *et al*, 1991). MPAs increase in numerous inflammatory and thrombotic disorders and are usually associated with poor outcomes (Allen *et al*, 2019; Liang *et al*, 2015; Manne *et al*, 2020; Stephen *et al*, 2013).";

And in lines 549 - 553 of the discussion.

"Platelet-monocyte aggregates (MPAs), common in homeostasis (Rinder *et al*, 1991), increase during inflammation, serving as indicators of inflammatory diseases (Allen *et al*, 2019; Barrett *et al*, 2019; Manne *et al*, 2020), and chronic infections (DOI:

10.1126/science.adg7942)(10.1038/s41467-022-35638-y). Despite their prevalence, the precise functions of MPAs remain unclear."

We eliminated this statement from the discussion in the revised EMM-2023-18988-V3.

Lines 109 (EMM-2023-18988-V2) also stated: "Interaction with platelets amplifies several effector functions of monocytes in numerous diseases (D'Mello *et al*, 2017; Fu *et al*, 2021; Rong *et al*, 2014; Singhal *et al*, 2017)."

Which was redundant with: "Interaction with platelets enhances various monocyte functions, through mechanisms that remain partially understood, but with implications in innate and acquired immunity (D'Mello *et al*, 2017; Fu *et al*, 2021; Han *et al*, 2020; Rong *et al*, 2014; Singhal *et al*, 2017)(Kral *et al*, 2016)." in the discussion (EMM-2023-18988-V2).

The sentence was removed from the discussion in the revised EMM-2023-18988-V3.

2) Authors: We note a discrepancy of author names: Nathalia Sofia Rosero Reyes in the manuscript file vs. Nathalia R Rosero in our system, Matilde B Vasconcelos in the manuscript vs. Matilda B Vasconcelos in the system and Lino L Teichmann in the manuscript file vs. Lino L Teichman in the system. Please correct.

Thank you for pointing that out. The errors were corrected in the system and are now matching the names in the manuscript.

3) Author checklist: Please submit completed checklist. Currently there are many fields without selected response.

Thank you. It was done.

4) In the main manuscript file, please do the following:

- Please address all comments suggested by our data editors listed below:

o Figure legends:

1. Please note that a separate 'Data Information' section is required in the legends of figures 8b-d; EV 2c-d; EV 4b-d; EV 5b, d.

Thank you. It was done accordingly.

2. Please note that the legend for figure 6b-c is incorrectly labelled as 6b-d. This needs to be rectified.

Thank you. It was done.

3. Please note that the legend for figure 7f-g is incorrectly labelled as 7f. This needs to be rectified.

Thank you. It was done.

4. Please note that the legend for figure 7h is incorrectly labelled as 7g. This needs to be rectified.

Also done.

5. Please define the annotated p values */** in the legend of figure 5c; as appropriate.**

Thank you. It was done. Change highlighted in yellow in the revised manuscript.

6. Please indicate the statistical test used for data analysis in the legends of figures 1a, d-g; 3b, d; 4c, e; 5d; 6d; 7b; 8b-d; EV 1j; EV 4f; EV 5d.

It was done.

7. Please note that in figures EV 2c-d; there is a mismatch between the annotated p values in the figure legend and the annotated p values in the figure file that should be corrected.

It was done

8. Please note that the box plots need to be defined in terms of minima, maxima, centre, bounds of box and percentile in the legends of figures 1a, d-e; 3d; 4d; 6b; EV 4f.

Dear editors, these are Floating Bars, and not Box Plots.

Floating Bars are an different modality of data visualization which includes the full distribution of the data points (instead of ranges and whiskers/percentile in box plots). Therefore floating bars usually show all values (min to max) and the mean not needing the whiskers and bands (see illustration below). We also provide all the raw data used in these figures in the Source Data material that accompany this manuscript. Changing all graphs to box plots would unfortunately require us to remake all graph figures.

9. Please note that the box plots need to be defined in terms of minima, maxima, centre, bounds of box, and whiskers and percentile in the legends of figures 1f-h; 2d-g; 5c.

Most of our figures in the manuscript are Floating bars (example on the left panel) which shows the complete distribution of data points, instead of ranges (whiskers in an box plot, right panel) and percentiles. Conventionally floating bars display min to max distribution and the mean or median.

See answer above.

10. Please note that the box plots need to be defined in terms of bounds of box, and percentile in the legends of figures 5f; 6a; 7b, d-e, h; 8b-d; EV 1e, j; EV 2c-d; EV 4b-d EV 5b, d.

Same as above.

11. Please note that information related to n is missing in the legends of figures 1a, e; 2d; 3d; 4d; 6b; 7b, d; EV 4f.

We have now included them. Thank you.

12. Please note that the error bars are not defined in the legends of figures EV 1b, i; EV 3b.

Thank you. The refereed error bars were now indicated in the figure legends.

- Add callouts for Fig 3C-D, Fig 7F and updated Supplementary Fig. S6 to the correct callout Appendix Fig. S6.

Call outs were included in the respective and relevant parts of the results.

Fig. 3C (line: 273)

Fig. 3D (Line: 268)

Fig. 7F (line: 470)

Appendix Fig. S6 was also corrected.

- Author contributions: Please remove it from the manuscript and specify author contributions in our submission system. CRediT has replaced the traditional author contributions section because it offers a systematic machine-readable author contributions format that allows for more effective research assessment. You are encouraged to use the free text boxes beneath each contributing author's name to add specific details on the author's contribution. More information is available in our guide to authors: <https://www.embopress.org/page/journal/17574684/authorguide#authorshipguidelines>

The author contribution statement was removed from the main text of the manuscript.

Thank you. We have removed it from the main text and indicated in the submission system, with specific descriptions wherever necessary.

- Indicate in legends number and nature of replicates and exact p= values, not a range, along with the statistical test used. To keep the figures "clear" some authors found providing an Appendix table Sx with all exact p-values preferable. You are welcome to do this if you want to.

- Provide the statement that in addition to the WMA Declaration of Helsinki the experiments also conformed to the principles set out in the Department of Health and Human Services Belmont Report.

Done.

- Statistical paragraph should reflect all information that you have filled in the Authors Checklist, especially regarding randomization, blinding, replication.

- Remove "Data sharing availability".

Done.

- Please complete the data availability statement. The following sentences should be

removed: "All other data supporting the findings of this study are included in this manuscript. Source data are provided with this paper". Use the following format to report the accession number of your deposited data including source data:

Done.

The datasets produced in this study are available in the following databases: [data type]: [full name of the resource] [accession number/identifier] ([doi or URL or identifiers.org/DATABASE:ACCESSION])

Done.

Please check "Author Guidelines" for more information.

<https://www.embopress.org/page/journal/17574684/authorguide#availabilityofpublishedmaterial>

5) Funding: Please fuse it with "Acknowledgement" and make sure that information about all sources of funding are complete in both our submission system and in the manuscript. Currently, DFG's grants: 388158066, 216372545 and 388159768 are missing in our submission system.

Funding was now merged with Acknowledgments and the missing grant identifications were included in the system.

6) The Paper Explained: Please provide "The Paper Explained" and add it to the main manuscript text. Please check "Author Guidelines" for more information.

<https://www.embopress.org/page/journal/17574684/authorguide#researcharticleguide>

Done. See suggestion in the main manuscript text.

7) Synopsis: Every published paper now includes a 'Synopsis' to further enhance discoverability. Synopses are displayed on the journal webpage and are freely accessible to all readers. They include separate synopsis image and synopsis text.

- Synopsis image: Please provide a striking image or visual abstract as a high-resolution jpeg file 550 px-wide x (250-400)-px high to illustrate your article.

Done! We suggest this image, which was uploaded to the submission system.

- Synopsis text: Please provide a short standfirst (maximum of 300 characters, including space) as well as 2-5 one sentence bullet points that summarise the paper as a .doc file.

Done! We suggest this text:

In the circulation, platelets interact with blood monocytes enhancing their inflammatory capacity. Platelets are enriched in inflammatory transcriptional regulators which sustain human

monocyte inflammation. Thrombocytopenia, experimentally induced, or as a result of autoimmune diseases, hinders monocytes anergic to immune stimulation, weakening host defence programs.

Please write the bullet points to summarise the key NEW findings. They should be designed to be complementary to the abstract - i.e. not repeat the same text. We encourage inclusion of key acronyms and quantitative information (maximum of 30 words / bullet point). Please use the passive voice.

Here are our suggestions:

- Platelets are essential to PRR-induced cytokine responses of human monocytes,
- Immune thrombocytopenia leads to monocyte immunoparalysis;
- Platelet supplementation reverses monocyte immunoparalysis;
- Platelets overcome NF-kB2 and p38 MAPK ablation in human monocytes.

Done.

8) For more information: This space should be used to list relevant web links for further consultation by our readers. Could you identify some relevant ones and provide such information as well? Some examples are patient associations, relevant databases, OMIM/proteins/genes links, author's websites, etc...

9) As part of the EMBO Publications transparent editorial process initiative (see our Editorial at <http://embomolmed.embopress.org/content/2/9/329>), EMBO Molecular Medicine will publish online a Review Process File (RPF) to accompany accepted manuscripts. This file will be published in conjunction with your paper and will include the anonymous referee reports, your point-by-point response and all pertinent correspondence relating to the manuscript. Let us know whether you agree with the publication of the RPF and as here, if you want to remove or not any figures from it prior to publication. Please note that the Authors checklist will be published at the end of the RPF.

We agree with the publication of all content related to this manuscript, with a few exceptions of datasets included in the Rebuttal letter "*Point-by-point Response to Reviewers_LR_BF.pdf*" in the first revision (EMM-2023-18988-V2). Those were already discussed with the editor before the submission of the revised manuscript.

- **Rebuttal Fig. 4** - Bulk-RNA-Seq data of primary human monocyte-derived macrophages...
- **Rebuttal Fig. 5** - Seahorse assays on human monocyte-derived macrophages.

10) Please provide a point-by-point letter INCLUDING my comments as well as the reviewer's reports and your detailed responses (as Word file).

Done.

I look forward to reading a new revised version of your manuscript as soon as possible.

Referee #1 (Comments on Novelty/Model System for Author):

Results are highly relevant for the human population. In people with low platelet counts (thrombocytopenia), monocytes are hypo-responsive and show defective/weakened cytokine responses to immune challenges. A mechanism to explain this biology is offered in this study.

Referee #1 (Remarks for Author):

The authors have addressed my questions, and the revised manuscript is strong. Even when experiments proved technically challenging, they have provided data and clarified where technical limitations are, in the rebuttal letter.

We thank this reviewer for his time and comments.

Referee #2 (Comments on Novelty/Model System for Author):

This is a technical tour de force analysis of the monocyte-platelet interactions with the very exciting finding that platelet products are sufficient to for full activation of monocytes shown in multiple ways. The physiological context is a bit less clear but I feel the findings to be robust and an important contribution to the field.

Referee #2 (Remarks for Author):

The revised paper is significantly improved and is overall a technically remarkable study with a number of very interesting observations, and a clear message on the transfer of signaling molecules from platelets to monocytes. I therefore endorse publication.

A minor comment is that given the limited information on in vivo the platelet depletion in mice, probably the result of technical artifacts introduced by the depletion method, the section should be greatly reduced, and the conclusions should state not only the transcriptional changes but also its limitations. As is, I find that this section only distracts from the message of the paper. Finally, the Discussion is too long but paradoxically does not touch on what I think is the critical question of why such mechanism of factor relay has evolved in parallel with the own cell (the monocyte) being already able to produce the same (or similar) factors; why so much redundancy? Regardless on whether the authors would like to add ideas about this on the paper, I think the Discussion could be significantly shortened.

We highly appreciated this reviewer's comments.

As suggested, the results section of the mouse model was reduced. Additionally, the discussion section was shortened (from 1847 words in V2 to 1232 words in V3), and now includes a statement about the possible evolutionary origins and/or benefits of the phenomenon we describe here. Changes are highlighted in yellow.

6th Jun 2024

Dear Prof. Franklin,

We are pleased to inform you that your manuscript is accepted for publication and is now being sent to our publisher to be included in the next available issue of EMBO Molecular Medicine.
